# Boosting Open Set Recognition Performance through Modulated Representation Learning

**Amit Kumar Kundu[1,3], Vaishnavi S Patil[2,3] & Joseph Jaja[1,3]**
[1]Department of Electrical and Computer Engineering, [2]Department of Computer Science,
[3]Institute for Health Computing
University of Maryland, College Park, MD 20742, USA
`{amit314,vspatil,josephj}@umd.edu`

## Abstract

The open set recognition (OSR) problem aims to identify test samples from novel semantic classes that are not part of the training classes, a task that is crucial in many practical scenarios. However, the existing OSR methods use a constant scaling factor (the temperature) to the logits before applying a loss function, which hinders the model from exploring both ends of the spectrum in representation learning – from instance-level to class-specific features. In this paper, we address this problem by enabling temperature-modulated representation learning using a set of proposed temperature schedules, including our novel negative cosine schedule. Our temperature schedules allow the model to form a coarse decision boundary at the beginning of training by focusing on fewer neighbors, and gradually prioritizing more neighbors to smooth out the rough edges. This gradual task switching leads to a richer and more generalizable representation space. While other OSR methods benefit by including regularization or auxiliary negative samples, such as with mix-up, thereby adding a significant computational overhead, our schedules can be folded into any existing OSR loss function with no overhead. We implement the novel schedule on top of a number of baselines, using cross-entropy, contrastive and the ARPL loss functions and find that it boosts both the OSR and the closed set performance in most cases, especially on the tougher semantic shift benchmarks [1].

## 1 Introduction

Deep learning models have shown impressive performance by learning useful representations particularly for tasks involving the classification of examples into categories present in the training dataset, also known as the closed set. However during inference, in many practical scenarios, test samples may appear from unknown classes (termed as the open set), which were not a part of the training set. Hence, a more realistic task known as the open set recognition (OSR) (Scheirer et al. (2012); Chen et al. (2020a)) aims to simultaneously flag the test samples from unknown classes while accurately classifying examples from the known classes, requiring strong generalization beyond the support of training data.

Most of the early research attempts either model the unknown classes as long-tailed distributions (Vignotto & Engelke, 2018; Bendale & Boult, 2016), generate synthetic samples using generative models (Ge et al., 2017; Neal et al., 2018; Chen et al., 2021; Moon et al., 2022) or with mix-up (Chen et al., 2021; Xu et al., 2023; Li et al., 2024; Zhou et al., 2021) to represent novel classes, or train a secondary model with a separate objective, such as VAEs that include reconstruction based objective (Oza & Patel (2019); Yoshihashi et al. (2019); Zhou et al. (2024a)). The synthetic examples may not generalize well to a variety of unknown classes, whereas training generative or secondary models or with mix-up are computationally demanding and often require higher memory. Later methods add regularization (Zhou et al., 2021; Chen et al., 2021; 2020a) to explicitly bound the open space risks. In essence, these methods create more empty regions in the representation space by pushing the decision boundary tighter and hoping that unknown representations lie in those regions. Forcing the creation of empty spaces does not result in an improved OSR as it does not address the inherent

---

[1]The project codes are available at https://github.com/amit31416/NegCosSch/.

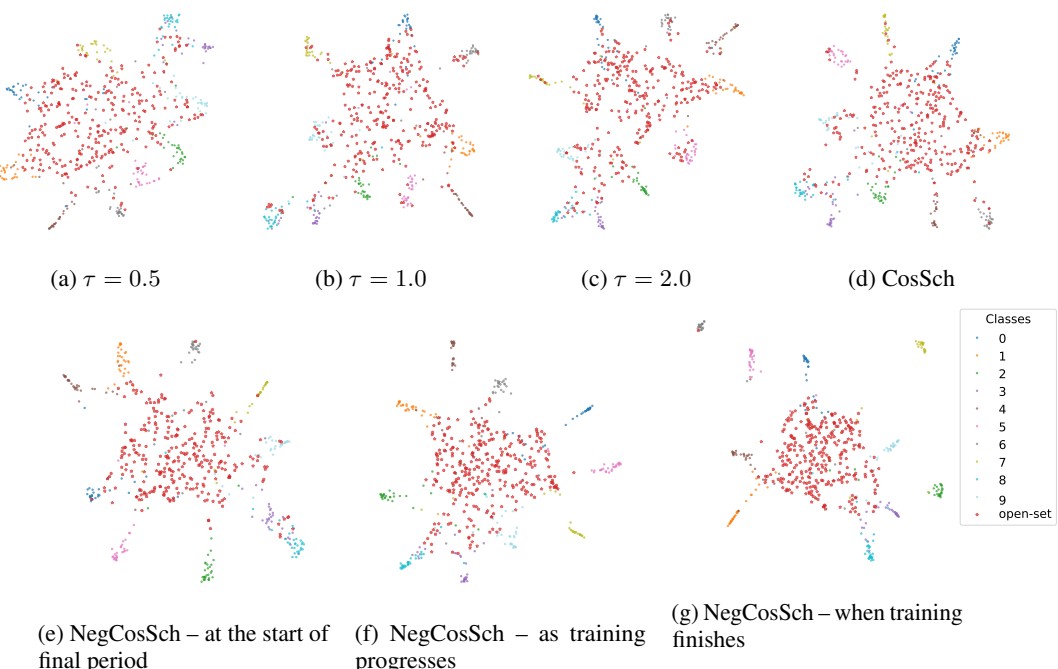

(a) $\tau = 0.5$      (b) $\tau = 1.0$      (c) $\tau = 2.0$      (d) CosSch

(e) NegCosSch – at the start of final period      (f) NegCosSch – as training progresses      (g) NegCosSch – when training finishes

Figure 1: UMAP projection of representation spaces for different temperature schedules on 10 classes of the Caltech-UCSD-Birds dataset. (a)- (c) show representations for constant temperatures ($\tau$). For lower $\tau$, the representations of unknown and known classes are distributed, leading to a sharp decision boundary. For higher $\tau$, the representations of known classes are more compact, making the decision boundary smoother. However, unknown samples overlap with the clusters of known classes. Mid value of $\tau$ achieves a trade-off but does not gain the benefits of both ends. (e)- (g) show representations of our temperature schedule NegCosSch as the training progresses. A lower $\tau$ at the start leads to a coarse decision and the model gradually makes the classes more compact and the unknown representations are pushed away. Finally, (d) show the representation space for a previous schedule CosSch, which is better than fixed temperatures but not as compact as our NegCosSch. Clustering diagnosis appears in Appendix E.1 and the experiment details appear in Appendix C.

semantic proximity of tougher unknown samples to the known classes, incurring significant similarity between them and reducing the effectiveness of such methods.

Vaze et al. (2022) establish new OSR baselines by training models with optimal design choices and argue that a well-trained closed set classifier achieves an improved OSR performance, where the unknown samples exhibit lower max-logit scores. This essentially has motivated the next generation of OSR methods to learn even better representations for improving performance through a better loss function, such as the contrastive loss (Khosla et al. (2020); Chen et al. (2020b)) with sample mix-up (Xu et al. (2023); Verma et al. (2018); Zhang et al. (2017)) and by adding different regularization schemes (Zhou et al. (2024a); Bahavan et al. (2025); Li et al. (2025); Wang et al. (2025)).

Moreover, the regular OSR benchmarks commonly used are small in scale. In this regard, semantic shift benchmarks (SSBs) are proposed by Vaze et al. (2022) on fine-grained datasets, having more classes with varying levels of OSR difficulty. Therefore, the methods that demonstrate improvement on smaller datasets but involve either data generation, mix-up, or training secondary models are unsuitable for the larger benchmarks as training a well-performing base model on them requires a significant compute and memory. Most of the latest research does not use these benchmarks. This necessitates the development of an advanced representation learning scheme that impose minimal computational overhead.

To achieve this, we need to explore the inner mechanisms of the losses that are the basis of most OSR methods, such as the cross-entropy (CE) and the contrastive loss. These loss functions compute probabilities by applying a temperature scaling to the logits- the model's raw outputs- where the

temperature coefficient adjusts the sharpness of resulting probability distributions. It is the key parameter to control the learned features for both losses. Prior works (Wang & Liu, 2021; Zhang et al., 2022; Kukleva et al., 2023; Zhang et al., 2021) demonstrate that a lower temperature encourages instance-specific representations while a higher value encourages class-specific ones. However, a fixed temperature throughout the training prevents the model from exploring both ends of this learning spectrum. In this regard, Kukleva et al. (2023) study the benefits of learning both instance-level and class-specific features primarily in the closed-set scenarios using self-supervised contrastive learning for long-tailed datasets using a cosine temperature schedule (TS), but the impact of temperature scaling or a TS remains largely unexamined for novel classes and for the context of different losses.

In OSR, learning a representation space that provides both instance-specific and class-specific features is also crucial to achieve improved open set and closed set performance. In this research, we analyze the representation space for different temperature scaling factors on both losses in an open set scenario. Based on the analysis, we propose novel temperature schedules for temperature modulated representation learning. We find temperature modulation with the proposed schedules is beneficial to create more compact clusters for representing the closed set classes, while keeping open set examples more distant from these clusters, resulting in overall improved representations.

The main contributions of this paper are summarized as follows:

- We analyze the effects of temperature scaling in an open set scenario, using a number of TSs, including our novel negative cosine schedule (NegCosSch), to explore temperature modulated representation learning. We find that the proposed schedules, even simple linear schedules, demonstrate better open and closed set performance compared to the usual constant temperature baselines and possible other schedules.

- Our schedules can be seamlessly integrated into any existing OSR loss, such as the CE, the losses based on contrastive learning and the ARPL loss by Chen et al. (2021), without any computational overhead. We show significant performance improvements on the TinyImageNet benchmark and the SSBs.

- Our strategy demonstrates strong performance improvements for the tougher SSBs over the baselines for both the closed set and the open set problems. We show that our scheme achieves stronger improvements with an increased number of training classes when the task becomes more difficult for the baseline model.

The rest of the paper is organized as follows. In Section 2, we discuss the relevant background on different losses and in Section 3, we discuss the effect of temperature scaling on known and unknown classes. In Section 4, we describe the proposed scheme. In Section 5, we discuss our results followed by related works in Section 6 and present the concluding remarks in Section 7.

## 2 BACKGROUND ON LOSSES

During training, the model is decomposed into two components: The first component is an encoder function $f(\cdot)$ which maps the input $x$ to a representation $z = f(x)$. The second component $h(\cdot)$ maps the representations to task specific outputs, which is either a linear classification layer if we train with the CE loss or a projection layer if we use the contrastive training. The final outputs, also called the logits, $l = h(z) = h(f(z))$ are then given as model predictions to the loss functions. We assume for a specific problem, a model is trained for a predefined number of epochs $E$. We further discuss the CE and the supervised contrastive (SupCon) losses and the effects of the temperature parameter, which provide a basis for our proposed scheme. The ARPL loss is discussed in Appendix D.

### 2.1 CROSS-ENTROPY LOSS

For a batch of training data $\mathcal{B} = \{(x_k, y_k)\}_{k=1}^{B}$, the CE loss is calculated as

$$L_{\text{CE}} = -\frac{1}{|\mathcal{B}|} \sum_{(x_k, y_k) \in \mathcal{B}} \text{one\_hot}(y_k) \cdot \log(p_k) \tag{1}$$

where $p_k = \text{softmax}(l_k/\tau)$, one\_hot$(y_k)$ is the one hot encoded vector of $y_k$ and $(\cdot)$ is the dot product. The parameter $\tau > 0$ is called the temperature.

## 2.2 SUPERVISED CONTRASTIVE LOSS

For a given batch $\mathcal{B}$, the SupCon training utilizes a multi-viewed batch by taking two augmented samples of the same original sample. The multi-viewed batch $\mathcal{B}' = \{(\tilde{x}_i, \tilde{y}_i)\}_{i=1}^{2B}$, where $\tilde{x}_{2k}$ and $\tilde{x}_{2k-1}$ are two random augmentations of $x_k$ $(1 \le k \le B)$, and $\tilde{y}_{2k-1} = \tilde{y}_{2k} = y_k$. Below we refer to $i$ as the anchor index from $I = \{1, ..., 2B\}$. The SupCon loss (Khosla et al. (2020)) is defined by:

$$L_{\text{SupCon}} = -\frac{1}{|I|} \sum_{i \in I} \frac{1}{|P(i)|} \left[ \sum_{p \in P(i)} \log \frac{\exp(\text{sim}(l_i, l_p)/\tau)}{\sum_{a \in A(i)} \exp(\text{sim}(l_i, l_a)/\tau)} \right] \qquad (2)$$

Here, $\text{sim}(l_i, l_j)$ is the cosine similarity between $l_i$ and $l_j$ and $A(i) = I \setminus \{i\}$ is the set of all samples in $\mathcal{B}'$ except $i$. $P(i) = \{p \ne i : \tilde{y}_i = \tilde{y}_p\}$ is the set of indices in $\mathcal{B}'$ having the same label as $\tilde{y}_i$ and distinct from $i$. Contrastive loss, by construction, gains its strength by pushing away the representations of the negative samples (samples of other classes) and by producing compact clusters of representations for the (positive) samples of the same class.

# 3 EFFECT OF TEMPERATURE ON KNOWN AND UNKNOWN SAMPLES

The SupCon loss applies hard negative mining from penalizing the harder negative samples more through the exponential function (Khosla et al. (2020)). The measure of hardness of a sample with respect to an anchor is determined by the scaled similarity. Therefore as a scaling factor, the temperature plays a critical role in controlling the trade-off between uniformity and semantic structure in the representation space as shown in Wang & Liu (2021); Kukleva et al. (2023) for the self-supervised loss (Chen et al. (2020b)). This effect mostly translates to the supervised case except for the fact that the definition of positive and negative samples are now different.

For any given anchor index $i$, the gradient of $L_{SupCon}$ with respect to a negative logit $l_j$ can be computed as shown in the following equation.

$$\frac{\partial L_{SupCon}}{\partial l_j} = \frac{\partial L_{SupCon}}{\partial \text{sim}(l_i, l_j)} \times \frac{\partial \text{sim}(l_i, l_j)}{\partial l_j} = \frac{1}{\tau}[\text{softmax}_{a \in I \setminus \{i\}}(\text{sim}(l_i, l_a)/\tau)]_j \times \frac{\partial \text{sim}(l_i, l_j)}{\partial l_j} \quad (3)$$

For smaller values of temperature $\tau$, as the differences in scaled similarity get amplified, the nearest negative samples receive the highest gradient (Wang & Liu (2021)) and the model minimizes similarity to them with respect to anchor $i$. The model aggressively pushes the nearest negative samples away, leading to features that are appropriate for instance-level discrimination and distributing the embeddings over the representation space. However, the positive samples do not cluster tightly because, like the negative samples, fewer positive neighbors get priority in the loss function (Figure 1a). The resulting decision boundary is sharper. The open set representations do not get closer to the known classes due to the heavy penalty of having slight dissimilarity.

With larger $\tau$, the differences in scaled similarity diminish and the repulsive force gets distributed to more negative neighbors. The model can decrease the loss by learning the class-specific features rather than the instance discriminating features to push away easy negatives, inducing semantic structures. Due to compact clusters of within-class representations, the resulting decision boundary is smoother. However, as the model is now less aggressive in removing the negatives, a lot of open set examples get close to the known classes (Figure 1c).

Similarly in CE loss, lower values of temperature ($\tau < 1$) leads to a sharper output probability distribution over the training classes (Guo et al. (2017)), while the higher values of $\tau > 1$ makes the output probability distribution smoother.

The value of $\tau$ is usually kept constant throughout the entire training for both losses, which is set either to a predefined value or chosen with hyperparameter tuning.

# 4 PROPOSED METHOD

In this section, we formally introduce our problem, describe our proposed temperature modulation and explain how our schedules lead to learning representations useful for OSR.

## 4.1 PROBLEM DEFINITION

We are given a labeled training dataset $\mathcal{D}_{tr} = \{(x_i, y_i)\} \subset \mathcal{X} \times \mathcal{Y}$, where $x_i$ is the training sample with label $y_i$. $\mathcal{X}$ is the input space and the labels of $\mathcal{D}_{tr}$ come from a closed set label space $\mathcal{Y}$, i.e. $y_i \in \mathcal{Y}, \forall i$. The total number of classes in the closed set is $C = |\mathcal{Y}|$. The test dataset $\mathcal{D}_{test} \subset \mathcal{X} \times \mathcal{Y} \cup \mathcal{O}$ consists of samples whose label space $\mathcal{Y} \cup \mathcal{O}$ is different than $\mathcal{Y}$, and $\mathcal{O} \cap \mathcal{Y} = \emptyset$. $\mathcal{O}$ is the set of unknown classes defined as the open set. The objective of OSR is to classify a test sample among the closed set classes or to flag it as belonging to an unknown class. We assume that information about the nature of unknown classes or any auxiliary samples are unavailable during training.

## 4.2 RATIONALE FOR TEMPERATURE SCHEDULING

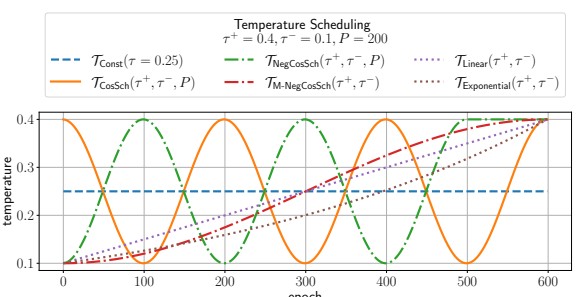

Figure 2: Different temperature schedules.

For a temperature dependent loss function, $\tau$ is usually kept constant throughout the training process. We denote a constant TS by $\mathcal{T}_{\text{Const}}$, defined as

$$\mathcal{T}_{\text{Const}}(e; \tau) = \tau, \forall e \qquad (4)$$

Where $e$ is the epoch number.

As discussed in Section 3, a lower temperature encourages the instance-specific representation learning, while a higher value pushes towards the class-specific features. For OSR, if the test representation from a novel class becomes too class-specific, the model easily finds its similarity to one of the known classes. On the other hand, if the feature is too instance-specific, the model is under-confident in assigning any sample to a known class. Therefore to avoid these pitfalls, a model needs to capture a delicate combination of both the desirable properties– good class-specific representations while having room for instance-level discriminating power within the class. Moreover, *Familiarity Hypothesis* by Dietterich & Guyer (2022) states that most existing OSR methods flag semantic novelty from the absence of learned class-specific features and it recommends to extract features for *interesting content* beyond the class-specific features for detecting novelty. If the model finds similarity between a novel sample and a known class because of the class-specific features, the instance-specific features should maintain the separation between them. A constant temperature throughout the training restricts the opportunity to traverse through this trade-off and fails to achieve the benefits of both extremes, limiting the quality of final representations. Utilizing the effects observed in Section 3 and to facilitate the traversal through the spectrum, we propose to gradually switch between the two objectives using a generalized cosine schedule, which we describe next.

## 4.3 PROPOSED TEMPERATURE SCHEDULES

Instead of a constant value, we propose to schedule the temperature (replacing $\tau$ by $\mathcal{T}(e)$) from a range $[\tau^-, \tau^+]$ as the training progresses using a generalized cosine schedule, which is defined as

$$\mathcal{T}_{\text{GCosSch}}(e; \tau^+, \tau^-, P, k) = \begin{cases} \tau^- + \frac{1}{2}(\tau^+ - \tau^-)(1 + \cos(\frac{2\pi e}{P} - k\pi)), & \text{if } e \leq E - \frac{kP}{2} \\ \tau^+, & \text{elsewhere} \end{cases} \qquad (5)$$

where value $k\pi$ represents the delay of cosine wave with respect to the starting epoch, $P$ is the period of the wave (Figure 2) and $k$ can be from $[0, 1]$. $k = 0$ reduces Eq. (5) to the regular cosine schedule (CosSch), proposed by Kukleva et al. (2023) for the self-supervised tasks on the long-tailed datasets. With CosSch, the model starts training with a higher temperature $\tau^+$ and goes to a lower temperature $\tau^-$.

$$\mathcal{T}_{\text{CosSch}}(e; \tau^+, \tau^-, P) = \mathcal{T}_{\text{GCosSch}}(e; \tau^+, \tau^-, P, k = 0) \qquad (6)$$

**Proposed Negative Cosine Schedule.** We find that rather than using Eq. (6), it is beneficial for the task switching if we start with a lower temperature $\tau^-$ and move towards a higher value $\tau^+$.

Starting with a lower temperature, the model provides priority to fewer neighbors, learning the coarse structure of representation space, resulting in a sharper decision boundary. The open set samples remain distributed and distant from any cluster of known classes (due to heavy penalization of slight dissimilarity for the lower $\tau^-$). As temperature increases, the model prioritizes more neighbors and gradually pulls the positive samples to its own cluster, refining on the coarse representation space. This makes within-class representations more compact and the decision boundary smoother while the core separation learned earlier is maintained. The open set samples are not pulled as tightly because their features are unknown to the model, maintaining the separation. This leads to a richer and potentially more generalized representation space for both open set and closed set performance.

The second half cycle (decreasing from a higher to a lower temperature) facilitates the exploration by refining the model again for the instance-specific features and a smooth transition for the restart of next periodic cycle. The periodic restart can help the model to refine its first solution and find a better one nearby to generalize more effectively, when tackling the challenging feature spaces. The model settles down better if the final few epochs maintain a higher temperature rather than follow the wave (epochs 500-600 in Figure 2). Figure 1 illustrates the concept with UMAP projection of representation spaces for different TSs.

For $k = 1$ in Eq. (5), the temperature starts with a lower value and goes to a higher one, looking like a negative cosine wave, hence the name negative cosine schedule (NegCosSch).

$$\mathcal{T}_{\text{NegCosSch}}(e; \tau^+, \tau^-, P) = \mathcal{T}_{\text{GCosSch}}(e; \tau^+, \tau^-, P, k = 1) \tag{7}$$

Our experiments demonstrate that NegCosSch surpasses both CosSch and GCosSch (with other values of $k \neq 1$) in OSR performance. Although initially aimed for OSR, NegCosSch also improves closed set classification, while being applicable to any model architecture and loss function, such as CE, SupCon and ARPL (Chen et al., 2021) and incurring no additional computational burden, as it only includes an epoch-dependent temperature in a loss function.

**Choice of $P$ and Other Proposed Monotonic Schedules.** We find that a single *monotonic* increase with the first half cycle of NegCosSch (termed as Monotonic-NegCosSch or M-NegCosSch), where $P = 2E$, is a sufficiently competitive TS, removing the need to tune for $P$.

$$\mathcal{T}_{\text{M-NegCosSch}}(e; \tau^+, \tau^-) = \tau^- + 0.5(\tau^+ - \tau^-)(1 - \cos(e\pi/E)); \forall e \tag{8}$$

Even a linear or an exponential temperature increase performs better than the baseline constant temperature. The exact formulations of these appear in Appendix A. Otherwise, $P$ in Eq. (7) can be chosen by dividing $E$ by the number of cycles. We denote periodic NegCosSch as P-NegCosSch. From ablation studies, we observe that varying $P$ or the number of cycles does not impact the performance significantly. $P$ needs to be within a functional range that allows for sufficient exploration - for example, $P = 200$ performs consistently well across the benchmarks.

**Choice of $(\tau^+, \tau^-)$.** Based on our ablation studies across the datasets (Appendix E.5), we derive that for any good value of constant temperature $\tau$ (which can be chosen from hyperparameter tuning), using NegCosSch is more effective by setting $\tau^+ = \tau + \Delta, \tau^- = \tau - \Delta$ (or alternatively, $\tau^+ = \tau + \Delta, \tau^- = \tau$) in the SupCon loss with the increment $\Delta \approx 0.1$ or $0.2$. The heuristic is also structurally informed: the placement of $\tau$ at the center ensures that the temperature is being varied around a good operating point. A reasonable value of $\Delta$ is crucial because a large $\Delta$ may collapse the semantic structure in the representation space: the excessively low $\tau^-$ may disrupt the initial formation of semantic structure, while the higher $\tau^+$ may remove the necessary instance-level discrimination. For example, hyperparameter tuning on TinyImageNet provides us a high OSR performance for $\mathcal{T}_{\text{Const}}$ with $\tau = 0.2$. Leveraging this, we derive that $\mathcal{T}_{\text{NegCosSch}}(\tau^+ = 0.4, \tau^- = 0.1)$ or $\mathcal{T}_{\text{NegCosSch}}(\tau^+ = 0.3, \tau^- = 0.2)$ are better choices than $\mathcal{T}_{\text{Const}}(\tau = 0.2)$, $\mathcal{T}_{\text{CosSch}}(\tau^+ = 0.4, \tau^- = 0.1)$ and $\mathcal{T}_{\text{CosSch}}(\tau^+ = 0.3, \tau^- = 0.2)$. For the CE loss, we find $\tau^+ = 2\tau, \tau^- = \tau/2$ as a good choice because here, the temperature scales the logits instead of similarities. The derived relations allow us to bypass the necessity of explicitly tuning for both $\tau^-$ and $\tau^+$.

**Inference.** For CE loss, we use the model as is for inference. However for SupCon loss, we remove the projection layer and a linear classifier is trained for evaluation. We use the maximum logit based scoring rule for OSR score.

## 5 RESULTS AND DISCUSSION

In this Section, we describe the benchmarks used for evaluating our method, the experiment settings followed by results and discussion.

**Benchmarks.** Here, we present the performance with different TSs on the TinyImageNet and the SSBs. The SSBs are defined on three fine-grained datasets: the Caltech-UCSD-Birds (CUB) (Wah et al. (2011)), FGVC-Aircraft (Krause et al. (2013)) and Stanford Cars (SCars) (Maji et al. (2013)). For SSBs, the open set classes are divided into 'Easy' and 'Hard' splits by computing the semantic similarity (based on the labeled visual attributes) of each pair of classes, the details of which can be found in Vaze et al. (2022). The different difficulty levels along with more training classes make these datasets harder OSR benchmarks than the other ones. Most of the OSR research does not report results on the SSBs. Moreover, we report the performance of our NegCosSch on the CIFAR benchmarks– CIFAR10, CIFAR+10 and CIFAR+50– from literature with their details in the Appendix.

**Training Details.** We mostly follow the experiment settings and design choices from Vaze et al. (2022). For TinyImageNet, we use a VGG32-like model and for SSBs, we use a ResNet50 model pretrained on the places365 dataset[2]. We run each experiment with 5 random seeds and report the average results. We also include results on a vision transformer model in Appendix E.4.

We perform ablations on $P$ from $\{100, 200, 1200\}$ and temperatures from $\mathbb{T}_{\text{SupCon}} = \{0.025, 0.05, 0.1, 0.2, 0.3, 0.4, 0.5\}$ for the SupCon loss. For TinyImageNet, we tune both the $\mathcal{T}_{\text{Const}}$ baseline ($\tau = 0.2$) and our schedules (($\tau^+, \tau^-) = (0.4, 0.1)$) on a validation set. For SSBs, we only tune $\tau = 0.2$ to optimize the constant baseline and apply the derived relationship, $(\tau^+, \tau^-) = (0.3, 0.1)$, as detailed in Section 4.3, which achieves strong improvements without extensive tuning. For CE loss, with the most utilized base temperature being 1.0, we set $(\tau^+, \tau^-)$ at $(0.5, 2.0)$. We set $P = 200$ for all periodic TSs and $P = 2E = 1200$ for M-CosSch and M-NegCosSch for consistent comparison. The training details appear in the Appendix.

**Metrics.** We report the closed set performance as a $C$-class classification using accuracy (%), the open set performance as known-unknown detection using AUROC (%), and the area under the open set classification rate curve (OSCR %). The OSCR curve measures the trade-off between Correct Classification Rate (CCR) for known samples and False Positive Rate (FPR) for unknowns (Dhamija et al. (2018)). We also implement the OpenAUC metric by Wang et al. (2022) and find that its scores are very similar to OSCR. We report OpenAUC in Appendix E.3. Now, we discuss the results.

### 5.1 ABLATION STUDY ON TEMPERATURE SCHEDULES

Here, we compare the closed set and open set performance among different TSs, such as a random schedule, a linear decrease and an increase, exponential and logarithmic monotonic increases, periodic and monotonic CosSch and NegCosSch on CE loss. We present the results in Table 1. For most cases, our proposed TSs, such as P-NegCosSch, M-NegCosSch, linear and exponential increases perform better than the constant baseline, CosSchs and other listed schedules in terms of all metrics. While the simpler schedules, such as linear and exponential increases demonstrate competitive results, in terms of a single best result across a column, P-NegCosSch wins at a maximum number (8 out of 18) of cases and M-NegCosSch wins at 4 cases. Their collective gains prove that the proposed temperature modulation scheme is fundamentally better than the constant baseline and other schedules. The standard deviations of these results across the trials are presented in Table 4 in the Appendix.

### 5.2 PERFORMANCE OF OUR TEMPERATURE SCHEDULES ACROSS VARIOUS LOSS FUNCTIONS

Here, we report the performance on different OSR loss functions and the recent BackMix method (Wang et al., 2025) by including and without our NegCosSch in Table 2. We implement the ARPL loss, the CE baseline by Vaze et al. (2022) and the widely implemented SupCon loss in the recent OSR literature (Xu et al., 2023; Zhou et al., 2024a). As label smoothing (LS) has shown significant performance improvements in several cases as demonstrated by Vaze et al. (2022), we experiment both with and without uniform LS, considering them as separate baselines for the CE and SupCon

---

[2]In spite of our efforts, we could not find the same pretrained model mentioned in Vaze et al. (2022) online. Therefore, we use the pretrained model from Zhou et al. (2017) trained on places365, which is completely unrelated to the SSBs.

Table 1: Comparison of different TSs on CE loss. For SSBs, the OSR results are shown on 'Easy/ Hard' splits. We bold the top three results for each metric and underline the best case.

| | Accuracy (%) | AUROC (%) | OSCR (%) | Accuracy (%) | AUROC (%) | OSCR (%) |
|---|---|---|---|---|---|---|
| Schedule | **CUB** | | | **Aircraft** | | |
| Constant (Baseline) | 84.43 | 83.55 / 74.98 | 70.49 / 63.34 | 90.88 | 90.35 / 81.48 | 82.05 / 74.25 |
| Linear decrease | 81.64 | 79.86 / 71.75 | 65.15 / 58.59 | 90.58 | 89.53 / 79.7 | 81.08 / 72.41 |
| Random | 85.06 | 85.02 / 75.54 | 72.28 / 64.32 | 91 | 90.76 / 82.37 | 82.55 / 75.17 |
| P-CosSch | 84.63 | 84.5 / 74.24 | 71.51 / 62.93 | 90.8 | 90.04 / 81.81 | 81.76 / 74.51 |
| M-CosSch | 81.77 | 79.55 / 71.4 | 64.96 / 58.35 | 90.62 | 88.63 / 80.92 | 80.35 / 73.57 |
| Logarithmic increase | 85.15 | 84.91 / 76.07 | 72.25 / 64.82 | **91.19** | 90.86 / 82.58 | 82.77 / 75.47 |
| Exponential increase (ours) | **86.12** | **86.65** / **78.05** | **74.64** / 67.35 | 90.88 | 90.92 / 82.93 | 82.54 / 75.54 |
| Linear increase (ours) | **86.22** | 86.54 / **78.01** | 74.58 / **67.32** | 90.97 | **91.11** / 83.25 | **82.87 / 76** |
| P-NegCosSch (ours) | 86.3 | **86.85** / 77.6 | **74.89** / 67.01 | **91.33** | **91.41** / **83.15** | **83.43 / 76.14** |
| M-NegCosSch (ours) | **86.12** | **86.79** / 78.08 | **74.7** / 67.3 | **91.15** | **91.15** / **83.23** | **82.99 / 76** |
| | **SCars** | | | **TinyImageNet** | | |
| Constant (Baseline) | 96.76 | 94.03 / 84.82 | 91.04 / 82.19 | 84.55 | 82.85 | 74.74 |
| Linear decrease | 96.25 | 92.51 / 83.47 | 89.14 / 80.51 | 84.12 | 82.64 | 74.28 |
| Random | 97.06 | 94.31 / 85.27 | 91.58 / 82.85 | 83.51 | 78.37 | 69.88 |
| P-CosSch | 96.63 | 93.85 / 84.88 | 90.75 / 82.14 | 84.41 | **83.12** | 74.79 |
| M-CosSch | 96.27 | 92.21 / 82.7 | 88.84 / 79.73 | 84.19 | 82.76 | 74.36 |
| Logarithmic increase | 97.07 | 94.92 / 85.42 | 92.18 / 83.03 | 84.74 | 82.96 | 74.91 |
| Exponential increase (ours) | **97.27** | **95.08** / 86.03 | **92.5** / 83.75 | 84.98 | 83.05 | 75.15 |
| Linear increase (ours) | 97.19 | 95.19 / **86.18** | **92.55** / **83.86** | **84.9** | 83.16 | 75.19 |
| P-NegCosSch (ours) | 97.3 | 95.03 / **86.05** | 92.49 / **83.81** | **84.85** | 83.02 | **75** |
| M-NegCosSch (ours) | **97.22** | **95.18** / 86.26 | **92.57 / 83.95** | 84.24 | 82.79 | 74.41 |

losses. We aim to investigate whether our proposed TSs offer orthogonal benefits irrespective of LS. Here, we do not optimize performance for the LS coefficient and temperatures but use a fixed set of hyperparameters for consistency.

We observe that including the proposed schedules (P-NegCosSch or M-NegCosSch) in any OSR loss function improves performances both for the closed set and open set problems over the corresponding constant temperature baseline for all cases except for two, such as for Aircraft on SupCon loss including LS and P-NegCosSch, and for TinyImageNet on CE loss including LS and M-NegCosSch. Our NegCosSch provides performance boost for up to 1.87% of accuracy, up to 3.3%/3.1% of open set AUROC in the 'Easy'/'Hard' splits and up to 4.4%/3.96% of OSCR. This amount of performance boost comes without any additional computational cost. Between our two schedules, the M-NegCosSch performs better in most cases than the periodic one, demonstrating that the primary benefit for most cases is derived from monotonic negative cosine increase. However, P-NegCosSch outperforms in several cases – such as for CUB and Aircraft benchmarks on CE and BackMix, and for the open set metrics on TinyImageNet with ARPL and BackMix, which confirms that refining with periodicity can help achieving an improved representation space depending on the data characteristics.

Moreover, our method can be used together with LS to further boost the performance in a few cases (for CUB- M-NegCosSch and TinyImageNet with CE loss). Even for cases where LS does not improve the constant baseline performance (for the Stanford cars -'Hard' split and Aircraft with CE loss), our NegCosSch outperforms the corresponding baseline. In these few cases, LS may cause drops in OSR metrics because it leads to max-logit suppression as shown in Xia et al. (2025), which degrades the ranking of scores by assigning relatively lower max-logit on the correct known samples and higher scores for unknowns compared to the without-LS case. In spite of this, our TSs boost OSR performance even where LS alone failed. Overall for SSBs, the CE loss performs better than SupCon loss. We believe that our scheme, in principle, can improve other OSR methods, such as the method by Jia et al. (2024).

## 5.3 Proposed schedule is more beneficial with more training classes

To show the strength of our NegCosSch with increased number of training classes, we train models on CE loss with $\{20\%, 40\%, 60\%, 80\%, 100\%\}$ of the randomly chosen training classes for the SSBs without changing the open set. In Figure 3, we plot the improvement of a metric $m$ over the corresponding baseline, defined as: improvement $= m[\mathcal{T}_{\text{NegCosSch}}(\tau^+, \tau^-)] - \max_{\tau \in \mathbb{T}_{CE}}\{m[\mathcal{T}_{\text{Const}}(\tau)]\}$. Here, $\mathbb{T}_{CE} = \{0.5, 1.0, 2.0\}$. We observe an overall upward improvement trend with more training classes in most cases for both the 'Easy' and 'Hard' OSR splits. The negative values in Figure 3c is

Table 2: Performance on different OSR loss functions, with and without the proposed schedules. Open set results are shown on 'Easy / Hard' splits. We highlight the cases where our TS produces better results than the baseline and underline the best result for each case.

| Loss | Schedule | Accuracy (%) | AUROC (%) | OSCR (%) | Accuracy (%) | AUROC (%) | OSCR (%) |
|---|---|---|---|---|---|---|---|
| | | **CUB** | | | **Aircraft** | | |
| CE (w/o LS) | Constant | 84.43 | 83.55 / 74.98 | 70.49 / 63.34 | 90.88 | 90.35 / 81.48 | 82.05 / 74.25 |
| | M-NegCosSch(ours) | 86.12 | 86.79 / 78.08 | 74.7 / 67.3 | 91.15 | 91.15 / 83.23 | 82.99 / 76 |
| | P-NegCosSch(ours) | 86.3 | 86.85 / 77.6 | 74.89 / 67.01 | 91.33 | 91.41 / 83.15 | 83.43 / 76.14 |
| CE + LS (Vaze et al. (2022)) | Constant | 85.53 | 85.15 / 77.44 | 72.77 / 66.26 | 90.73 | 86.85 / 79.72 | 78.84 / 72.55 |
| | M-NegCosSch(ours) | 86.21 | 87.66 / 79.06 | 75.53 / 68.23 | 91.34 | 88.25 / 81.19 | 80.62 / 74.36 |
| | P-NegCosSch(ours) | 86.12 | 86.43 / 78.03 | 74.36 / 67.22 | 91.1 | 87.25 / 80.03 | 79.55 / 73.17 |
| SupCon (w/o LS) | Constant | 83.43 | 86.94 / 73.95 | 72.42 / 61.66 | 90.71 | 88.78 / 81.79 | 80.51 / 74.37 |
| | M-NegCosSch(ours) | 85.3 | 88.14 / 75.81 | 75.09 / 64.72 | 91.43 | 90.45 / 82.49 | 82.57 / 75.51 |
| | P-NegCosSch(ours) | 84.12 | 87.5 / 74.95 | 73.54 / 63.13 | 90.61 | 89.27 / 81.97 | 80.96 / 74.55 |
| SupCon + LS | Constant | 83.72 | 86.43 / 73.69 | 72.3 / 61.74 | 90.05 | 88.97 / 81.81 | 80.11 / 73.85 |
| | M-NegCosSch(ours) | 85.28 | 88.05 / 75.78 | 74.97 / 64.63 | 90.55 | 89.47 / 81.85 | 80.95 / 74.28 |
| | P-NegCosSch(ours) | 84.38 | 87.26 / 75.16 | 73.5 / 63.39 | 90.43 | 88.77 / 81.78 | 80.2 / 74.08 |
| ARPL (Chen et al. (2021)) | Constant | 85.8 | 86.93 / 79.7 | 78.64 / 73.36 | 90.88 | 90.75 / 81.77 | 85.98 / 78.26 |
| | M-NegCosSch(ours) | 86.47 | 87.6 / 80.53 | 79.65 / 74.41 | 91.26 | 91.55 / 82.01 | 86.65 / 78.39 |
| | P-NegCosSch(ours) | 86.51 | 87.57 / 80.12 | 79.61 / 74.05 | 90.98 | 91.52 / 82.02 | 86.55 / 78.38 |
| BackMix (Wang et al. (2025)) | Constant | 82.12 | 82.39 / 72.99 | 67.94 / 60.32 | 90.53 | 92.41 / 82.47 | 83.75 / 75.09 |
| | M-NegCosSch (ours) | 82.84 | 83.97 / 74.66 | 69.71 / 62.1 | 91.37 | 92.2 / 84.43 | 84.19 / 77.36 |
| | P-NegCoSch (ours) | 83.98 | 84.54 / 74.13 | 71.23 / 62.66 | 91.49 | 92.3 / 83.68 | 84.35 / 76.72 |
| | | **SCars** | | | **TinyImageNet** | | |
| CE (w/o LS) | Constant | 96.76 | 94.03 / 84.82 | 91.04 / 82.19 | 81.95 | 78.6 | 69.22 |
| | M-NegCosSch(ours) | 97.22 | 95.18 / 86.26 | 92.57 / 83.95 | 81.98 | 79.21 | 69.84 |
| | P-NegCosSch(ours) | 97.3 | 95.03 / 86.05 | 92.49 / 83.81 | 82.23 | 79.05 | 69.91 |
| CE + LS | Constant | 97.05 | 94.67 / 84.35 | 91.95 / 82.02 | 84.55 | 82.85 | 74.74 |
| | M-NegCosSch(ours) | 97.23 | 95 / 85.06 | 92.42 / 82.83 | 84.24 | 82.79 | 74.41 |
| | P-NegCosSch(ours) | 97.23 | 94.82 / 84.54 | 92.24 / 82.31 | 84.85 | 83.02 | 75 |
| SupCon (w/o LS) | Constant | 96.58 | 92.99 / 82.8 | 89.92 / 80.12 | 85.37 | 82.87 | 70.61 |
| | M-NegCosSch(ours) | 96.79 | 93.57 / 82.76 | 90.66 / 80.26 | 85.4 | 83.21 | 70.98 |
| | P-NegCosSch(ours) | 96.68 | 93.32 / 83.16 | 90.31 / 80.53 | 85.18 | 83.09 | 70.71 |
| SupCon + LS | Constant | 96.6 | 93.03 / 83.32 | 89.95 / 80.63 | 85.18 | 82.65 | 70.31 |
| | M-NegCosSch(ours) | 96.84 | 93.58 / 83.15 | 90.69 / 80.62 | 85.57 | 83.11 | 71.04 |
| | P-NegCosSch(ours) | 96.69 | 93.45 / 83.29 | 90.43 / 80.66 | 85.23 | 83.05 | 70.72 |
| ARPL | Constant | 97.37 | 95.22 / 85.89 | 93.46 / 84.7 | 85.02 | 83 | 74.89 |
| | M-NegCosSch(ours) | 97.29 | 95.27 / 86.03 | 93.48 / 84.82 | 84.83 | 83.07 | 75.03 |
| | P-NegCosSch(ours) | 97.21 | 95.25 / 85.71 | 93.47 / 84.52 | 85 | 83.12 | 75.07 |
| BackMix | Constant | 96.81 | 93.23 / 84.39 | 90.33 / 81.82 | 82.32 | 81.23 | 67.1 |
| | M-NegCoSch (ours) | 97.52 | 94.86 / 86.48 | 92.56 / 84.48 | 82.6 | 81.62 | 67.37 |
| | P-NegCoSch (ours) | 97.37 | 94.76 / 86.04 | 92.31 / 83.89 | 82.5 | 81.72 | 67.42 |

(a) CUB     (b) SCars     (c) FGVC-Aircraft

Figure 3: Effect on performance improvement for our proposed schedule over the baselines with varying number of training classes. Increasing the number of training classes tends to yield greater improvements in OSCR across all datasets, along with significant improvements in AUROC and accuracy, with the effect being most pronounced for CUB and least for FGVC-Aircraft. Error bars represent the standard deviations across trials with random training classes.

due to the fact that we measure improvement over the maximum score of three baselines. With more training classes, the task becomes harder for the base model, which is observed by the performance decline. Nonetheless, our schedule gains higher improvement with more training classes.

We observe that the benefits of our schedules may reduce when the number of training classes is relatively small, which also occur in the CIFAR benchmarks (discussed in Appendix E.8). The baseline performance on the CIFAR benchmarks are already substantial, whereas the tougher SSBs require significant improvement, where the benefits of our proposed schedules are realized most. We leave extending our temperature modulation to these smaller benchmarks as a future work.

## 6 RELATED WORKS

**Open Set Recognition.** Since the introduction of OSR problem, it has received a significant interest in the research community. Most of the research attempts can be summarized into several common categories, some of which are discussed in Section 1. Besides the use of generative models, input or mani-fold mix-up, other works add auxiliary samples with different strong augmentations for training models (Wang et al., 2025; Jiang et al., 2023; Jia et al., 2024; Xu & Keuper). Another huge group of research depends on training an additional model with a secondary objective function (Oza & Patel (2019); Sun et al. (2020); Perera et al. (2020); Yoshihashi et al. (2019); Zhang et al. (2020); Jia et al. (2024); Zhou et al. (2024a)). However, training a generative model or a secondary VAE model is a cumbersome task on the real-life larger benchmarks as it requires significant computation overhead and therefore, is not practical. Moreover mix-up based methods, such as manifold mix-up can increase the amount of computation during backpropagation as the interpolation of samples occurs in a hidden layer, changing the standard forward-backward pass procedure(Verma et al., 2019).

Another set of methods either construct a different loss function (Chen et al. (2021; 2020a); Wang et al. (2022)) or add regularization to bound the open set risks (Zhou et al. (2021); Lu et al. (2022); Yang et al. (2024a)). For example, the method by Zhou et al. (2021) learns additional place-holders for the novel classes. Methods by Chen et al. (2021; 2020a) learn the reciprocal points of known classes representing the 'otherness' corresponding to each class. These methods try to create additional empty regions in the representation space hoping that open set representations lie in those regions. The new baseline by Vaze et al. (2022) with well-trained closed set classifiers has triggered the OSR research for better representation learning schemes. For example, methods by Xu et al. (2023); Xu (2024); Li et al. (2024); Bahavan et al. (2025); Li et al. (2025) train models using the contrastive loss with regularization and heavy augmentations (Wang et al. (2025); Jiang et al. (2023); Jia et al. (2024)). The method by Wang et al. (2024) trains multiple experts for extracting diverse representations, and Yang et al. (2024b) proposes an open set self-learning framework, which adapts the model according to the test data assuming that it is available. Furthermore, several prior works have focused on developing fine-grained OSR methods (Lang et al., 2024; Bao et al., 2023; Sun et al., 2023).

**Temperature Scaling.** Temperature scaling in the CE loss plays a crucial role in knowledge distillation (Hinton et al. (2015)), model calibration (Guo et al. (2017)) and so on. Since the contrastive loss has become popular for many tasks (Chen et al. (2020b); Khosla et al. (2020)), studies have aimed to understand its behavior. Recently, methods by Jeong et al. (2024); Qiu et al. (2024) utilize temperature cool-down in language models.

## 7 CONCLUSION

We develop novel temperature schedules, which can be folded into any existing OSR loss function, such as cross-entropy, contrastive or ARPL without any computational overhead. We find that starting with a lower temperature and moving towards a higher temperature results in making tighter representation clusters for the closed set classes, while the representations of the open set examples remain more distant. This process is more effective than using a fixed temperature or the opposite schedule. Our proposed schedules demonstrate strong performance improvements on the regular and the tougher semantic shift benchmarks for both closed set and open set problems for some of the well-known OSR loss functions, even on top of label smoothing. The benefit of our scheme can be better realized with a larger number of training classes.

### ETHICS STATEMENT

Open set recognition is crucial for enhancing safety and reliability in machine learning systems operating in changing environments by detecting novel patterns. For instance, all categories of interest

may not be represented in the training set due to their rarity or new categories may emerge due to dynamic nature. The capability of a deep model of knowing what it doesn't know enhances trust across various critical applications.

The solution for the OSR problem is yet to be improved, especially for larger datasets. Their performance depends on the semantic closeness between the known and unknown classes. Hence, the methods cannot be solely relied upon in deployment. For example, an over-sensitive OSR system can lead to a high false alarm rate.

ACKNOWLEDGMENTS

The authors would like to acknowledge the partial support provided for this work by the University of Maryland Institute for Health Computing (UM-IHC) and the University of Maryland Institute for Advanced Computer Studies (UMIACS).

REPRODUCIBILITY STATEMENT

Our implementation adheres rigorously to the benchmarks, i.e., the set of known- unknown splits defined in the standard OSR literature, such as in Vaze et al. (2022). For consistent comparison, we use the same experiment settings and design choices in model architecture and hyperparameters, the details of which can be found in Appendix C. Detailed information on the hardware and software utilized is provided in Appendix G. Project codes are available at: https://github.com/amit31416/NegCosSch/.

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

# Supplementary Materials

## A  MONOTONIC TEMPERATURE SCHEDULES

Different monotonic TSs– such as the linear, exponential, and logarithmic increases, as well as the linear decrease, over the range $[\tau^-, \tau^+]$– are listed below. Similar to our negative cosine TS, the linear and exponential schedules also start with a lower value of $\tau$ and gradually switches the task with a higher value. For a random TS, we pick a random temperature from $[\tau^-, \tau^+]$ at each epoch.

$$\mathcal{T}_{\text{linear}}(e; \tau^+, \tau^-) = \tau^- + \frac{e}{E}(\tau^+ - \tau^-) \tag{9}$$

$$\mathcal{T}_{\text{exponential}}(e; \tau^+, \tau^-) = \tau^- \times (\frac{\tau^+}{\tau^-})^{e/E} \tag{10}$$

$$\mathcal{T}_{\text{logarithmic}}(e; \tau^+, \tau^-) = \tau^- + (\tau^+ - \tau^-) \times \frac{\log(e)}{\log(E)} \tag{11}$$

$$\mathcal{T}_{\text{linear-decrease}}(e; \tau^+, \tau^-) = \tau^+ - \frac{e}{E}(\tau^+ - \tau^-) \tag{12}$$

## B  GRADIENTS OF LOSS FUNCTIONS

### B.1  GRADIENT OF SUPCON LOSS

For any sample $i \in I$, the gradient of $L_{\text{SupCon}}$ in Eq. (2) with respect to a negative logit $l_j$

$$\frac{\partial L_{SupCon}}{\partial l_j} = \frac{\partial}{\partial \text{sim}(l_i, l_j)} \left[ \frac{1}{|P(i)|} \sum_{p \in P(i)} \left( -\frac{1}{\tau}\text{sim}(l_i, l_p) + \log \sum_{a \in I \setminus \{i\}} \exp(\text{sim}(l_i, l_a)/\tau) \right) \right]$$
$$\times \frac{\partial \text{sim}(l_i, l_j)}{\partial l_j}$$
$$= \frac{1}{|P(i)|} \sum_{p \in P(i)} \frac{\frac{1}{\tau}\exp(\text{sim}(l_i, l_j)/\tau)}{\sum_{a \in I \setminus \{i\}} \exp(\text{sim}(l_i, l_a)/\tau)} \times \frac{\partial \text{sim}(l_i, l_j)}{\partial l_j}$$
$$= \frac{1}{\tau}[\text{softmax}_{a \in I \setminus \{i\}}(\text{sim}(l_i, l_a)/\tau)]_j \times \frac{\partial \text{sim}(l_i, l_j)}{\partial l_j}$$

We already discussed this in Section 3. Similarly, the gradient of $L_{\text{SupCon}}$ with respect to a positive logit $l_j$

$$\frac{\partial L_{SupCon}}{\partial l_j} = \frac{1}{\tau|P(i)|} \left( |P(i)|[\text{softmax}_{a \in I \setminus \{i\}}(\text{sim}(l_i, l_a)/\tau)]_j - 1 \right) \times \frac{\partial \text{sim}(l_i, l_j)}{\partial l_j}$$

To push the gradient towards 0, value of the softmax function should approach towards $\frac{1}{|P(i)|}$. For large value of $\tau$, this is possible when all negatives are far away than the positives, to have minimum effects in the denominator of softmax function. Moreover, the differences of scaled similarities between anchor and the positives diminish, inducing the class-specific features.

### B.2  GRADIENT OF CE LOSS

The gradient of $L_{CE}$ with respect to the logit of output node $j$ corresponding to the true label of sample $k$,

$$\frac{\partial L_{CE}}{\partial l_{k,j}} = -\frac{\partial}{\partial l_{k,j}} \log \frac{\exp(l_{k,j}/\tau)}{\sum_i \exp(l_{k,i}/\tau)} = \frac{1}{\tau}[\text{softmax}_k(l_k/\tau)_j - 1]$$

For small $\tau$, the differences of scaled logits will be amplified, and the softmax will approach towards an indicator function. The same softmax value in this gradient term computes the probability for the output node of the true class which will approach towards 1.0. Therefore, the resulting probability distribution is sharper. For large $\tau$, the differences of scaled logits will diminish, and the softmax will approach towards $1/C$, making the resulting probability distribution smoother.

## C    TRAINING DETAILS

**Benchmarks.** There are 5 known-unknown random splits defined in the regular OSR benchmarks. We consider 4 regular benchmarks, such as the CIFAR10, CIFAR+10. CIFAR+50 and the TinyImageNet benchmarks. The CIFAR10 benchmark has 6 closed set classes and 4 open set classes, whereas CIFAR+10 and CIFAR+50 have 4 closed set classes from the CIFAR10 dataset and 10 and 50 open set classes from CIFAR100 dataset respectively. TinyImageNet has 20 training classes and 180 closed set classes. The SSBs are defined with 50% of the classes in training and rest 50% classes are divided into 'Easy', 'Medium' and 'Hard' splits. Similar to Vaze et al. (2022), we combine the 'Medium' and 'Hard' splits to report as the 'Hard' split in our paper. For each split, we run each experiment with 5 different random seeds and we report average results.

**Model Architecture.** We follow experimental settings similar to the existing literature, such as in Vaze et al. (2022). For the regular benchmarks, we train VGG32-like models from scratch and for the SSBs, we train ResNet50 models pretrained on the places365 dataset for a supervised task. The feature dimensions are 128 and 2048 for the regular benchmarks and the SSBs respectively. The linear projection layer for SupCon training has the same number of input and output nodes as the feature dimension.

**Hyperparameters.** We train all models for 600 epochs with the SGD optimizer with a momentum of 0.9 and a weight decay of $10^{-4}$. We use a cosine learning rate scheduler with warm-ups and 2 restarts at the 200-th and 400-th epoch. The initial learning rate is set to 0.1 for the CIFAR benchmarks and 0.001 for the SSBs. For TinyImageNet, it is set to 0.01 for the CE loss and we tune it to 0.05 for the SupCon loss. Rand-Augment is used for data augmentations in all cases. While tuning, we select the hyperparameters by maximizing the closed-set performance on a validation set. The validation set is constructed by holding out 20% random training data from one known-unknown split. Batch size is set to 128 for the regular benchmarks. For the SSBs, the batch size is set to 12 as only this amount can be accommodated in our single GPU for each experiment in the SupCon training. The images are resized to $32 \times 32$, $64 \times 64$, and $448 \times 448$ respectively for the CIFAR, the TinyImageNet benchmarks and the SSBs.

**Label Smoothing.** For TinyImageNet and the SSB datasets, we report results both including and without uniform LS as LS has shown improvements for these datasets. For the CE loss, we choose the LS coefficients from Vaze et al. (2022). For LS in SupCon loss, we implement the following function instead of (2):

$$L_{\text{SC,LS}} = -\frac{1}{|I|} \sum_{i \in I} \sum_{j \in I \setminus \{i\}} \frac{1}{N_i(\alpha)} \left[ (1-\alpha)\mathbf{1}_{y_i=y_j} + \frac{\alpha}{C-1}\mathbf{1}_{y_i \neq y_j} \right] \left[ \log \frac{\exp(\text{sim}(z_i, z_j)/\tau)}{\sum_{a \in I \setminus \{i\}} \exp(\text{sim}(z_i, z_a)/\tau)} \right]$$
(13)

with $N_i(\alpha) = \sum_{k \in I \setminus \{i\}}[(1-\alpha)\mathbf{1}_{y_i=y_k} + \frac{\alpha}{C-1}\mathbf{1}_{y_i \neq y_k}]$, where $\alpha$ is the smoothing coefficient and $\mathbf{1}$ is the indicator function. For contrastive loss, we tune $\alpha$ from $\{0.1, 0.2, 0.3\}$; however, we use $\alpha = 0.2$ in Table 2 for consistent comparison.

**Details for the UMAPs in Figure 1.** We randomly choose 10 training classes from the defined closed set of the CUB benchmark and keep the open set as it is. We train models with constant temperatures of 0.5, 1.0, 2.0, $\mathcal{T}_{\text{CosSch}}(\tau^+ = 2.0, \tau^- = 0.5)$ and our $\mathcal{T}_{\text{NegCosSch}}(\tau^+ = 2.0, \tau^- = 0.5)$ with CE loss and without LS. To show the training progress in our method, we plot the features at the beginning, the middle and the end of the last scheduling period starting at epoch 400. We standardize the features by subtracting the mean and scaling them to unit variance before applying UMAP transformation. For clear visualization, we plot features of all the closed set samples and 10% random open set samples.

## D    DISCUSSIONS ON THE ADVERSARIAL RECIPROCAL POINT LEARNING

The adversarial reciprocal point learning (ARPL) method (Chen et al., 2021) defines a reciprocal point for each category $c$, denoted by $r_c$, which is regarded as the latent representation of the 'otherness' corresponding to each class. The reciprocal points $\{r_c\}_{c=1}^{C}$ are learnable parameters. Given a logit $l_i = f(x_i)$ and a reciprocal point $r_c$, their distance $d(l_i, r_c)$ is calculated by combining the Euclidean

distance and the dot product as the following:

$$d_e(l_i, r_c) = \frac{1}{D}||l_i - r_c||_2^2, \quad d_d(l_i, r_c) = l_i \cdot r_c$$

$$d(l_i, r_c) = d_e(l_i, r_c) - d_d(l_i, r_c)$$

$D$ is the number of feature dimension in $l$. The final classification probability is calculated as:

$$p(y_i = c|x_i, f, \{r_c\}_{m=1}^C) = \frac{\exp(d(l_i, r_c)/\tau)}{\sum_{m=1}^C \exp(d(l_i, r_m)/\tau)}$$

The total loss is calculated as

$$L_{\text{ARPL}} = -\log p(y_i = c|x_i, f, \{r_c\}_{c=1}^C) + \lambda \max(d_e(l_i, r_c) - R, 0) \tag{14}$$

We observe that the distance is also scaled with the temperature parameter ($\tau$) in this loss. $\lambda$ is to adjust the trade-off between the two loss terms and is set to 0.1 and, $R$ is the learnable margin parameter.

# E   ADDITIONAL RESULTS

Here, we discuss the performance variability of the proposed TSs, ablation studies on $(\tau^+, \tau^-), P$, and $k$, some results on a vision transformer model, and performance on the CIFAR benchmarks.

Table 3: Representation space geometry analysis. Metrics measure cluster quality (intra-class and inter-class scatter) and average distance from the unknown samples to the nearest prototype of known classes. NegCosSch achieves a competitive inter-class separability and the highest average distance from unknown samples to nearest prototype, demonstrating that our proposed schedule learns an improved representation space geometry for both the tasks.

| | CUB | | | SCars | | |
|---|---|---|---|---|---|---|
| Schedule | intra-class scatter ($\downarrow$) | inter-class margin ($\uparrow$) | unknowns' distance to nearest prototype ($\uparrow$) | intra-class scatter ($\downarrow$) | inter-class margin ($\uparrow$) | unknowns' distance to nearest prototype ($\uparrow$) |
| Const. ($\tau = 0.5$) | 0.1463 | 0.5503 | **0.4172** | 0.0919 | 0.6587 | **0.3869** |
| Const. ($\tau = 2.0$) | **0.1252** | **0.7805** | 0.3974 | **0.0609** | **0.9129** | 0.3797 |
| Const. ($\tau = 1.0$) | 0.1349 | 0.6586 | 0.41 | 0.0754 | 0.7923 | 0.3814 |
| P-CosSch | 0.1287 | 0.6832 | 0.4037 | 0.072 | 0.8127 | 0.3826 |
| P-NegCosSch | **0.1318** | **0.7741** | **0.423** | **0.068** | **0.8951** | **0.4002** |

## E.1   REPRESENTATION SPACE ANALYSIS WITH GEOMETRIC PROPERTIES

To strengthen our claim that the proposed schedules improve the overall representation learning, we conduct a quantitative diagnosis to analyze the learned space with the following geometric properties– intra and inter-class scatter, and the average distance from unknown samples to the nearest prototype of known classes, which are defined as

$$\text{intra-class scatter} = \frac{1}{|Z_k|} \sum_{c=1}^C \sum_{\{z_i \in Z_k|y_i=c\}} ||z_i - p_c||^2$$

$$\text{inter-class margin} = \frac{1}{\binom{N}{2}} \sum_{i=1}^C \sum_{j=1, i<j}^C ||p_i - p_j||$$

$$\text{distance of unknowns to nearest prototype} = \frac{1}{|Z_u|} \sum_{z_u \in Z_u} \min_{1 \leq c \leq C} ||z_u - p_c||$$

Where, $Z_k$ and $Z_u$ is the set of known and unknown representations respectively and $p_c$ is the prototype for class $c$. The metrics are reported in Table 3. We observe that a lower temperature achieves a higher intra-class scatter and a lower inter-class margin (as encouraged by the instance-specific learning), indicating poor separation. It also achieves a higher average distance from

Table 4: Performance standard deviation of different TSs on CE loss across various seeds.

| | Accuracy (%) | AUROC (%) | OSCR (%) | Accuracy (%) | AUROC (%) | OSCR (%) |
|---|---|---|---|---|---|---|
| **TS** | | **CUB** | | | **Aircraft** | |
| Const. (Baseline) | 0.2 | 0.26 / 0.26 | 0.41 / 0.37 | 0.39 | 0.77 / 0.52 | 0.42 / 0.47 |
| Linear decrease | 0.78 | 0.35 / 1.02 | 0.76 / 1.03 | 0.29 | 0.42 / 0.53 | 0.46 / 0.56 |
| Random | 0.64 | 0.23 / 0.44 | 0.67 / 0.74 | 0.21 | 0.27 / 0.75 | 0.25 / 0.75 |
| P-CosSch | 0.42 | 0.29 / 0.52 | 0.6 / 0.7 | 0.31 | 0.74 / 0.27 | 0.85 / 0.36 |
| M-CosSch | 0.28 | 0.26 / 0.66 | 0.06 / 0.63 | 0.32 | 0.82 / 1.05 | 0.85 / 1.14 |
| Logarithmic increase | 0.12 | 0.39 / 0.48 | 0.39 / 0.46 | 0.32 | 0.34 / 0.49 | 0.53 / 0.45 |
| Exponential increase (ours) | 0.2 | 0.28 / 0.18 | 0.29 / 0.15 | 0.27 | 0.26 / 0.54 | 0.12 / 0.54 |
| Linear increase (ours) | 0.31 | 0.21 / 0.19 | 0.25 / 0.15 | 0.29 | 0.12 / 0.56 | 0.23 / 0.73 |
| P-NegCosSch (ours) | 0.47 | 0.38 / 0.23 | 0.34 / 0.4 | 0.34 | 0.25 / 0.43 | 0.26 / 0.5 |
| M-NegCosSch (ours) | 0.27 | 0.44 / 0.28 | 0.6 / 0.45 | 0.31 | 0.15 / 0.47 | 0.26 / 0.3 |
| | | **SCars** | | | **TinyImageNet** | |
| Const. (Baseline) | 0.1 | 0.52 / 0.49 | 0.51 / 0.52 | 0.25 | 0.24 | 0.21 |
| Linear decrease | 0.29 | 0.76 / 0.77 | 0.9 / 0.89 | 0.18 | 0.25 | 0.21 |
| Random | 0.1 | 0.5 / 0.79 | 0.54 / 0.77 | 0.19 | 1.23 | 1.1 |
| P-CosSch | 0.08 | 0.56 / 0.98 | 0.55 / 0.96 | 0.16 | 0.34 | 0.3 |
| M-CosSch | 0.13 | 0.69 / 0.52 | 0.61 / 0.59 | 0.1 | 0.32 | 0.28 |
| Logarithmic increase | 0.15 | 0.4 / 0.56 | 0.41 / 0.67 | 0.2 | 0.28 | 0.33 |
| Exponential increase (ours) | 0.1 | 0.31 / 0.57 | 0.3 / 0.63 | 0.12 | 0.2 | 0.22 |
| Linear increase (ours) | 0.1 | 0.23 / 0.53 | 0.27 / 0.58 | 0.24 | 0.32 | 0.37 |
| P-NegCosSch (ours) | 0.1 | 0.29 / 0.62 | 0.21 / 0.59 | 0.29 | 0.37 | 0.36 |
| M-NegCosSch (ours) | 0.09 | 0.22 / 0.51 | 0.26 / 0.55 | 0.15 | 0.22 | 0.19 |

unknown samples to the nearest class prototype, which is a desirable property for better OSR. A higher temperature achieves the opposite, resulting from the class-specific learning, indicating a better closed set separation. A mid value of $\tau$ achieves a trade-off in terms of these geometric properties, while our proposed schedule (P-NegCosSch) successfully achieves a necessary combination from both the extremes – a lower intra-class scatter, a higher inter-class margin and a higher average distance from unknowns to prototypes. This structural superiority confirms that our schedule better utilizes the entire representation space.

## E.2 Performance Variability

Here, we present the standard deviations of performance metrics across trials with 5 different seeds. Tables 4 and 5 present the standard deviations for the performance results reported in Tables 1 and 2, respectively. The proposed TSs demonstrate either better or similar standard deviation compared to the baseline and the other TSs (presented in Table 4) and for all losses (presented in Table 5) considering the significant performance boost achieved by our proposed ones.

**Statistical Significance Test.** We perform one-sided non-parametric Wilcoxon rank tests to evaluate the statistical significance of improvements for CE and SupCon losses in Table 2. We test the hypothesis that our proposed schedules achieve higher performance than the corresponding constant temperature baseline. The resulting $p$-values are presented in the table 6. In the majority of cases, our schedules achieve the minimum possible $p$-value ($p = 0.03125$), indicating a consistent improvement across all trials. Even in cases with slightly higher $p$-values, our schedule surpasses the baseline in the majority of trials. We underline the only two instances where our schedule does not outperform the baseline and a higher p-value is expected.

## E.3 Impact of Different Inference Scoring Rules

To test the robustness, we evaluate our schedules across multiple other OSR scoring rules, such as the energy score (Liu et al., 2020), ODIN (Liang et al., 2017), Cosine-margin (Deng et al., 2019), max-logit, confidence (or max-probability) and OpenAUC (Wang et al., 2022) scores. While the OSCR curve measures the trade-off between CCR and FPR across all thresholds, the OpenAUC is a simplified threshold-free ranking score that can be expressed as the sum of pair-wise loss terms and removes the need to calculate the numerical integral with histograms. The results presented in Table 7 confirm that our TSs outperform the baseline in the majority of cases, irrespective of the scoring rule used. The improved representations learned through our schedules confirm that the performance benefits are transferable across different scoring rules.

Table 5: Performance standard deviation of constant baseline and our NegCosSch on different losses across various seeds.

| Loss | Schedule | Accuracy (%) | AUROC (%) | OSCR (%) | Accuracy (%) | AUROC (%) | OSCR (%) |
|---|---|---|---|---|---|---|---|
| | | | **CUB** | | | **Aircraft** | |
| CE (w/o LS) | Constant | 0.2 | 0.26 / 0.26 | 0.41 / 0.37 | 0.39 | 0.77 / 0.52 | 0.42 / 0.47 |
| | M-NegCosSch(**ours**) | 0.27 | 0.44 / 0.28 | 0.6 / 0.45 | 0.31 | 0.15 / 0.47 | 0.26 / 0.3 |
| | P-NegCosSch(**ours**) | 0.47 | 0.38 / 0.23 | 0.34 / 0.4 | 0.34 | 0.25 / 0.43 | 0.26 / 0.5 |
| CE + LS (Vaze et al. (2022)) | Constant | 0.14 | 0.58 / 0.34 | 0.4 / 0.23 | 0.16 | 0.27 / 0.54 | 0.19 / 0.49 |
| | M-NegCosSch(**ours**) | 0.34 | 0.59 / 0.36 | 0.59 / 0.38 | 0.26 | 0.58 / 0.29 | 0.54 / 0.18 |
| | P-NegCosSch(**ours**) | 0.22 | 0.77 / 0.31 | 0.66 / 0.41 | 0.25 | 0.83 / 0.35 | 0.77 / 0.22 |
| SupCon (w/o LS) | Constant | 0.29 | 0.23 / 0.29 | 0.35 / 0.23 | 0.58 | 1.79 / 0.53 | 1.99 / 0.77 |
| | M-NegCosSch(**ours**) | 0.28 | 0.18 / 0.38 | 0.34 / 0.53 | 0.28 | 0.38 / 0.24 | 0.39 / 0.34 |
| | P-NegCosSch(**ours**) | 0.21 | 0.13 / 0.31 | 0.26 / 0.35 | 1.08 | 1.37 / 0.55 | 2.07 / 1.18 |
| SupCon + LS | Constant | 0.2 | 0.24 / 0.32 | 0.44 / 0.38 | 0.69 | 1.03 / 0.55 | 1.35 / 0.85 |
| | M-NegCosSch(**ours**) | 0.38 | 0.36 / 0.17 | 0.53 / 0.18 | 0.19 | 0.41 / 0.37 | 0.31 / 0.43 |
| | P-NegCosSch(**ours**) | 0.32 | 0.62 / 0.4 | 0.78 / 0.57 | 0.53 | 1.77 / 0.42 | 1.82 / 0.51 |
| ARPL (Chen et al. (2021)) | Constant | 0.17 | 0.55 0.56 | 0.37 0.47 | 0.46 | 0.47 0.56 | 0.41 0.58 |
| | M-NegCosSch(**ours**) | 0.19 | 0.85 / 0.42 | 0.54 / 0.26 | 0.42 | 0.62 / 0.59 | 0.67 / 0.51 |
| | P-NegCosSch(**ours**) | 0.17 | 0.87 / 0.45 | 0.49 / 0.3 | 0.47 | 0.63 / 0.58 | 0.62 / 0.57 |
| | | | **SCars** | | | **TinyImageNet** | |
| CE (w/o LS) | Constant | 0.1 | 0.52 / 0.49 | 0.51 / 0.52 | 0.2 | 0.21 | 0.23 |
| | M-NegCosSch(**ours**) | 0.09 | 0.22 / 0.51 | 0.26 / 0.55 | 0.27 | 0.27 | 0.32 |
| | P-NegCosSch(**ours**) | 0.1 | 0.29 / 0.62 | 0.21 / 0.59 | 0.38 | 0.32 | 0.4 |
| CE + LS | Constant | 0.1 | 0.26 / 0.22 | 0.25 / 0.25 | 0.25 | 0.24 | 0.21 |
| | M-NegCosSch(**ours**) | 0.08 | 0.3 / 0.32 | 0.33 / 0.29 | 0.15 | 0.22 | 0.19 |
| | P-NegCosSch(**ours**) | 0.14 | 0.26 / 0.3 | 0.34 / 0.31 | 0.29 | 0.37 | 0.36 |
| SupCon (w/o LS) | Constant | 0.19 | 0.12 / 0.45 | 0.25 / 0.39 | 0.18 | 0.01 | 0.17 |
| | M-NegCosSch(**ours**) | 0.15 | 0.18 / 0.18 | 0.18 / 0.24 | 0.2 | 0.04 | 0.15 |
| | P-NegCosSch(**ours**) | 0.14 | 0.2 / 0.29 | 0.2 / 0.32 | 0.37 | 0.08 | 0.33 |
| SupCon + LS | Constant | 0.06 | 0.18 / 0.36 | 0.19 / 0.35 | 0.25 | 0.16 | 0.27 |
| | M-NegCosSch(**ours**) | 0.1 | 0.2 / 0.38 | 0.23 / 0.33 | 0.16 | 0.16 | 0.09 |
| | P-NegCosSch(**ours**) | 0.05 | 0.19 / 0.25 | 0.19 / 0.25 | 0.15 | 0.12 | 0.14 |
| ARPL | Constant | 0.47 | 0.08 / 0.92 | 0.27 / 0.79 | 0.1 | 0.13 | 0.12 |
| | M-NegCosSch(**ours**) | 0.34 | 0.13 / 0.5 | 0.07 / 0.37 | 0.37 | 0.29 | 0.24 |
| | P-NegCosSch(**ours**) | 0.19 | 0.12 / 0.67 | 0.18 / 0.7 | 0.17 | 0.25 | 0.17 |

Table 6: $p$-values from one-sided Wilcoxon rank tests comparing the proposed schedules to the baseline over 5 random trials. A $p$-value of 0.031 indicates the proposed schedule outperforms the baseline in all 5 trials. We underline the only two instances where our schedule does not outperform the baseline and a higher p-value is expected.

| Loss | comparing schedule | Accuracy (%) | AUROC (%) | OSCR (%) | Accuracy (%) | AUROC (%) | OSCR (%) |
|---|---|---|---|---|---|---|---|
| | | | **CUB** | | | **Aircraft** | |
| CE | M-NegCosSch | 0.03 | 0.03 / 0.03 | 0.03 / 0.03 | 0.09 | 0.03 / 0.03 | 0.03 / 0.03 |
| | P-NegCosSch | 0.03 | 0.03 / 0.03 | 0.03 / 0.03 | 0.03 | 0.03 / 0.03 | 0.03 / 0.03 |
| SupCon | M-NegCosSch | 0.03 | 0.03 / 0.03 | 0.03 / 0.03 | 0.03 | 0.06 / 0.06 | 0.06 / 0.06 |
| | P-NegCosSch | 0.03 | 0.03 / 0.03 | 0.03 / 0.03 | 0.19 | 0.15 / 0.15 | 0.31 / 0.31 |
| | | | **Scars** | | | **TinyImageNet** | |
| CE | M-NegCosSch | 0.03 | 0.03 / 0.03 | 0.03 / 0.03 | 0.15 | 0.15 | 0.03 |
| | P-NegCosSch | 0.03 | 0.03 / 0.03 | 0.03 / 0.03 | 0.35 | 0.15 | 0.59 |
| SupCon | M-NegCosSch | 0.09 | 0.03 / 0.59 | 0.03 / 0.21 | 0.15 | 0.03 | 0.03 |
| | P-NegCosSch | 0.31 | 0.03 / 0.15 | 0.06 / 0.15 | 0.94 | 0.03 | 0.31 |

## E.4 Performance on Vision Transformer

We evaluate our proposed NegCosSch on SSBs using a tiny vision transformer (ViT) (Wu et al., 2022), to demonstrate the robustness and general applicability of our schedules across the contemporary ViT architectures, with the results are presented in Table 8. The improvements observed in this table confirm the benefits of our temperature modulation when integrated into a transformer backbone. The

Table 7: Evaluation of robustness to different open-set scoring rules. Our proposed schedules maintain a performance improvement across all tested scoring rules for the majority of the cases.

| | metric → scoring rule → | Max-logit | Max-prob. | AUROC Energy | ODIN | cosine head | OpenAUC Max-logit |
|---|---|---|---|---|---|---|---|
| Loss | schedule | | | **CUB** | | | |
| CE | Constant | 83.55 / 74.98 | 83.81 / 78.05 | 83.11 / 74.19 | 83.17 / 73.26 | 85.33 / 76.26 | 70.5 / 63.35 |
| | M-NegCoSch (ours) | **86.79 / 78.08** | **86.43 / 80** | **87.21 / 78.18** | **87.6 / 77.38** | **88.05 / 78.86** | **74.71 / 67.31** |
| | P-NegCoSch (ours) | **86.85 / 77.6** | **86 / 79.76** | **86.91 / 77.55** | **87.04 / 76.71** | **88.09 / 78.47** | **74.9 / 67.01** |
| SupCon | Constant | 86.94 / 73.95 | 86.48 / 76.35 | 86.05 / 72.57 | 86.75 / 73.2 | 86.57 / 76.24 | 72.42 / 61.67 |
| | M-NegCoSch (ours) | **88.14 / 75.81** | **87.82 / 77.27** | **87.56 / 74.35** | **88.21 / 74.81** | **88.66 / 78.34** | **75.1 / 64.73** |
| | P-NegCoSch (ours) | **87.5 / 74.95** | **87.3 / 76.78** | **87.05 / 73.75** | **87.69 / 74.31** | **87.9 / 77.69** | **73.55 / 63.14** |
| | | | | **Aircraft** | | | |
| CE | Constant | 90.35 / 81.48 | 84.96 / 81.53 | 87.63 / 81.05 | 87.3 / 79.64 | 85.05 / 80 | 82.05 / 74.26 |
| | M-NegCoSch (ours) | **91.15 / 83.23** | **88.34 / 82.18** | **92.19 / 84.67** | **92.64 / 84.5** | **89.26 / 83.42** | **83 / 76.01** |
| | P-NegCoSch (ours) | **91.41 / 83.15** | **85.24 / 79.98** | **89.1 / 82.11** | **89.32 / 81.13** | **86.56 / 81.12** | **83.44 / 76.15** |
| SupCon | Constant | 88.78 / 81.79 | 84.59 / 80.66 | 85.96 / 81.1 | 85.84 / 80.5 | 84.1 / 80.79 | 80.52 / 74.39 |
| | M-NegCoSch (ours) | **90.45 / 82.49** | **90.55 / 82.3** | **90.84 / 82.46** | **91.14 / 82.37** | **91.33 / 81.93** | **82.57 / 75.52** |
| | P-NegCoSch (ours) | **89.27 / 81.97** | **86.29 / 81.27** | **87.11 / 81.44** | **87.43 / 80.97** | **85.48 / 81.55** | **80.97 / 74.56** |
| | | | | **SCars** | | | |
| CE | Constant | 94.03 / 84.82 | 93.82 / 84.99 | 93.04 / 84.2 | 92.89 / 83.93 | 93.49 / 84.16 | 91.05 / 82.2 |
| | M-NegCoSch (ours) | **95.18 / 86.26** | **94.84 / 83.58** | **95.14 / 84.86** | **95.15 / 84.8** | **94.78 / 84.41** | **92.57 / 83.96** |
| | P-NegCoSch (ours) | **95.03 / 86.05** | **94.59 / 85.68** | **94.55 / 86.18** | **94.49 / 86.11** | **94.28 / 85.4** | **92.5 / 83.82** |
| SupCon | Constant | 92.99 / 82.8 | 93.71 / 83.56 | 92.41 / 82.42 | 93.07 / 82.85 | 94.28 / 83.57 | 89.93 / 80.13 |
| | M-NegCoSch (ours) | **93.57 / 82.76** | **94.34 / 83.1** | **93.39 / 82.26** | **93.88 / 82.56** | **95.1 / 83.25** | **90.67 / 80.27** |
| | P-NegCoSch (ours) | **93.32 / 83.16** | **93.73 / 83.39** | **92.72 / 82.56** | **93.23 / 82.9** | **94.13 / 82.84** | **90.32 / 80.54** |
| | | | | **TinyImageNet** | | | |
| CE | Constant | 78.6 | 79.59 | 80.95 | 78.39 | 80.11 | 69.23 |
| | M-NegCoSch (ours) | **79.21** | 78.11 | **81.19** | 78.09 | **80.44** | **69.85** |
| | P-NegCoSch (ours) | **79.05** | 77.65 | **81.09** | 78.08 | **80.37** | **69.92** |
| SupCon | Constant | 82.87 | 83.05 | 82.93 | 80.12 | 81.16 | 70.63 |
| | M-NegCoSch (ours) | **83.21** | 82.99 | **83.11** | **80.2** | **81.49** | **70.9** |
| | P-NegCoSch (ours) | **83.09** | 82.9 | **83.05** | **80.19** | **81.42** | **70.72** |

results in the table, along with previous results from VGG and ResNet-based architectures, confirm the applicability of our proposed TSs across a diverse range of model architectures.

Table 8: Performance on SSBs with a tiny ViT architecture.

| | CUB | | | Aircraft | | | SCars | | |
|---|---|---|---|---|---|---|---|---|---|
| method | Acc. (%) | AUROC (%) | OSCR (%) | Acc. (%) | AUROC (%) | OSCR (%) | Acc. (%) | AUROC (%) | OSCR (%) |
| Const. (baseline) | 90.83 | 91.41 / 79.88 | 82.99 / 72.59 | 88.26 | 87.75 / 76.31 | 77.82 / 68.08 | 95.62 | 92.5 / 83.67 | 88.54 / 80.13 |
| M-NegCosSch (ours) | 91.07 | 92.06 / 80.67 | 83.79 / 73.51 | 88.51 | 89/78.78 | 79.12/70.41 | 95.8 | 93.1/83 | 89.28/79.61 |
| P-NegCosSch (ours) | 91.2 | 92.3 / 80.04 | 84.1 / 73.03 | 88.92 | 88.65/79.68 | 79.12/71.46 | 96.03 | 93.08/83.08 | 89.47/79.9 |

## E.5 ABLATIONS ON $(\tau^+, \tau^-)$

Figure 4 presents the OSR performance of our NegCosSch along with the regular cosine TS and constant temperatures in SupCon loss on the regular benchmarks. We vary $(\tau^+, \tau^-)$ from $\mathbb{T}^2_{SupCon}$ and compare $\mathcal{T}_{\text{NegCosSch}}(\tau^+, \tau^-, P)$ with $\mathcal{T}_{\text{CosSch}}(\tau^+, \tau^-, P)$, $\mathcal{T}_{\text{Const}}(\tau = \text{nearest}(\frac{1}{2}[\tau^+ + \tau^-]))^3$ and $\mathcal{T}_{\text{Const}}(\tau = \tau^-)$. The objective of the quadruplet-wise comparisons is to determine if our proposed TS outperforms a regular cosine TS, a constant temperature set to the midpoint of $(\tau^+, \tau^-)$, or set to $\tau^-$ with various pairs of $(\tau^+, \tau^-)$. We observe that for the CIFAR10, CIFAR+10 and TinyImageNet benchmarks, our proposed TS yields a better open set AUROC than CosSch and constant temperatures for most of the quadruplet comparisons. We find the improvements or degradations to be insignificant for the CIFAR+50 benchmark with the highest AUROC found for $\mathcal{T}_{\text{CosSch}}(0.3, 0.1)$. By observing the best performances of our NegCosSch, we formulate the strategy mention in Section 4.3 for choosing $(\tau^+, \tau^-)$.

---

[3] By 'nearest', we mean a nearest temperature is chosen from $\mathbb{T}_{\text{SupCon}}$.

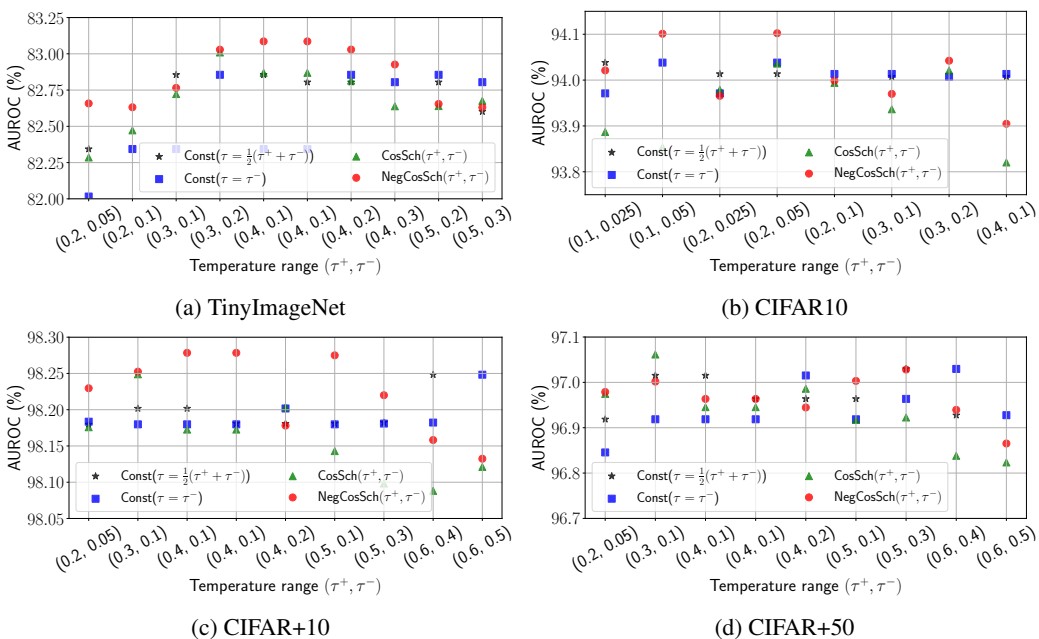

Figure 4: Open Set AUROC of different TSs for the SupCon loss on the regular OSR benchmarks.

### E.6 ABLATIONS ON $P$ AND $k$ IN $\mathcal{T}_{\text{GCosSch}}$

We perform an ablation study on $P$ in $\mathcal{T}_{\text{NegCosSch}}$ using TinyImageNet with the SupCon loss for different pairs of $(\tau^+, \tau^-)$ and the open set AUROC are presented in Table 9. We observe that different choices of $P$ produce similar OSR performance.

We also compare the open set AUROC on TinyImageNet among different values of $k$ in $\mathcal{T}_{\text{GCosSch}}$. The temperatures are set to: $\tau^+ = 0.4$ and $\tau^- = 0.1$. From Table 10, we observe that the open set AUROC increases with the value of $k$, with the highest AUROC observed for $k = 1$ or our NegCosSch.

Table 9: Open set AUROC (%) on TinyImageNet for different values of $P$ in $\mathcal{T}_{\text{NegCosSch}}$.

| $(\tau^+, \tau^-) \setminus P \rightarrow$ | 100 | 200 |
|---|---|---|
| (0.3,0.2) | 83.10 | 83.03 |
| (0.4,0.1) | 82.91 | 83.09 |
| (0.4,0.2) | 82.99 | 83.03 |

Table 10: Open set AUROC on TinyImageNet for different values of $k$ in $\mathcal{T}_{\text{GCosSch}}$.

| Schedule | AUROC (%) |
|---|---|
| $\mathcal{T}_{\text{GCosSch}}(k = 0)$ or $\mathcal{T}_{\text{CosSch}}$ | 82.87 |
| $\mathcal{T}_{\text{GCosSch}}(k = 0.25)$ | 82.93 |
| $\mathcal{T}_{\text{GCosSch}}(k = 0.50)$ | 82.99 |
| $\mathcal{T}_{\text{GCosSch}}(k = 0.75)$ | 83.03 |
| $\mathcal{T}_{\text{GCosSch}}(k = 1)$ or $\mathcal{T}_{\text{NegCosSch}}$ | **83.09** |

### E.7 PERFORMANCE ON PROTOTYPICAL CONTRASTIVE LEARNING

We also show results with our NegCosSch on TinyImageNet and Aircraft using the prototypical contrastive (ProtoCon) learning in Table 11. The ProtoCon loss is recently used by Bahavan et al. (2025); Li et al. (2025) for OSR. We observe improvements for all three metrics.

In ProtoCon, instead of contrasting an anchor representation with another sample, we contrast with the prototypes of known classes. We randomly initialize one prototype per known class $\{p_c\}_{c=1}^C$. The loss function forces all representations of the same class to lie near its prototype and to move away from other prototypes, which is given as:

$$L_{ProtoCon} = -\frac{1}{|I|} \sum_{i \in I} \log \frac{\exp(\text{sim}(l_i, p_{\tilde{y}_i})/\tau)}{\sum_{c=1}^C \exp(\text{sim}(l_i, p_c)/\tau)} \tag{15}$$

We update the prototypes on the fly at each iteration $t$. $\sigma$ is the learning rate for prototypes.

$$p_c^t = \begin{cases} p_c^{t-1} & ; \text{ if } |\{i \in I : \tilde{y}_i = c\}| = 0 \\ (1 - \sigma)p_c^{t-1} + \frac{\sigma}{|\{i \in I : \tilde{y}_i = c\}|} \sum_{\{i \in I : \tilde{y}_i = c\}} l_i, & ; \text{ otherwise} \end{cases} \tag{16}$$

Table 11: Performance using prototypical contrastive learning

| TS | TinyImageNet | | | Aircraft | | |
|---|---|---|---|---|---|---|
| | Accuracy (%) | AUROC (%) | OSCR (%) | Accuracy (%) | AUROC (%) | OSCR (%) |
| Const. (baseline) | 85.36 | 82.79 | 70.53 | 86.29 | 84.32 / 78.46 | 72.28 / 68.26 |
| NegCosSch | **85.72** | **83.04** | **71.12** | **87.03** | **84.93 / 79.01** | **73.29 / 66.33** |

### E.8 PERFORMANCE ON THE CIFAR BENCHMARKS

Here, we evaluate our periodic NegCosSch using the SupCon loss on the CIFAR benchmarks – such as CIFAR10, CIFAR+10 and CIFAR+50 and the results are presented in Table 12. The values of $\tau$ for $\mathcal{T}_{\text{Const}}$ are chosen as $0.05, 0.5$, and $0.4$ respectively with hyperparameter tuning for the CIFAR10, CIFAR+10, and CIFAR+50 benchmarks and the values of $(\tau^+, \tau^-)$ in our TS are $(0.2, 0.05), (0.4, 0.1)$, and $(0.5, 0.3)$ respectively. We observe that the closed set accuracy is similar to the baseline methods when we include our TS on these benchmarks, whereas we gain slight improvements in the open set performance. The open set performance depends on the nature of the unknown classes and their semantic similarity with the known classes. We suspect that the benefits of our TS reduce when the number of training classes is relatively small, which occur in the CIFAR benchmarks. For example, there are only 6 training classes in CIFAR10 and 4 training classes in the CIFAR+10 and CIFAR+50 benchmarks. Moreover, the OSR AUROC on the CIFAR+10 and CIFAR+50 benchmarks are $> 97\%$ with tuned constant temperature baselines, leaving only a little scope for improvements. However, as mentioned before, we observe significant improvements both for the open set and closed set performance on the TinyImageNet and the SSBs, where they have a larger number of training classes.

Table 12: Closed set accuracy, open set AUROC and OSCR (in %) for the SupCon baseline without and including the proposed NegCosSch on the CIFAR benchmarks.

| Methods | CIFAR10 | | | CIFAR+10 | | | CIFAR+50 | | |
|---|---|---|---|---|---|---|---|---|---|
| | Accuracy | AUROC | OSCR | Accuracy | AUROC | OSCR | Accuracy | AUROC | OSCR |
| Const. (Baseline) | 96.95 | 94.04 | 91.13 | 98.05 | 98.25 | 96.32 | 98.13 | 97.03 | 95.21 |
| NegCosSch (**ours**) | 96.91 | **94.10** | **91.18** | 98.02 | **98.28** | **96.33** | 98.10 | 97.03 | 95.17 |

## F RELATED WORKS (CONTINUED)

Here, we discuss the recent OSR methods in details. Wang et al. (2024) propose to extract diverse features from multiple experts with an attention diversity regularization to ensure the attention maps are mutually different. Zhou et al. (2024a) propose a framework with contrastive training for classification and implement an additional VAE for reconstruction to compute an unknown score based on intermediate features. Yang et al. (2024b) propose a self-learning framework for test time adaptation.

Another line of work utilizes data augmentation. For example, Jia et al. (2024) propose an asymmetric distillation to feed the teacher model with extra data through augmentation, filtering out the wrong prediction from the teacher model and assigning a revised label to them to train the student model. The method in Wang et al. (2025) augments the dataset by mixing the foreground of images with different backgrounds. Xu & Keuper propose new data augmentation with the help of visual explanation techniques, such as the LayerGAM to mask out the activated areas so that models can learn beyond the discriminative features.

The other methods are based on contrastive learning with different regularization. For example, Xu et al. (2023); Li et al. (2024) train models with contrastive loss, sample mix up and label smoothing

for better representation learning. Bahavan et al. (2025) also propose a prototypical contrastive loss to pull all samples to its class prototype and push away the prototypes of other classes. Li et al. (2025) propose a regularization inspired from the neural collapse perspective – the closed set classes are aligned with a simplex equiangular tight frame geometric structure. Recent works by Zhou et al. (2024b); Hua et al. (2025) introduce open world prompt tuning methods that improve a vision language model's performance in an open-world scenario to make better predictions from a mix of known and unknown classes.

Although the recent methods aim for better representation learning, some of them achieve this through feeding more data to the model with augmentation. On the other side, a few recent OSR methods do not use the same experiments settings maintained in most of works in the literature. For example, Wang et al. (2025; 2024); Jia et al. (2024) use different backbone models for evaluation, which makes it harder to compare their methods with others.

## G   IMPLEMENTATION

Each model is trained on a single NVIDIA-RTX2080Ti GPU requiring from 2 to 32 hours depending on the model and the dataset. Our implementation utilizes Python (v3.7) and PyTorch (v1.12), accelerated with CUDA (v11.3) and cuDNN (v8.2). Our codes are mostly built on top of the codebase by Vaze et al. (2022) and the implementation of SupCon loss is taken from the official GitHub page by Khosla et al. (2020). Our periodic NegCosSch schedule can be integrated into any existing loss with a few lines of codes as the following:

```python
import math
class GCosineTemperatureScheduler:
  def __init__(self, t_p=2.0,t_m=0.5, P=200,shift=1.0,epochs=600):
    self.t_p = t_p
    self.t_m = t_m
    self.epochs = epochs
    self.P = P
    self.s = shift
    self.e = int(self.epochs - 0.5 * self.s * self.P)
  def get_temperature(self, epoch):
    if(t<self.e):
      t = self.t_m + (self.t_p - self.t_m) *
        (1+ math.cos(2*math.pi* (epoch-self.s * self.P/2)/self.P))/2
    else:
      t = self.t_p
    return t

if(args.temperature_scheduling):
  TS=GCosineTemperatureScheduler()
for epoch in range(1,N_epochs+1):
  if(args.temperature_scheduling):
    criterion.temperature = TS.get_temperature(epoch)
  # rest of the code
  ... ...
```

