# OpenReview forum: "Boosting Open Set Recognition Performance through Modulated Representation Learning"
_ICLR.cc/2026/Conference — ICLR 2026 Poster_

### Official Review · Reviewer_TEAQ · 2025-10-23

**Soundness:** 2
**Presentation:** 3
**Contribution:** 3
**Rating:** 6
**Confidence:** 4

**Summary:**

The paper addresses open-set recognition by replacing the fixed training temperature used in common OSR losses with epoch-wise temperature schedules that modulate representation learning from instance-level to semantic-level focus.

It proposes a generalized cosine framework and a new negative cosine schedule (NegCosSch) that starts at a low temperature to form coarse, separative boundaries and then increases to compact within-class clusters; linear and exponential increasing variants are also considered. The schedules plug into Cross-Entropy, Supervised Contrastive, and ARPL losses without extra computational cost.

Experiments on TinyImageNet and the Semantic Shift Benchmarks (CUB, FGVC-Aircraft, Stanford Cars) demonstrate consistent gains over constant-temperature and prior cosine schedules in closed-set accuracy and open-set metrics (AUROC, OSCR), with improvements that tend to increase as the number of training classes increases.

The claimed contributions are (i) analysis of temperature effects for OSR, (ii) plug-and-play schedules, especially NegCosSch, applicable across losses, and (iii) empirical improvements on tougher benchmarks.

**Strengths:**

The paper’s strengths span four fronts. On originality, it repurposes temperature scheduling, previously explored mostly for closed-set/self-supervised contrastive learning, into a unified, loss-agnostic mechanism for open-set recognition, with a negative cosine schedule that explicitly traverses instance→semantic regimes. This is a simple yet creative removal of the fixed-τ limitation that has constrained prior OSR training dynamics.

In terms of quality, the method is evaluated across multiple datasets of varying difficulty, multiple loss families (CE, SupCon, ARPL), and reports averages over five random seeds, showing consistent AUROC/OSCR and accuracy gains without requiring extra computation or architectural changes.

The paper clearly motivates the role of temperature, presents the proposed schedules using a generalized cosine formulation, and supplements intuition with gradient analyses and UMAP visualizations that illustrate how clusters evolve during training.

In terms of significance, the contribution is pragmatically valuable: it provides a drop-in schedule that improves both open- and closed-set metrics, scales favorably as the number of training classes increases, and can be integrated into existing OSR pipelines with negligible engineering overhead, thereby lowering the barrier to adoption on more challenging semantic-shift benchmarks.

**Weaknesses:**

The paper’s novelty is primarily curatorial, adapting known temperature-scheduling ideas (e.g., cosine schemes) to OSR, without a stronger theoretical account tying the proposed schedules to measurable properties of open-set boundaries; parameter choices such as (τ+, τ−), the period P, and the “finish high-τ” heuristic are fixed largely by heuristic guidance rather than principled sensitivity analyses.

Empirically, improvements are modest and reported without dispersion or significance testing; although the text states five seeds were used, tables report means only, so it is unclear whether +1–3% AUROC/OSCR gains exceed run-to-run noise.

Evaluation breadth is limited: results focus on TinyImageNet and SSBs with a narrow architecture set (VGG-like, ResNet-50; ViT deferred to the appendix) and do not probe domain shift or cross-dataset OSR, leaving external validity uncertain. Comparisons emphasize constant-τ and cosine baselines; several competitive contemporary OSR methods in the related work are not reproduced under the same protocol, making relative advantage ambiguous.

Finally, the “no computational overhead” claim overlooks the practical cost of tuning τ-ranges and schedule shapes, as well as the UMAP visualizations. While intuitive, these visualizations lack quantitative cluster diagnostics (e.g., intra/inter-class scatter, uniformity) to substantiate the representation-learning narrative.

**Questions:**

1. Describe the dispersion and significance of improvements. Report mean±std over the five seeds you mention and include 95% CIs and paired tests for AUROC/OSCR/accuracy across all tables; currently tables list means only.

2. Run factorial sweeps over (τ+, τ−) and period P for NegCosSch and monotonic variants, showing Pareto frontiers of closed- vs open-set metrics. Your choices (e.g., τ+∈{0.3–0.4}, τ−=0.1; “finish with high τ”) are presently heuristic. Quantify the robustness of performance to these settings.

3. Justify why starting low-τ then increasing is fundamentally better for OSR.

4. You use max-logit for unknown detection under SupCon/CE. Compare against energy score, ODIN-style perturbations, and cosine-margin heads, holding the backbone/schedule fixed. Does NegCosSch still win irrespective of the scoring rule?

5. Breadth and strength of baselines
   Reproduce stronger contemporary OSR baselines under your exact protocol/splits, not only constant-τ or CosSch: e.g., BackMix (TPAMI 2025), iCausalOSR (PR 2024a). This clarifies whether scheduling alone is competitive at today’s frontier.

6. You claim plug-and-play for “any OSR loss”. Test at least one additional family—e.g., OpenAUC-style objectives or prototype-based methods—and report whether schedule benefits persist without retuning.

---

> ### Author Response · Authors · 2025-11-21
> **Response to comments by Reviewer  TEAQ**
>
> **Overall Response:** We thank the reviewer for mentioning the strengths of our work in detail and expressing their constructive concerns. We hope that the following discussions with additional results will provide further clarifications for a better understanding of the work. We believe that addressing the concerns with additional experimental results will certainly improve the quality of the paper. We have uploaded the revised manuscript.
>
> **Comment:** The paper’s novelty is primarily curatorial, .... heuristic guidance rather than principled sensitivity analyses.
>
> **Response:**
>
> The reviewer expressed concerns regarding the novelty of the work, theoretical tie to measurable properties and parameter sensitivity analysis. Our response touches on the three aspects. First, we would like to clarify and reiterate the key distinctions of our proposed schedule from prior work. Second, we want to further clarify on the fact that our derived heuristics are guided by our sensitivity analysis and finally, we would like to address the measurable properties of open-set boundaries.
>
> **Novelty:**
>
> The novelty our work lies in the following aspects:
>
> **Why our temperature modulation works better for OSR.** The work by Kukleva et al. [2023] along with the other previous works, such as Wang and Liu [2021], analyze the effect of temperature scaling, in the context of **closed set self-supervised contrastive learning and applied on long-tailed data**. However, in our work, we **extend the effect of temperature scaling in the context of supervised learning and in an open set scenario**. Our temperature modulation is not just a mere variation of the existing cosine temperature scheduling, but the novelty lies in its **clever way for performing modulation** using our negative cosine scheduling to achieve better representation learning for both tasks.
>
> Some existing OSR methods propose to create more empty regions in the representation space. Forcing the creation of empty spaces in the representation space with additional regularization, such as in the adversarial reciprocal point learning (ARPL) by Chen et al. [2021], does not result in a better open set recognition (please see lines 50-53 in page 1). In such a case, the model learns such feature space so that there are some similarities between the harder novel and known samples, making the harder unknown samples unable to be detected, whereas **our method benefits from traversing the spectrum of instance-specific to class-specific features.**
>
> We explain how our temperature modulation should create more separation among the known and the unknown classes. Please see Section 5.3.
>
> **Extension to multiple loss types.** We demonstrate that our NegCosSch brings improvement for multiple types of losses. The work in Kukleva et al. [2023] does not show the applicability of temperature modulation for the cross-entropy loss, which is still a widely applied loss function for many applications.
>
> **Significant improvement on the tougher cases.** Finally, we gain significant improvements for the semantic shift benchmarks, which was **mostly ignored in the previous literature** despite having a good number of new open set recognition methods since their introduction.
>
> However, we appreciate the concern of the reviewer regarding the novelty of our work and hope that this discussion clarifies further on the novelty. In the revised draft, we have added the details briefly to focus on these key differences from the existing literature.

---

> ### Author Response · Authors · 2025-11-21
>
> **Justification for the derived heuristic.**
>
> The users do not need to explicitly tune for both $\tau^+$ and $\tau^-$. To support this, we show that the proposed heuristics, such as $\tau^- = \tau - \Delta$, $\tau^+ = \tau + \Delta$, are **not mere heuristics but they are guided by parameter sensitivity analysis**. The **heuristic is designed to maintain structural control**: maximizing performance by ensuring the range is centered precisely around the optimal operating point while avoiding structural collapse caused by excessive perturbation in the temperature.
>
> We clarify that the single optimal value for constant temperature $\tau$ is inherently dataset-dependent and must be determined by empirical tuning.
>
> Our heuristic relationship ($\tau^+ = \tau + \Delta$, $\tau^- = \tau - \Delta$) is guided by our analysis in Section 3. Since our gradient analysis established that the optimal $\tau$ provides the best starting trade-off between the instance-specific and class-specific features in the constant temperature regime, our schedule is engineered to perturb this optimum ($\tau$ serves as the center in our heuristics).
>
> Crucially, if the range becomes too wide with large $\Delta$, and the structural stability in the representation space breaks down, as the high $\Delta$ disrupts the cluster boundaries beyond recovery. Starting with excessively low $\tau^-$ disrupts the initial formation of the semantic structure, which may not be recovered later. Furthermore, an excessively high $\tau^+$ encourages over-adjustment from the learned features in the low-$\tau$ phase, pushing the model toward extreme class-specific features and removing instance-level discriminating power.
>
> **We derived our parameter heuristics from a systematic analysis.**
>
> For this purpose, we highlight the experiments presented in **Figure 4** Appendix F.5. For example, in Figure 4(a), we varied the range $(\tau^+, \tau^-)$ in our schedules and $\tau$ for constant baseline from {0.025, 0.05, 0.1, 0.2, 0.3, 0.4, 0.5} while implementing SupCon on TinyImageNet and performed the sensitivity analysis. We present the results in the form of quadruplet-wise comparisons in Figure 4 among the obtained performances. In these quadruplet comparisons, we compare our NegCosSch($\tau^+$, $\tau^-$) with CosSch($\tau^+$, $\tau^-$)  and Const($\tau$) when the constant temperature is either set to the mid point of the range or to lower value. We observe that our proposed TS yields a better performance than CosSch and constant temperatures for most quadruplet comparisons.
>
> **These results derive our proposed heuristics** that for a given $\tau$, we can choose $\tau^- = \tau - \Delta, \tau^+ = \tau + \Delta$ and this works reasonably well without the need to extensive tuning both $\tau^+$ and $\tau^-$. The proposed heuristic is a principled simplification that is derived directly from the observed performance trends in Figure 4, which showed that the relationship between $\tau^+$ and $\tau^-$ is consistently good.
>
> **Choice of $P$.** Based on this observation (Table 9), P=200 can be a robust default choice for our P-NegCosSch that performs well across our benchmarks as we demonstrated in most of our results. The main takeaway is that the users do not need to invest in a comprehensive tuning for P; a reasonable value is sufficient. Users who wish to avoid this hyperparameter entirely can adopt our **M-NegCosSch method as a competitive alternative** as shown in most of our results in Table 1 and 2 that does not require setting a period P, thus incurring no tuning expense for this parameter.
>
> We iterate that the heuristic relationship between parameters, which were actually derived from the sensitivity analysis. However, we understand that it was not well mentioned in the previous submission. We have added these details for further clarification (lines 305 - 313 in Section 4.3) in the revised manuscript.
>
> **Theoretical tie to measurable properties of open set boundaries:**
>
> We reiterate that our design is rooted in theory, starting with a gradient analysis under constant temperature scaling (Section 3). These gradient results align with existing theoretical analyses in the self-supervised domain and serve us to understand the foundational geometric properties for constant scaling factors at both extremes.
>
> We utilize this geometric understanding—the specific pattern of scatter and margin created by constant temperature in the two extreme ends—to propose a novel temperature modulation. Our schedule is precisely engineered to systematically traverse the trade-off, better than any single fixed temperature.
>
> Moreover, following your suggestion, we **compute the geometric properties in the representation space** to analyze both closed set and open set boundaries: intra/inter-class scatter and the average distance of unknown samples to the nearest prototype. **Please see the response to "lack quantitative cluster diagnostics".**

---

> ### Author Response · Authors · 2025-11-21
>
> $\textbf{Comment:}$ Empirically, improvements are modest and reported without dispersion or significance testing; ... ... whether +1–3% AUROC/OSCR gains exceed run-to-run noise.
>
> $\textbf{Response:}$
>
> We thank the reviewer for emphasizing the importance of statistical rigor.
>
> Regarding the reporting of dispersion, please note that we $\textbf{already report the standard deviations across seeds in Table 4 and Table 5 in Appendix F.2.}$ We presented these separately from the main performance table to maintain readability within space constraints, but they serve to demonstrate the stability of our method across the random trials. The proposed schedules demonstrate $\textbf{similar standard deviations compared to the corresponding baselines}$ and the other TSs (presented in Table 4) and for all losses (presented in Table 5), while $\textbf{achieving higher average performance}$ for the three metrics in most cases.
>
> Furthermore, we $\textbf{perform}$ a one-sided non-parametric Wilcoxon rank test to evaluate the $\textbf{statistical significance}$ for the results on the CE and SupCon losses presented in Table 2. We test the hypothesis that our proposed schedules (M-NegCosSch and P-NegCosSch) achieve higher performance than the corresponding baseline (constant temperature). The resulting p-values are presented in $\textbf{Table 6 in the revised manuscript}$.
>
> We observe that in the majority of cases, our schedule achieves the $\textbf{minimum possible p-value}$ (p=0.03125) for 5 trials, indicating a consistent improvement across all trials. Even in cases with slightly higher p-values, our schedule surpasses the baseline in the majority of trials. We underline the only two instances where our schedule does not outperform the baseline and a higher p-value is expected.
>
> Regarding the 95% Confidence Intervals (CIs), we note that they can be inferred from the reported standard deviations and the number of trials. To avoid redundancy and save space, we only report the standard deviations as our primary measure of dispersion, which is consistent with standard practice. These additions and $\textbf{discussions have been integrated}$ into the updated manuscript (Appendix F.2).
>
> $\textbf{Comment:}$  Evaluation breadth is limited: results focus on TinyImageNet and SSBs ... ... same protocol, making relative advantage ambiguous.
>
> $\textbf{Response:}$
>
> We also appreciate the reviewer's concern to include a broader range of comparisons. We agree that rigorous validation across modern architectures and competitive methods is crucial for establishing external validity. We have $\textbf{significantly expanded our experimental scope}$ based on your suggestions (details in the follow up responses). The new results are incorporated in the revised manuscript.
>
> 1. Architectural and Comparative Breadth:
>
>     $\textbf{ViT Architecture:}$ we implement a tiny ViT model on all three SSBs and combine the results in Table 8. We observe that the proposed schedules improve the performance for most of the cases (Appendix F.4).
>
>     $\textbf{SOTA Baseline:}$ We implemented and evaluated our schedules on the recent BackMix (TPAMI 2025) baseline. This demonstrates that our schedules offer an  additive benefit that enhances contemporary augmentation-based methods, such as the BackMix (Table 2).
>
>     $\textbf{Prototypical methods:}$ We included results on prototypical contrastive learning (Appendix F.7).
>
> 2. Analytical and Statistical Strength: to demonstrate the superiority of the proposed schedule, we have incorporated:
>
>     $\textbf{Scoring robustness:}$ a comprehensive study showing that the benefits of our schedules are maintained irrespective of the OSR scoring rule (e.g., Energy, ODIN, confidence, OpenAUC, Cosine- head) (Table 7).
>
>     $\textbf{Feature space analysis:}$ We introduced a new discussion to analyze the geometry of the learned space, demonstrating that our schedules result in superior intra/inter-class scatter and separability of unknowns to the known cluster (Table 4, Appendix F.1).
>
>     $\textbf{Statistical Significance:}$ We performed Wilcoxon signed-rank tests across all relevant metrics to rigorously validate the level of significance of the observed improvements (Table 6).
>
> These comprehensive additions significantly expand our evaluation and clarify the general applicability of our proposed temperature schedules.
>
> While the challenge of domain shift is critically important, we must clarify the scope of our work. Open Set Recognition (OSR) already requires the classifier to handle novel classes (category shift), which implies a distribution shift. However, integrating this with the additional complexity of Domain shift—where the appearance of the known classes also changes —makes the problem significantly more complex. This integrated challenge (Open Set Recognition under Domain shift) is beyond the scope of the present study.

---

> ### Author Response · Authors · 2025-11-21
>
> $\textbf{Comment:}$ Finally, the “no computational overhead” claim overlooks the practical cost of tuning τ-ranges and schedule shapes, as well as the UMAP visualizations.
>
> $\textbf{Response:}$
>
> We acknowledge the reviewer’s concern that tuning additional hyperparameters can be costly to optimize for the performance. Our claim is that users do not need to explicitly tune for both $\tau^+$ and $\tau^-$, which we derived from a systematic analysis.
>
> Our parameter sensitivity **analysis reveals the following derived heuristics works sufficiently well**: $\tau^- = \tau - \Delta$, $\tau^+ = \tau + \Delta$, where $\tau$ can chosen from the standard tuning for the constant baseline. This **removes the need to explicitly tune for both $\tau^+$ and $\tau^-$.**
>
> Moreover, regarding $P$, we found $P=200$ can be a robust default choice for our P-NegCosSch (Table - 8 ) that performs well across our benchmarks as demonstrated in most of our results. The main takeaway is that the users do not need to invest in a comprehensive tuning for P. **Users who wish to avoid this hyperparameter entirely can adopt our M-NegCosSch method as a competitive alternative** as shown in most of our results in Table 1 and 2.
>
> For details, please see the response above: “Justification for the derived heuristics”. We hope this discussion clarifies on the issue of cost for tuning.
>
>
> $\textbf{Comment:}$ While intuitive, these visualizations lack quantitative cluster diagnostics (e.g., intra/inter-class scatter, uniformity) to substantiate the representation-learning narrative.
>
> $\textbf{Response:}$
>
> We highly appreciate the reviewer’s suggestion to conduct a $\textbf{quantitative clustering diagnosis}$ to strengthen our representation learning narrative.
>
> Following this suggestion, we now compute the geometric properties in the representation space for our proposed schedules and the constant baselines: intra/inter-class scatter, and the average distance of unknown samples to the nearest prototype.
>
> Our analysis of the constant temperature baselines reveals the critical OSR trade-off discussed in Figure 1 and in Section 3. A $\textbf{lower temperature}$ achieves a $\textbf{higher intra-class scatter and a lower inter-class margin}$ (because of the instance specific features), indicating poor separation. However, this also achieves a $\textbf{higher average distance from unknown samples to the nearest prototype}$ of known classes, indicating a $\textbf{desirable property for better OSR}$.
> A higher temperature achieves a lower intra-class scatter and a higher inter-class margin, resulting in better closed set separation.
>
> Table: Representation space geometry analysis.
> |  | | CUB | | | Scars  | |
> | :--- | :--- | :--- | :--- | :--- | :---  | :---  |
> | Schedule  | intra-class scatter ($\downarrow$) | inter-class margin ($\uparrow$) | unknowns' distance to nearest prototype ($\uparrow$)  | intra-class scatter ($\downarrow$) | inter-class margin ($\uparrow$) | unknowns' distance to nearest prototype ($\uparrow$) |
> | Const. ($\tau = 0.5$)  | 0.1463  | 0.5503  | $\textbf{0.4172}$   | 0.0919  | 0.6587  | $\textbf{0.3869}$  |
> | Const. ($\tau = 2.0$)  | $\textbf{0.1252}$  | $\textbf{0.7805}$  | 0.3974   | $\textbf{0.0609}$  | $\textbf{0.9129}$  | 0.3797  |
> | Const. ($\tau = 1.0$)  | 0.1349  | 0.6586  | 0.41   | 0.0754  | 0.7923  | 0.3814  |
> | P-CosSch  | 0.1287  | 0.6832  | 0.4037   | 0.072  | 0.8127  | 0.3826  |
> | P-NegCosSch  | $\textbf{0.1318}$  | $\textbf{0.7741}$  | $\textbf{0.423}$    | $\textbf{0.068}$  | $\textbf{0.8951}$  | $\textbf{0.4002}$  |
>
>  The ideal OSR solution requires simultaneously achieving $\textbf{a lower intra-class scatter and a high inter-class margin and a higher average distance from unknowns to prototypes}$. The table demonstrates that our proposed schedule (NegCosSch) successfully achieves $\textbf{towards both the desirable properties}$. This structural superiority confirms that our schedule better utilizes the entire representation space.
>
> We believe this quantitative analysis significantly boosts our representation learning narrative and provides a crucial justification for the performance improvements observed. These results and discussions are included in the revised manuscript ($\textbf{Section F.1, Table 3)}$.

---

> ### Author Response · Authors · 2025-11-21
>
> $\textbf{Comment:}$ Question 1: Describe the dispersion ... ... currently tables list means only.
>
> $\textbf{Response:}$
>
> Please see the response above to the comment "Empirically, improvements are modest ... ... +1–3% AUROC/OSCR gains exceed run-to-run noise. "
>
> $\textbf{Comment:}$ Question 2: Run factorial sweeps ... ... performance to these settings.
>
> $\textbf{Response:}$
>
> Please see the response above for "Justification for the derived heuristics". Moreover, as per the suggestion, we present exactly the same results of Figure 4(a) in the Tables below.
>
> Table: AUROC for the constant baseline by varying $\tau$.
> | $\tau$ | 0.05 | 0.1 | 0.2 | 0.3 | 0.4 | 0.5 |
> | :--- | :--- | :--- | :--- | :--- | :--- | :--- |
> | AUROC | 82.02 | 82.34 | $\textbf{82.85}$ | 82.80 | 82.60 | 82.13 |
>
> Table: AUROC for NegCosSch by varying $\tau^-$ and $\tau^+$.
> | $\tau^-$ | 0.05 | 0.1 | 0.2 | 0.3 | 0.4 |
> | :--- | :--- | :--- | :--- | :--- | :--- |
> | $\tau^+$ |
> | 0.1 | 82.53 |
> | 0.2 | 82.66 | 82.63 |
> | 0.3 | 82.65 | 82.76 | $\textbf{83.03}$ |
> | 0.4 | 82.61 | $\textbf{83.09}$ | $\textbf{83.03}$ | $\textbf{82.93}$ |
> | 0.5 | 82.53 | 82.72 | 82.65 | 82.63 | 82.33 |
>
> We want to reiterate that $\textbf{these sensitivity analysis derive our proposed heuristics}$ that for a given constant temperature, we can construct the range in this manner $\tau^- = \tau - \Delta, \tau^+ = \tau + \Delta$ and this works reasonably well without the need to extensive tuning both $\tau^+$ and $\tau^-$. These results are presented in Figure 4 Appendix F.5. in the form of quadruplet-wise comparisons to justify our derived heuristics. We hope this clarifies on the derived heuristics.
>
> $\textbf{Comment:}$ Question 3: Justify why starting low-$\tau$ then increasing is fundamentally better for OSR.
>
> $\textbf{Response:}$
>
> Thank you for your comment. We discussed this in Section 4.3. We hope the discussion below clarifies further. We reiterate that our schedule is $\textbf{rooted in a gradient analysis}$ under constant temperature scaling, which $\textbf{informs the design of our modulation.}$
>
> These gradient results serve us to understand the foundational geometric properties for constant scaling factors at both extremes. We utilize this geometric understanding—the specific pattern of scatter and margin created by constant temperature in the two extreme ends—to propose a novel temperature modulation. Our schedule is precisely engineered to systematically traverse the trade-off at the both ends, better than any single fixed temperature.
>
> $\textbf{Starting with a lower temperature}$, the model provides priority to fewer neighbors, learning the coarse structure of representation space, resulting in a sharper decision boundary. The $\textbf{learned representation space is distributed}$. This coarse and distributed structure is essential to ensure that the $\textbf{unknown samples remain distant}$ from the known samples.
>
> $\textbf{As the temperature increases}$, the model prioritizes more neighbors and gradually pulls the positive samples to their own cluster. The model $\textbf{refines over the distributed representation space learned initially}$.  The $\textbf{unknown samples remain distributed}$ or distant to the known samples $\textbf{due to the initial coarse structure}$, thereby achieving benefits for both known and novel samples simultaneously.. The observed effects can also be realized from the computed geometric metrics of the learned representation space discussed above.

---

> ### Author Response · Authors · 2025-11-21
>
> $\textbf{Comment:}$ Question 4: You use max-logit for unknown detection ... .... Does NegCosSch still win irrespective of the scoring rule?
>
> $\textbf{Response:}$
>
> We appreciate the reviewer’s comment for this great suggestion to test the robustness of our scheduling across different scoring rules.
>
> $\textbf{Yes, NegCosSch wins irrespective of the scoring rule}$ for majority cases. To address this, we conduct evaluations comparing our proposed schedules against the baselines using multiple established OSR scoring rules: $\textbf{Energy}$ score (Liu et al. (2020)), $\textbf{ODIN}$ (Liang et al. (2017)), $\textbf{Cosine-Margin}$ (Deng et al. (2019)), max-Logit, and $\textbf{confidence score}$.
>
> The results, presented in the table below, confirm that our proposed schedules consistently outperform the baseline in the majority of cases, irrespective of the scoring rule used. This indicates that the performance $\textbf{gains are driven by the better representations}$ learned during training $\textbf{via our scheduling}$ strategy, rather than due to a specific metric. Therefore, the benefits of the schedules are transferable across different scoring rules. We append these results in $\textbf{Table 7, Appendix F.3}$ of the updated manuscript.
>
> Liu, W., Wang, X., Owens, J., & Li, Y. (2020). Energy-based out-of-distribution detection. NeurIPS.
>
> Liang, S., Li, Y., & Srikant, R. (2017). Enhancing the reliability of out-of-distribution image detection in neural networks.
>
> Deng, J., Guo, J., Xue, N., & Zafeiriou, S. (2019). Arcface: Additive angular margin loss for deep face recognition. In CVPR.
>
> Table: Evaluation of robustness to different open-set scoring rules.
> | | scoring rule | Max-logit | Max-prob. | Energy | ODIN | cosine head |
> | :--- | :--- | :--- | :--- | :--- | :--- | :--- |
> |Loss | schedule | CUB |
> |CE | Constant | 83.55 / 74.98 | 83.81 / 78.05 | 83.11 / 74.19 | 83.17 / 73.26 | 85.33 / 76.26 |
> | | M-NegCoSch (ours) | $\textbf{86.79 / 78.08}$ | $\textbf{86.43 / 80}$ | $\textbf{87.21 / 78.18}$ | $\textbf{87.6 / 77.38}$ | $\textbf{88.05 / 78.86}$ |
> | | P-NegCoSch (ours) | $\textbf{86.85 / 77.6}$ | $\textbf{86 / 79.76}$ | $\textbf{86.91 / 77.55}$ | $\textbf{87.04 / 76.71}$ | $\textbf{88.09 / 78.47}$ |
> |SupCon | Constant | 86.94 / 73.95 | 86.48 / 76.35 | 86.05 / 72.57 | 86.75 / 73.2 | 86.57 / 76.24 |
> | | M-NegCoSch (ours) | $\textbf{88.14 / 75.81}$ | $\textbf{87.82 / 77.27}$ | $\textbf{87.56 / 74.35}$ | $\textbf{88.21 / 74.81}$ | $\textbf{88.66 / 78.34}$ |
> | | P-NegCoSch (ours) | $\textbf{87.5 / 74.95}$ | $\textbf{87.3 / 76.78}$ | $\textbf{87.05 / 73.75}$ | $\textbf{87.69 / 74.31}$ | $\textbf{87.9 / 77.69}$ |
>  | | | $\textbf{Aircraft}$ |
> | CE | Constant | 90.35 / 81.48 | 84.96 / 81.53 | 87.63 / 81.05 | 87.3 / 79.64 | 85.05 / 80 |
> | | M-NegCoSch (ours) | $\textbf{91.15 / 83.23}$ | $\textbf{88.34 / 82.18}$ | $\textbf{92.19 / 84.67}$ | $\textbf{92.64 / 84.5}$ | $\textbf{89.26 / 83.42}$ |
> | | P-NegCoSch (ours) | $\textbf{91.41 / 83.15}$ | $\textbf{85.24}$ / 79.98 | $\textbf{89.1 / 82.11}$ | $\textbf{89.32 / 81.13}$ | $\textbf{86.56 / 81.12}$ |
> |SupCon | Constant | 88.78 / 81.79 | 84.59 / 80.66 | 85.96 / 81.1 | 85.84 / 80.5 | 84.1 / 80.79 |
> | | M-NegCoSch (ours) | $\textbf{90.45 / 82.49}$ | $\textbf{90.55 / 82.3}$ | $\textbf{90.84 / 82.46}$ | $\textbf{91.14 / 82.37}$ | $\textbf{91.33 / 81.93}$ |
> | | P-NegCoSch (ours) | $\textbf{89.27 / 81.97}$ | $\textbf{86.29 / 81.27}$ | $\textbf{87.11 / 81.44}$ | $\textbf{87.43 / 80.97}$ | $\textbf{85.48 / 81.55}$ |
> | | | $\textbf{SCars}$ |
> |CE | Constant | 94.03 / 84.82 | 93.82 / 84.99 | 93.04 / 84.2 | 92.89 / 83.93 | 93.49 / 84.16 |
> | | M-NegCoSch (ours) | $\textbf{95.18 / 86.26}$ | $\textbf{94.84}$ / 83.58 | $\textbf{95.14 / 84.86}$ | $\textbf{95.15 / 84.8}$ | $\textbf{94.78 / 84.41}$ |
> | | P-NegCoSch (ours) | $\textbf{95.03 / 86.05}$ | $\textbf{94.59 / 85.68}$ | $\textbf{94.55 / 86.18}$ | $\textbf{94.49 / 86.11}$ | $\textbf{94.28 / 85.4}$ |
> | SupCon | Constant | 92.99 / 82.8 | 93.71 / 83.56 | 92.41 / 82.42 | 93.07 / 82.85 | 94.28 / 83.57 |
> | | M-NegCoSch (ours) | $\textbf{93.57}$ / 82.76 | $\textbf{94.34}$ / 83.1 | $\textbf{93.39}$ / 82.26 | $\textbf{93.88}$ / 82.56 | $\textbf{95.1}$ / 83.25 |
> | | P-NegCoSch (ours) | $\textbf{93.32 / 83.16}$ | $\textbf{93.73}$ / 83.39 | $\textbf{92.72 / 82.56}$ | $\textbf{93.23 / 82.9}$ | $\textbf{94.13}$ / 82.84 |
> | | | $\textbf{TinyImageNet}$ |
> | CE | Constant | 78.6 | 79.59 | 80.95 | 78.39 | 80.11 |
> | | M-NegCoSch (ours) | $\textbf{79.21}$ | 78.11 | $\textbf{81.19}$ | 78.09 | $\textbf{80.44}$ |
> | | P-NegCoSch (ours) | $\textbf{79.05}$ | 77.65 | $\textbf{81.09}$ | 78.08 | $\textbf{80.37}$ |
> | SupCon | Constant | 82.87 | 83.05 | 82.93 | 80.12 | 81.16 |
> | | M-NegCoSch (ours) | $\textbf{83.21}$ | 82.99 | $\textbf{83.11}$ | $\textbf{80.2}$ | $\textbf{81.49}$ |
> | | P-NegCoSch (ours) | $\textbf{83.09}$ | 82.9 | $\textbf{83.05}$ |$\textbf{80.19}$ |$\textbf{81.42}$ |

---

> > ### Author Response · Authors · 2025-11-21
> >
> > $\textbf{Comment:}$ Question 5: Breadth and strength of baselines Reproduce  ... .... clarifies whether scheduling alone is competitive at today’s frontier.
> >
> >
> > $\textbf{Response:}$
> >
> > We thank the reviewer for this excellent suggestion to evaluate our method's complementarity with stronger, contemporary OSR baselines. This helps us clarify that our scheduling mechanism is an independent contribution that can enhance existing methods.
> >
> > As requested, we have $\textbf{integrated our proposed schedules}$ (P-NegCosSch and M-NegCosSch) with $\textbf{BackMix by Wang et al.}$ (2025) (from their official implementation) and evaluated it on TinyImageNet and the SSBs. The BackMix method is an augmentation-based scheme designed to improve OSR by reducing learned spurious correlations between an image's foreground and its background.
> >
> > First, to validate our setup, we reproduced the BackMix results on TinyImageNet and achieved 81.23 AUROC, which is consistent with the 80.4 AUROC reported in the original paper.
> >
> > The results of applying our schedules on top of the BackMix framework are presented in the table below. We observe that our $\textbf{proposed schedules boost the performance of the BackMix}$ baseline across all three metrics in the majority of cases. This is a significant finding, as it shows our method's benefits generalize beyond simple baselines to SOTA augmentation-based frameworks.
> >
> > Table: Performance on BackMix both including and without the proposed schedules. Results shown on Easy/Hard splits.
> > | | | CUB | | | Aircraft | |
> > | :--- | :--- | :--- | :--- | :--- | :--- | :--- |
> > | Method | Acc | AUROC | OSCR | Acc | AUROC | OSCR |
> > | Constant | 82.12 | 82.39 / 72.99 | 67.94 / 60.32 | 90.53 | 92.41 / 82.47 | 83.75 / 75.09 |
> > | M-NegCoSch (ours) | $\textbf{82.84}$ | $\textbf{83.97 / 74.66}$ | $\textbf{69.71 / 62.1}$ |$\textbf{ 91.37}$ | 92.2 / 84.43 | $\textbf{84.19 / 77.36}$ |
> > | P-NegCoSch (ours) | $\textbf{83.98}$ | $\textbf{84.54 / 74.13}$ |$\textbf{ 71.23 / 62.66}$ | $\textbf{91.49}$ | 92.3 / 83.68 | $\textbf{84.35 / 76.72}$ |
> > | | | Scars | | | TinyImageNet | |
> > | Constant | 96.81 | 93.23 / 84.39 | 90.33 / 81.82 | 82.32 | 81.23 | 67.1 |
> > | M-NegCoSch (ours) | $\textbf{97.52}$ | $\textbf{94.86 / 86.48}$ |$\textbf{ 92.56 / 84.48}$ |$\textbf{ 82.6}$ | $\textbf{81.62}$ | $\textbf{67.37}$ |
> > | P-NegCoSch (ours) | $\textbf{97.37}$ | $\textbf{94.76 / 86.04}$ | $\textbf{92.31 / 83.89}$ | $\textbf{82.5}$ | $\textbf{81.72}$ | $\textbf{67.42}$ |
> >
> > We have added these results in the revised manuscript ($\textbf{Section 5.2, Table 2}$).
> >
> > Regarding iCausalOSR (PR 2024a), despite our efforts, we were unable to locate a public implementation to conduct a fair comparison within the limited rebuttal period. If we figure out a way to obtain the implementation, we will perform experiments on this method and append the results before the camera ready version. The paper reported OSR performance only on the regular OSR benchmarks. If we compare the reported open set AUROC on the TinyImageNet ($\textbf{0.828}$) against our $\textbf{P-NegCoshSch on cross-entropy loss with label smoothing (83.02)}$, we can infer that $\textbf{a simple inclusion of our proposed schedule can outperform iCausalOSR}$ method.
> >
> > We thank the reviewer for this suggestion, as these new results significantly strengthen the paper by demonstrating the robustness and general applicability of our schedules.
> >
> > Wang, Y., Mu, J., Huang, H., Wang, Q., Zhu, P., & Hu, Q. (2025). BackMix: Regularizing Open Set Recognition by Removing Underlying Fore-Background Priors. IEEE Transactions on Pattern Analysis and Machine Intelligence.
> >
> > Yang, F., Li, B., & Han, J. (2024). iCausalOSR: invertible causal disentanglement for open-set recognition. Pattern Recognition, 149, 110243.

---

> ### Author Response · Authors · 2025-11-21
>
> $\textbf{Comment}:$ Question 6: You claim plug-and-play for “any OSR loss”. Test at least one additional family—e.g., OpenAUC-style objectives or prototype-based methods—and report whether schedule benefits persist without retuning.
>
> $\textbf{Response:}$
>
> We thank the reviewer for their concern regarding the performance of our schedules with other loss families.
>
> As per suggestion, we implement our proposed schedules on top of the prototypical contrastive learning on TinyImageNet and Aircraft to investigate the effects. The results are reported in the Table below. We observe **performance improvements on prototypical contrastive learning with our schedule**. We append the results for prototypical network and the discussions in the revised manuscript **(Table 11, Appendix F.7).**
>
> Regarding implementing the OpenAUC loss, please note that the **implementing OpenAUC loss to train a model requires synthetic unknown representations, which is obtained from the manifold mixup** procedure. Manifold mix-up can **increase the computational complexity** significantly while training when implemented with larger benchmarks and with model architecture of larger size such as the ResNet50 architecture.
>
> | | | TinyImageNet | | | Aircraft | |
> | :--- | :--- | :--- | :--- | :--- | :--- | :--- |
> | Schedule | Acc | AUROC | OSCR | Acc | AUROC | OSCR |
> | Const.  | 85.36  | 82.79  | 70.53  |  86.29  |  84.32 / 78.46  |  72.28 / 68.26|
> | NegCosSch  | $\textbf{85.72}$  | $\textbf{83.04}$  | $\textbf{71.12}$  |  $\textbf{87.03}$  |  $\textbf{84.93 / 79.01}$  |  $\textbf{73.29 / 66.33}$ |

---

> > ### Author Response · Authors · 2025-12-04
> > **Summary of Comprehensive Responses to Reviewer TEAQ**
> >
> > We are grateful for the reviewer's detailed comments and have executed extensive new experiments and analyses to address all concerns regarding novelty, empirical breadth, and statistical significance.
> >
> > The perceived lack of novelty was addressed by clarifying that our work provides a novel extension of known scheduling ideas to the supervised Open Set Recognition (OSR) scenario across multiple loss types. The true novelty lies in the geometric rationale of our Negative Cosine Scheduling (NegCosSch): it is engineered based on the gradient analysis to traverse the trade-off inherent in fixed temperature, ensuring that the model initially learns a distributed representation space where unknowns remain distant (low $\tau$) and refines tight clusters for known classes (high $\tau$).
> >
> > The concern regarding the heuristic choice of parameters $(\tau^+, \tau^-)$ is addressed by clarifying that the relationship of our heuristic is structurally informed and derived from  sensitivity analysis, not by arbitrary choice. This strategy ensures that our schedule is centered around the optimal constant temperature $\tau$ (from tuning), thereby maintaining the structural stability of the learned representation space and preventing collapse caused by excessive width $\Delta$. Furthermore, we clarified that the period $P$ is not highly sensitive, which simply needs to be within a functional range for sufficient optimization exploration (Section 4.3). This eliminates the need for extensive tuning.
> >
> > We substantiated our feature learning narrative with a new quantitative clustering diagnosis, confirming the structural superiority of the learned feature space and establishing ties between the proposed schedules and the measurable open set properties. Moreover, we rigorously validated our improvements by performing statistical significance tests, which confirmed the statistical significance of our gains.
> >
> > Finally, to broaden the empirical validation, we successfully integrated our schedules with the state-of-the-art augmentation baseline, BackMix (TPAMI 2025) and prototypical contrastive learning, confirming our schedules provide an orthogonal, additive performance boost. We also demonstrated architectural robustness by testing on ViT models and confirmed scoring robustness by showing consistent wins irrespective of the OSR scoring rule used (e.g., Energy score, ODIN, cosine heads). While a public implementation for iCausalOSR was unavailable, we noted that the performance of our simpler P-NegCosSch on Cross-Entropy loss on TinyImageNet exceeds the reported AUROC for iCausalOSR.

---

### Official Review · Reviewer_u1CY · 2025-10-27

**Soundness:** 2
**Presentation:** 3
**Contribution:** 3
**Rating:** 4
**Confidence:** 4

**Summary:**

This paper explores the influence of the temperature parameter in loss of OSR and introduces a novel temperature scheduling strategy NegCosSch. It varies the temperature during training, enabling the model to balance instance-level and semantic-level features in representation learning without additional computational cost. Extensive experiments on the CIFAR and SSB benchmarks demonstrate consistent performance improvements.

**Strengths:**

+ This paper is well-written and easy to follow. I favor studies that derive methodological innovations from experimental observations.

+ The details provided in the Appendix are clear and crucial, enhancing the paper’s overall logical flow and strengthening the credibility of some experiments.

+ The empirical evaluation is extensive, covering multiple benchmarks and ablation studies.

**Weaknesses:**

+ The choice of $P$, $(\tau ^{+}, \tau ^{-})$ in the method is mainly heuristic. A more thorough theoretical analysis would improve the quality of the paper.

+ Some of the authors’ statements lack experimental or theoretical support. For example, the discussion regarding empty regions in the representation space, the significant computational overhead in existing methods, and the statement *'Therefore, the methods that demonstrate improvement on smaller datasets…'*(lines 096–098).

+ From my perspective, Sec. 4.2 constitutes one of the key sections for motivation. However, it remains unclear how the subsequent discussion is derived from Eq. (4), further clarification is needed.

+ The figures (especially Fig.1 and Fig. 2) are not positioned near the relevant text, which hinders readability.

+ Typos issue: line 135 *Section, 3*

**Questions:**

My questions are mainly focused on the experimental section:

+ Why is label smoothing introduced in Sec. 5.2? If it is only because “As label smoothing (LS) has shown performance improvements,” the purpose is not clearly justified.

+ Why are the results of the Vision Transformer model on TinyImageNet presented? If this section is necessary, shouldn’t results on larger-scale datasets be shown?

+ I would like to see the results about the metric OpenAUC [1].

+ The paper does not mention a validation set. How were the hyperparameters determined? Was the test set used for tuning, and are the final results reported based on that? I am mainly concerned.

Minor point:

+ Do some of the terms refer to the same concept (semantic-level, class-level, group-wise)? If so, please unify them.

[1] Wang, Zitai, et al. "Openauc: Towards auc-oriented open-set recognition." Advances in Neural Information Processing Systems 35 (2022): 25033-25045.

If authors address the Weaknesses and Questions, I will increase my score.

---

> ### Author Response · Authors · 2025-11-21
> **Response of comments by Reviewer u1CY**
>
> $\textbf{Overall Response:}$
> We thank the reviewer for acknowledging the strengths of our work, appreciating the extensive empirical studies both in the main texts and the appendices and complimenting the write-up. We appreciate the concerns expressed by the reviewer, which certainly requires further clarifications. We hope that the following discussions will shed more clarity about our work and in re-evaluating the quality of the work. We believe that addressing the concerns will improve the quality of the paper. We have uploaded the revised manuscript.
>
> $\textbf{Comment:}$ Weakness: The choice of ... ...  improve the quality of the paper.
>
> $\textbf{Response:}$
>
> We thank the reviewer for this comment, which has helped us clarify an important aspect of our proposed method. We confirm that $\textbf{we derived our parameter heuristics from a systematic analysis}$ and its purpose is that users do not need to explicitly tune for both $\tau^+$ and $\tau^-$. To support this, we show that the proposed heuristics, such as $\tau^- = \tau - \Delta$, $\tau^+ = \tau + \Delta$, are not mere heuristics but they are guided by parameter sensitivity analysis.
>
> For this purpose, we reiterate and highlight the experiments presented in $\textbf{Figure 4}$ Appendix F.5. For example, in Figure 4(a), we varied the range $(\tau^+, \tau^-)$ for our proposed schedules and $\tau$ for constant temperature baseline from {0.025, 0.05, 0.1, 0.2, 0.3, 0.4, 0.5} while implementing SupCon on TinyImageNet and performed the sensitivity analysis. We present the results in the form of quadruplet-wise comparisons in Figure 4 among the obtained performances. In these quadruplet comparisons, we compare our NegCosSch($\tau^+$, $\tau^-$) with CosSch($\tau^+$, $\tau^-$)  and Const($\tau$) when the constant temperature is either set to the mid point of the range or to lower value. We observe that our proposed TS yields a better performance than CosSch and constant temperatures for most quadruplet comparisons.
>
> $\textbf{These results derive our proposed heuristics}$ that for a given temperature, we can choose $\tau^- = \tau - \Delta, \tau^+ = \tau + \Delta$ and this works reasonably well without the need to extensive tuning both $\tau^+$ and $\tau^-$. The proposed heuristic is a principled simplification that is derived directly from the observed performance trends in Figure 4, which showed that the relationship between $\tau^+$ and $\tau^-$ is consistently good.
>
> $\textbf{Regarding the choice of}$ $P$: Based on this observation (Table 9), we found P=200 can be a robust default choice for our P-NegCosSch that performs well across our benchmarks as we demonstrated in most of our results. The main takeaway is that the users do not need to invest in a comprehensive tuning for P; a reasonable value is sufficient. Users who wish to avoid this hyperparameter entirely can adopt our $\textbf{M-NegCosSch method as a competitive alternative}$ as shown in most of our results in Table 1 and 2 that does not require setting a period P, thus incurring no tuning expense for this parameter.
> This allows users to rely on the derived relationship instead of incurring the expense of a full parameter sweep for every new dataset.
>
> We iterate that the heuristic relationship between parameters, which were actually derived from the sensitivity analysis. However, we understand that it was not well mentioned in the previous submission. We have added these details for further clarification (lines 305 - 313 in Section 4.3) in the revised manuscript.
>
> $\textbf{Theoretical analysis:}$
> The authors appreciate the concern of the reviewer regarding the lack of theoretical analysis.
>
> We want to reiterate that we derived the effects of temperature scaling on supervised losses with gradient analysis (discussed in Section 3). These gradient results align with the existing literature and these results are not only built just on intuition but also based on theoretical analysis for the self-supervised case. For example, the work in Wang and Liu [2021] calculated gradients of losses with respect to pairwise similarity to see the effect of a fixed temperature scaling. Moreover, Kukleva et al. [2023] analyzed the gradient w.r.t. the representation of a negative sample.
>
> Essentially, we utilize these observations of constant scaling's limitations in both extremes to $\textbf{propose a novel and clever temperature modulation}$ that replaces the fixed $\tau$ with an epoch-dependent $\tau(e)$.
>
> To summarize the benefit: Starting with a lower temperature, the model learns a $\textbf{distributed representation space}$ where open set samples have higher distance to the known samples. Gradually increasing the temperature then $\textbf{pulls the known samples towards their own cluster, keeping the unknowns distant}$, thereby achieving benefits for both known and novel samples simultaneously.
>
> Also, please see the response to weakness 3.

---

> ### Author Response · Authors · 2025-11-21
>
> **Comment:** Weakness: Some of the authors’ statements lack experimental or theoretical support .... ....
>
> **Response:**
>
> We appreciate the reviewer’s concern regarding the lack of support for some of the statements. We agree that these statements are not intuitive and require further support. We aim to address them separately with reasonable justification.
>
> 1. “Forcing the creation of empty spaces does not result in an improved OSR”.
>
> Forcing large empty spaces such as through learning additional placeholders and manifold mix-up by Zhou et al (2021) to push the decision boundary of known classes much tighter. However, this fails to address the inherent semantic closeness of hard unknown samples. Since the **tougher unknown samples** share features with known classes, they naturally **embed close to the known cluster** boundaries, yielding high similarity. Methods relying solely on **forceful space creation fail to resolve this semantic proximity**, making them ineffective against the most critical failures in OSR.
>
> Furthermore, forcing the creation of large empty spaces through heavy regularization can encourage **under-utilization of the representation space**. While this may initially reduce intra-class scatter and increase inter-class separability (benefiting closed-set separation), it can also **encourage unknown samples to not fall into the empty space** (as the model is trained to extract representations to not utilize certain areas in the space), leading them to find spurious similarity with one of the known classes. A similar effect is also observed when scaling with higher temperature as discussed in section 3.
>
> 2. “Significant computational overhead of the existing methods” and
>
> 3. “Therefore, the methods that demonstrate improvement on smaller datasets…”
>
> In both the statements, we primarily comment on the methods that require data generation, mix- up and training secondary models. Methods training VAEs or GANs for data generation on top of the closed set task significantly adds to the training computational requirement. Moreover, training the generative models on complex real-life data is a cumbersome task itself, which is not practical while solving a problem whose original goal is a classification task. Moreover, in some cases, the dataset may not have enough samples per class to train such secondary models. Rather, a simple add on is desirable, such as our proposed schedule. Moreover, Mix-up based methods, such as manifold mix-up can increase the amount of compute during back-propagation as the interpolation of samples happens in a hidden layer changing the standard forward-backward pass.
>
> We understand that these statements require further clarifications. We **added the details briefly** in the revised manuscript **(lines 50-53 in the Introduction Section and 501 -505 in the related works)**.
>
> Da-Wei Zhou, Han-Jia Ye, and De-Chuan Zhan. Learning placeholders for open-set recognition. In Proceedings of the IEEE/CVF conference on computer vision and pattern recognition, pp. 4401–4410, 2021.

---

> ### Author Response · Authors · 2025-11-21
>
> **Comment:** Weakness: From my perspective, Sec. 4.2 constitutes one of the key sections for motivation. However, it remains unclear how the subsequent discussion is derived from Eq. (4), further clarification is needed.
>
> **Response:**
>
> We thank the reviewer for identifying this crucial gap in the motivation between Sections 4.2 and 4.3 and agree that the transition requires clearer justification. Here, first we try to clarify further on the semantic features vs instance features and then, we aim to provide further analysis based on the geometric properties of the learned representation space.
>
> The confusion stems from the limitations inherent in constant temperature scaling, which we have now explicitly defined in the revised manuscript. In section 4.2, we discussed the extreme ends of having either too class-specific or instance-specific representation spaces. **To reiterate**, a lower temperature forces the model toward learning instance-specific features. The model cannot generalize. A higher temperature forces the model toward class-specific (semantic) features. It encourages smoother decision boundaries and better generalization. However, the unknown representations also get close to the known clusters, resulting in high open set risk.
>
> **Motivation.** Every **constant temperature** is locked into a **compromise between these two extremes**. However, for **effective OSR**, a model needs to capture a **delicate combination of both the desirable properties**– good class-specific representations while having room for instance-level discriminating power within the class. The way to minimize the total OSR risk is to **dynamically leverage the benefits of both regimes**.
>
> Therefore, we introduce the cosine schedule in Section4.3 as a logical solution: it is a smooth function designed to **traverse the spectrum from instance-specific representation learning to class-specific generalization.** This systematic, smooth adjustment maximizes the model's ability to maintain tight clusters for known classes while still generalizing effectively for the unknown samples.
>
> We understand that the clarifications were not clear enough. We have **revised section 4.2**.
>
> **Further analysis.** To further enhance the representation learning narrative and to demonstrate the observed effects, we conduct a **quantitative clustering diagnosis.**
>
> We compute the geometric properties in the representation space for our proposed schedules and the constant baselines: intra/inter-class scatter, and the average distance from unknown samples to the nearest class prototype. Our Schedules can simultaneously achieve **a lower intra-class scatter, a high inter-class margin and a higher average distance from unknowns to prototypes.**
>
> Our analysis of the constant temperature baselines reveals the critical OSR trade-off discussed in Figure 1 and in Section 3. A **lower temperature** achieves a **higher intra-class scatter and a lower inter-class margin** (because of the instance specific features), indicating poor separation. However, this also achieves a **higher average distance from unknown samples to the nearest prototype, desirable for better OSR**. A higher temperature achieves a lower intra-class scatter and a higher inter-class margin, resulting in better closed set separation.
>
> Table: Representation space geometry analysis.
> |  | | CUB | | | Scars  | |
> | :--- | :--- | :--- | :--- | :--- | :---  | :---  |
> | Schedule  | intra-class scatter ($\downarrow$) | inter-class margin ($\uparrow$) | unknowns' distance to nearest prototype ($\uparrow$)  | intra-class scatter ($\downarrow$) | inter-class margin ($\uparrow$) | unknowns' distance to nearest prototype ($\uparrow$) |
> | Const. ($\tau = 0.5$)  | 0.1463  | 0.5503  | $\textbf{0.4172}$   | 0.0919  | 0.6587  | $\textbf{0.3869}$  |
> | Const. ($\tau = 2.0$)  | $\textbf{0.1252}$  | $\textbf{0.7805}$  | 0.3974   | $\textbf{0.0609}$  | $\textbf{0.9129}$  | 0.3797  |
> | Const. ($\tau = 1.0$)  | 0.1349  | 0.6586  | 0.41   | 0.0754  | 0.7923  | 0.3814  |
> | P-CosSch  | 0.1287  | 0.6832  | 0.4037   | 0.072  | 0.8127  | 0.3826  |
> | P-NegCosSch  | $\textbf{0.1318}$  | $\textbf{0.7741}$  | $\textbf{0.423}$    | $\textbf{0.068}$  | $\textbf{0.8951}$  | $\textbf{0.4002}$  |
>
> The ideal OSR solution requires simultaneously achieving a lower intra-class scatter, a high inter-class margin and a higher average distance from unknowns to prototypes. The table demonstrates that our proposed schedule (NegCosSch) successfully achieves **towards both the desirable properties**. This structural superiority confirms that our schedule better utilizes the entire representation space.
>
> We believe this quantitative analysis significantly boosts our representation learning narrative and provides a crucial justification for the performance improvements observed. The results are included in the revised manuscript **(Section F.1, Table 3)**.

---

> ### Author Response · Authors · 2025-11-21
>
> $\textbf{Comment:}$  Weakness: The figures (especially Fig.1 and Fig. 2) are not positioned near the relevant text, which hinders readability.
>
> $\textbf{Response:}$
>
> We acknowledge the reviewer’s comment for the figure placements aimed at improving readability.
>
> We have attempted to move Figure 2 near the location of its first mention but relocating it causes a severe formatting issue: it pushes Equations (5) and (6) outside the page margins, which compromises the readability. Given this constraint, we prefer the current placement of Figure 2. However, we will try again.
>
> Regarding Figure 1, we respectfully prefer to retain its current placement near the beginning of the manuscript. In our work, $\textbf{Figure 1 serves as a high-level visual summary}$ to immediately establish the contribution and contrast it with the standard constant baseline. Therefore, we strategically place this figure here, ensuring that the reader can quickly grasp our claim.
>
> $\textbf{Comment:}$ Typos issue: line 135 Section, 3
>
> $\textbf{Response:}$
> Thank you for pointing it out. We fixed the typo in the updated manuscript.
>
>
> $\textbf{Comment:}$  Question: Why is label smoothing introduced in Sec. 5.2? If it is only because “As label smoothing (LS) has shown performance improvements,” the purpose is not clearly justified.
>
> $\textbf{Response:}$
>
> We thank the reviewer for this insightful question. We introduced label smoothing (LS) in Sec. 5.2 to $\textbf{investigate the complementarity of our proposed schedules with label smoothing}$.
>
> Since Vaze et al. (2022) demonstrated that LS can significantly improve OSR performance, we considered it a relevant baseline modification. Our core goal was to investigate whether our proposed TS offers an orthogonal benefit—meaning its improvements are gained irrespective of Label Smoothing.
>
> Our experiments show that our proposed TS consistently improves performance both including and without Label Smoothing. We found that TS provides a performance boost even in cases where LS alone failed to improve upon the baseline, such as for the Stanford cars -‘Hard’ split and Aircraft with CE loss. This demonstrates that our $\textbf{temperature schedules can boost the performance independently of label smoothing}$.
>
> We understand that the motivation for introducing label smoothing was not clear enough. We $\textbf{added these clarifications}$ in the updated manuscript (lines 429-431, Section 5.2).
>
> $\textbf{Comment:}$  Question: Why are the results of the Vision Transformer model on TinyImageNet presented? If this section is necessary, shouldn’t results on larger-scale datasets be shown?
>
> $\textbf{Response:}$
>
> We appreciate the reviewer’s concern regarding the performance of Vision Transformer models on the larger benchmarks. We want to respectfully reiterate that in Appendix F.4, the vision transformer (ViT) model was $\textbf{not tested on the TinyImagenet benchmark}$; rather, the reported results are on California UCSD Birds (CUB) datasets, which is one of the SSBs. We may understand the cause of confusion here as the ViT model that was implemented is a $\textbf{tiny architecture (hence tiny ViT)}$ by Wu et al. 2022.
> As your suggestion, we $\textbf{implement the tiny ViT model on the other 2 SSBs}$ and combine the results in the following table. We observe that the $\textbf{proposed schedules improve} the performance for most of the cases in these benchmarks. These results along with the other results with VGG and ResNet-based architectures confirm the applicability of our proposed schedules across a range of model architectures.
> We append these results in the updated manuscript. Please see Table 7 in Appendix F.4 in the updated manuscript.
>
> Kan Wu, Jinnian Zhang, Houwen Peng, Mengchen Liu, Bin Xiao, Jianlong Fu, and Lu Yuan. Tinyvit: Fast pretraining distillation for small vision transformers. In European conference on computer vision, pp. 68–85. Springer, 2022
>
> Table: Performance on SSBs with a tiny ViT architecture
> | | | CUB | | | Aircraft | | | Scars | |
> | :--- | :--- | :--- | :--- | :--- | :--- | :--- | :--- | :--- | :--- |
> | Method | Acc | AUROC | OSCR | Acc | AUROC | OSCR | Acc | AUROC | OSCR |
> | Const. (baseline) | 90.83 | 91.41/79.88 |82.99/72.59 | 88.26 | 87.75/76.31 | 77.82/68.08 | 95.62 | 92.5/83.67 | 88.54/80.13 |
> | M-NegCosSch (ours) | $\textbf{91.07}$ | $\textbf{92.06/80.67}$ | $\textbf{83.79/73.51}$ | $\textbf{88.51}$ | $\textbf{89/78.78}$ | $\textbf{79.12/70.41}$ | $\textbf{95.8}$ | $\textbf{93.1}$/83 | $\textbf{89.28/79.61}$ |
> | P-NegCosSch (ours) | $\textbf{91.2}$ | $\textbf{92.3/80.04}$ | $\textbf{84.1/73.03}$ | $\textbf{88.92}$ | $\textbf{88.65/79.68}$ | $\textbf{79.12/71.46}$ | $\textbf{ 96.03}$ | $\textbf{93.08}$/83.08 | $\textbf{89.47/79.9}$ |

---

> ### Author Response · Authors · 2025-11-21
>
> $\textbf{Comment:}$
> I would like to see the results about the metric OpenAUC [1].
>
> $\textbf{Response:}$
>
> We thank the reviewers for their concern on the performance on the OpenAUC metric. Following the public implementation of OpenAUC by Wang et al. (2022), we implemented the metric on CE and SupCon baselines and the results are presented in the table below. By comparing the OpenAUC scores in Table below and the OSCR scores in Table 1, we find that the numerical values of these scores are very close to each other. Moreover we find that our $\textbf{proposed schedules outperform the constant baseline on this metric}$. We include these discussions in the main text and $\textbf{append these results in Table 6}$ in the appendix F.2 in the updated manuscript.
>
> Table: OpenAUC scores on the SSBs and TinyImageNet
> | Loss | Schedule | CUB | Aircraft | Scars | TinyImageNet |
> | :--- | :--- | :--- | :--- | :--- | :--- |
> | CE | Constant | 70.5 / 63.35 | 82.05 / 74.26 | 91.05 / 82.2 | 69.23 |
> | | M-NegCoSch (ours)	 | $\textbf{74.71 / 67.31}$ | $\textbf{83/ 76.01}$ | $\textbf{92.57 / 83.96}$	 | $\textbf{69.85}$ |
> | | P-NegCoSch (ours)	 | $\textbf{74.9 / 67.01}$ | $\textbf{83.44 / 76.15}$ | $\textbf{92.5 / 83.82}$	 | $\textbf{69.92}$ |
> |SupCon | Constant | 72.42 / 61.67 | 80.52 / 74.39 | 89.93 / 80.13 | 70.63 |
> | | M-NegCoSch (ours)	 | $\textbf{75.1 / 64.73}$ | $\textbf{82.57 / 75.52}$ | $\textbf{90.67 / 80.27}$	 | $\textbf{70.9}$ |
> | | P-NegCoSch (ours) | $\textbf{73.55 / 63.14}$ | $\textbf{80.97 / 74.56}$ | $\textbf{90.32 / 80.54}$	 | $\textbf{70.72}$ |
>
> $\textbf{Comment:}$
> The paper does not mention a validation set. How were the hyperparameters determined? Was the test set used for tuning, and are the final results reported based on that? I am mainly concerned.
>
> $\textbf{Response:}$
>
> We thank the reviewer for raising this excellent critical concern regarding our hyperparameter tuning protocol. We understand that this point was not fully clarified and requires further discussion.
>
> Although not explicitly detailed in the original submission, tuning of parameters ($\tau^+, \tau^-$ and $P$) for the SupCon loss follows a rigorous procedure. While tuning for TinyImageNet, we split the training data into a training portion and a validation set. The optimal parameters are selected based $\textbf{on the validation set}$. Only these tuned parameters are then used to report the final test results. For SSBs, we tuned $\textbf{only for constant temperature}$ to optimize the constant baseline. For our schedules, we do not perform extensive tuning, rather we $\textbf{apply the derived heuristics}$ from our analysis mentioned in Section 4.3, which works really well.
>
> For the CE (Cross-Entropy) loss, as the constant temperature baseline uses $\tau=1$, which is the standard and most common choice in literature. For our proposed schedules, we set $\tau^- = 0.5$ and $\tau^+ = 2.0$.
>
> We agree that some of these crucial steps regarding the hyperparameter tuning require greater clarity. We have $\textbf{revised the Training Details subsection}$ in the revised manuscript (lines 357-362 in page 7).
>
> $\textbf{Comment:}$
> Do some of the terms refer to the same concept (semantic-level, class-level, group-wise)? If so, please unify them.
>
> $\textbf{Response:}$
> Yes, these terms refer to the same concept. Thank you for pointing out this issue. All these terms are unified as “class-specific features” in the updated manuscript.

---

> > ### Comment · Reviewer_u1CY · 2025-11-26
> >
> > I appreciate the authors’ efforts. While some of my concerns have been addressed, I still expect the authors to provide further clarification on the following points.
> >
> > 1. Although Sec. 3 outlines the relationship between $\tau$ and its role in different loss functions, the paper does not offer any theoretical justification for how the proposed method determines the specific value of $\tau$(The choice appears to rely solely on heuristic empirical observations). Furthermore, the paper provides no theoretical discussion regarding $P$.
> >
> > 2. In fact, the motivations behind experiments *LS* and *ViT backbone* remain unclear:
> >
> > + As noted in the manuscript, Table 2 shows that in some cases, incorporating LS results in even worse performance compared to not using LS. A reasonable explanation for this phenomenon is needed.
> >
> > + I am familiar with existing OSR benchmark datasets, and both traditional (TinyImageNet) and SSBs(CUB, Aircraft and Scars) are, in fact, considered small-scale by current standards. Therefore, the rationale for adopting ViT as the backbone is unclear.
> >
> > 3. Finally, please clarify the differences between OpenAUC and OSCR, and provide detailed information regarding the validation set split.
> >
> > I appreciate works that are concise yet effective. Therefore, I hope the authors can provide improved and more substantial responses. Thanks.

---

> > > ### Author Response · Authors · 2025-11-27
> > >
> > > Dear Reviewer,
> > >
> > >
> > > Thank you for taking the time to review our revised manuscript and provide these additional comments. ​We really appreciate your remaining concerns. We are currently working on the response that addresses each point and we will submit our revised response and manuscript soon.
> > >
> > > Best Regards,
> > >
> > > Authors

---

> ### Author Response · Authors · 2025-12-02
> **Response to Additional Comments by Reviewer u1CY**
>
> We thank the reviewer for acknowledging that our responses could address their concerns and believe the following discussions will address the remaining concerns.
>
> **Comment:**
>
> Although Sec. 3 outlines the relationship between $\tau$ ... ... discussion regarding $P$.
>
> **Response:**
>
> We thank the reviewer for this critical question regarding the theoretical determination of our scheduling parameters. We agree that providing non-empirical justification is paramount.
>
> **The heuristic is designed to maintain structural control:** maximizing performance by ensuring the range is centered precisely around the optimal operating point while avoiding structural collapse caused by excessive perturbation in the temperature, whereas $P$ is not highly sensitive and simply needs to be within a working range that allows for sufficient exploration.
>
> We clarify that the single optimal value for constant temperature $\tau$ is inherently dataset-dependent and determined by empirical tuning.
>
> However, we argue that the heuristic relationship ($\tau^+ = \tau + \Delta$, $\tau^- = \tau - \Delta$) is guided by our analysis in Section 3. Since our gradient analysis established that the optimal $\tau$ provides the best starting trade-off between the instance-specific and class-specific features in the constant temperature regime, our schedule is **engineered to perturb this optimum** (**$\tau$ serves as the center** in our heuristics). **To recap**, we start with a lower value than $\tau$ and transition to a higher value than $\tau$.
>
> Crucially, **if $\Delta$ is chosen too large**, the range becomes too wide, and **the structural stability in the representation space may break down**, as the high $\Delta$ disrupts the cluster boundaries beyond recovery. Starting with excessively low $\tau^-$ disrupts the initial formation of the semantic structure, which may not be recovered later. Furthermore, an excessively high $\tau^+$ encourages over-adjustment from the learned features in the low-$\tau$ phase, pushing the model toward extreme class-specific features and removing instance-level discriminating power.
>
> **Regarding the period $P$,** our ablations confirmed that performance is not highly sensitive to its exact value. This confirms that the schedule's benefit is driven by the smooth traversal of temperature, rather than the absolute duration of the cycle. The period **$P$ simply needs to be within a functional range that allows for sufficient optimization exploration**.
>
> We **added this discussion** briefly in the revised manuscript **(lines 306-307, 311-316 Section 4.3)**.
>
> **Comment:**
>
> In fact, the motivations behind experiments LS and ViT backbone remain unclear:
>
> 1. As noted in the manuscript, Table 2 shows that in some cases, incorporating LS results in even worse performance compared to not using LS. A reasonable explanation for this phenomenon is needed.
>
> **Response:**
>
> We appreciate the reviewer asking for a clearer explanation regarding the occasional negative performance of Label Smoothing (LS) in Table 2.
>
> We reiterate that our **primary objective** remains unchanged: to **investigate whether our proposed temperature schedules provide an orthogonal benefit**, irrespective of the benefits of LS. Our results confirm this, showing **our temperature schedules consistently boost the baseline performance even where LS alone failed**.
>
> **To summarize** the results in Table 2, our schedules can be used together with LS to further boost the performance in a few cases (for the CUB and TinyImageNet with CE loss). Even for cases where LS does not improve the constant baseline performance (for the Stanford cars -‘Hard’ split and Aircraft with CE loss), our NegCosSch **outperforms the corresponding constant baseline**.
>
> **Explanation for LS results in drops.** The reasonable explanation for LS resulting in drops in OSR metrics (like AUROC or OSCR) **in these few cases** is due to its effect on uncertainty ranking: LS causes imbalanced **logit suppression as shown in Xia et al. [2025]**. Specifically, the amount LS suppresses the maximum logit directly corresponds to the true probability of error (or uncertainty). This means LS enforces relatively higher uncertainty on correct known samples and lower relative uncertainty on unknowns, compared to vanilla CE, degrading the ranking needed for rejection (Xia et al. [2025]).
>
> Since OSR requires a binary rejection decision based on the model's uncertainty, this **degraded ranking directly hurts** performance, as the model is unable to reliably identify which samples to 'reject' if uncertain.
>
> We briefly added clarifications regarding our results and the explanation for LS-induced performance drops in the revised manuscript **(lines 418-422 Section 5.2)**.
>
> Guoxuan Xia, Olivier Laurent, Gianni Franchi, and Christos-Savvas Bouganis. Towards understanding why label smoothing degrades selective classification and how to fix it. In International conference on learning representations, 2025.

---

> ### Author Response · Authors · 2025-12-02
> **Response to Additional Comments by Reviewer u1CY (Cont.)**
>
> **Comment:**
>
> 2. I am familiar with existing OSR benchmark datasets, and both traditional (TinyImageNet) and SSBs(CUB, Aircraft and Scars) are, in fact, considered small-scale by current standards. Therefore, the rationale for adopting ViT as the backbone is unclear.
>
> **Response:**
>
> We thank the reviewer for the observation regarding the size of the OSR benchmarks. However, our primary focus for this section (Appendix F.4) is **not the dataset scale but the generalizability of the proposed schedules across different model architectures, such as the contemporary vision transformer backbone**, which is a crucial factor in modern deep learning.
>
> We demonstrate that our proposed schedules consistently boost performance regardless of the underlying backbone. While our main results showcase improvements using conventional CNNs (VGG and ResNet), we include the more recent Vision Transformer (ViT) architecture in our evaluation in this section to explicitly **prove architectural robustness**. The results confirm that our temperature modulation provides significant benefits even when integrated into the contemporary vision transformer backbone.
>
> We **refine the discussion in Appendix F.4** to highlight the rationale for adopting ViT in the revised manuscript.
>
> **Comment:**
>
> Finally, please clarify the differences between OpenAUC and OSCR, and provide detailed information regarding the validation set split.
>
> **Response:**
>
> **Difference between the metrics:** We appreciate the concern regarding the clarification on the differences between OpenAUC and OSCR. However, we emphasize that our primary **focus** is rather to demonstrate that our **proposed schedules consistently outperform the baseline** in the majority of cases **irrespective of the metric used**.
>
> **The difference is structural and methodological:** the OSCR curve measures the trade-off between the Correct Classification Rate (CCR) for known samples and the False Positive Rate (FPR) for unknowns across all thresholds, whereas the OpenAUC is a simplified, threshold-free ranking score that can be expressed as the sum of pair-wise loss terms. This pairwise formulation removes the need to calculate the numerical integral with histograms, which involves multiple non-differential operators, thereby making the metric differentiable and suitable for direct optimization.
>
> We provide this detailed comparison, along with a discussion of the differences between OpenAUC and OSCR, in the **new Appendix F.3** of the revised manuscript.
>
> **Validation Split:** We thank the reviewer for their concern regarding the validation set split for hyperparameter tuning. Here are the details in addition to the discussion from our previous response.
>
> While tuning the hyperparameters, we split the training data into a training portion and a validation set. The validation set is constructed by holding out 20% random training data from one known-unknown split. The hyperparameters are selected by maximizing the closed set performance on the validation set.
>
> We agree that these crucial details regarding the hyperparameter tuning require further clarity. We have revised the Training Details subsection **(lines 352-354 in page 7) in Section 5** and Hyperparameters subsection **(lines 843-845 in page 7)** in Appendix D in the revised manuscript.

---

> > ### Author Response · Authors · 2025-12-04
> > **Summary of Comprehensive Responses to Reviewer u1CY**
> >
> > We thank the reviewer for their careful and critical engagement, which led to significant revisions and detailed justifications addressing all the concerns.
> >
> > The concern regarding the heuristic choice of parameters $(\tau^+, \tau^-)$ is addressed by clarifying that the relationship of our heuristic is structurally informed and derived from sensitivity analysis, not by arbitrary choice. This strategy ensures that our schedule is centered around the optimal constant temperature $\tau$ (from tuning), thereby maintaining the structural stability of the learned representation space and preventing collapse caused by excessive width $\Delta$. We provided a discussion supporting this necessity. Furthermore, we clarified that the period $P$ is not highly sensitive, which simply needs to be within a functional range for sufficient optimization exploration (Section 4.3).
> >
> > We addressed the reviewer's confusion regarding the transition from the constant baseline to the motivation for temperature modulation (Section 4.2). We clarified that any constant temperature scaling is locked in a suboptimal trade-off between instance-specific (low $\tau$) and class-specific (high $\tau$) features. Our temperature modulation is the logical solution, designed to smoothly traverse this spectrum and capture the benefits of both regimes for effective OSR. This narrative is strongly supported by a new quantitative clustering diagnosis (Table 3), which demonstrates that our NegCosSch successfully achieves the better geometric structural properties (low intra-class scatter, high inter-class margin, and large distance for unknowns from the known clusters) simultaneously that any fixed-temperature baselines cannot match.
> >
> > We provided justification for statements the reviewer initially deemed unsupported: (1) OSR methods relying on forceful creation of empty spaces often fail because they don't resolve the inherent semantic closeness of hard unknown samples. (2) Statements regarding significant computational overhead refer to existing OSR methods that require data generation or training secondary models, contrasting with our simple schedule, requiring no overhead.
> >
> > Finally, we clarified several issues in our experimental setup. Regarding Label Smoothing (LS), we demonstrated our temperature schedules provide an orthogonal benefit by boosting baseline performance even when LS alone causes performance drops. This degradation is attributed to LS due to logit suppression, which harms the ranking needed for OSR. The Vision Transformer backbone was adopted not due to dataset size, but to demonstrate the architectural robustness and generalizability of our temperature modulation across contemporary models. Finally, we differentiate between the evaluation metrics OSCR and OpenAUC and confirm that our schedules outperform the baseline regardless of the OSR metric/ score, and detailed our hyperparameter tuning protocol using a 20% random validation set split from the training data.
> >
> > We appreciate the reviewer's acknowledgement of our efforts, noting that many concerns have already been addressed. We believe the additional detailed discussion, particularly the structural rationale for the parameter relationships of our heuristic, resolves the remaining points and significantly strengthens the quality and clarity of the paper.

---

### Official Review · Reviewer_T1Y8 · 2025-10-28

**Soundness:** 2
**Presentation:** 3
**Contribution:** 2
**Rating:** 4
**Confidence:** 3

**Summary:**

The authors propose temperature schedules for reprsentation learning
in open set recognition.  They use supervised contrastive loss for
representation learning.  For each sample in a batch, they create two
augmentations. Positive pairs are from the same class.  They use
temperature in softmax to regulate the representation of intra-class
samples (relative to inter-class samples).  According to the
Familiarity Hypothesis (D&G, 2022), most OSR methods flag novelty
based on the absence of semantic features and recommends to extract
"interesting content features" beyond the semantic features. The
authors propose varying the temperature to learn semantic features and
instance ("content") features.  Based on CosSch (starting with higher
temperature and going toward lower), they propose NegCosSch, which
starts from lower temperature and going higher.  Both follows a cosine
function with a certain period.  During the first half of a cosine
cycle, increasing temperature encourages semantic features.  During
the second half of a cosine cycle, decreasing temperature encourages
instance features.  Besides the periodic version (P-NegCosSch), they
propose a monotonic version (M-NegCosSch) which only increases in
temperature.

For evaluation, they compare 6 existing methods over 4 datasets.  On
methods, they added exponential increase and linear increase as well
in addition the two versions of NegCosSch.  The empirical results in
Table 1 indicate their 4 methods generally outperform compared
methods.  However, among their four methods, a generally more
effective methods is not apparent.  When used with different loss
functions, P-NegCosSch and M-NegCosSch generally performs better than
constant temperature, while P-NegCosSch generally outperform the
others.  They also found that their method can benefit from more
classes.

**Strengths:**

1.  For open set recognition, Negative Cosine Schedule (NegCosSch) is
proposed for temperature scheduling in softmax used in supervised
contrastive learning.

2.  The proposed variations of P-NegCosSch and M-NegCosSch together
with exponential increase and linear increase generally outperforms
the other 6 existing methods over 4 datasets.

3.  The paper is generally well written.

**Weaknesses:**

1.  The proposed method is a minor variation of the existing NegCosSch
method for temperature scheduling.  Hence, the novelty level is not
high.

2.  The P-NegCosSch and M-NegCosSch do not seem to perform better than
the simpler exponential increase and linear increase.

3.  The motivation for the periodic temperature schedule could be
further discussed.  Also, it does not seem to outperform the monotonic
increasing version.

4.  Part of the motivation is from semantic features vs
instance/content features, further analysis with NegCosSch would be
significant.

**Questions:**

1.  Since P-NegCosSch does not seem to generally outperform
M-NegCosSch, what are the main reasons?  Could the 2nd half of a cycle
"undo" the 1st half of a cycle?

2.  p6: "Otherwise, P in Eq. (7) can be chosen by dividing E by the
number of cycle?"  How does one choose the number of cycles?

---

> ### Author Response · Authors · 2025-11-21
> **Response to the Comments by Reviewer T1Y8**
>
> **Overall Response:** We thank the reviewer for mentioning the strengths and complimenting the write-up. We appreciate the concerns expressed by the reviewer. We hope that the following discussion will shed more clarity about our work and in re-evaluating the quality of the work. We have uploaded the revised manuscript.
>
>
> **Comment:**  Weaknesses 1: The proposed method is a minor variation of the existing NegCosSch method for temperature scheduling. Hence, the novelty level is not high.
>
> **Response:**
>
> The reviewer expressed concerns regarding the novelty of the work. We would like to clarify and reiterate the key distinctions of our proposed schedule relative to  prior work.
>
> **a. Extension to the open set scenario and for multiple loss types.** The work by Kukleva et al. [2023] along with the other previous works, such as Wang and Liu [2021], analyze the effect of temperature scaling – the use of a higher temperature and a lower temperature, in the context of **closed set self-supervised contrastive learning and applied on long-tailed data**, which we discussed and cited in our work. However, in our work, we **extend the effect of temperature scaling in the context of supervised learning and in an open set scenario** focusing on where the representations of novel samples lie in the representation space. The effect of temperature scaling was not discussed in the previous literature for unknown samples. We demonstrate that our NegCosSch brings improvement for multiple types of losses. The **prior works do not show the applicability of temperature modulation for other loss families, such as the cross-entropy loss**, which is still a widely applied loss function for many applications. Please see **lines 108-114 in Page 3** in the revised manuscript.
>
> **b. Why our temperature modulation is fundamentally better.** Although a cosine temperature scheduling is proposed by Kukleva et al. [2023], the advantage of the cosine scheduling is mostly applied for the self-supervised long-tailed data. Our temperature modulation is **not just a mere variation** of the existing cosine temperature scheduling (CosSch), but the novelty lies in its **clever way for performing modulation** using our negative cosine scheduling to **gain benefits for both open set recognition as well as closed set classification**. For example, if we **observe the results for cosine schedules in Table 1, we observe a significant drop in performance, which demonstrates our NegCosSch is fundamentally different from CosSch.**
>
> **To summarize the benefits**, starting with a low temperature, the model learns a distributed representation space where open set samples have higher distance to the known samples. Gradually increasing the temperature refines the model over the initial coarse representation space, pulling the known samples towards their own cluster, keeping the unknowns distant, thereby achieving benefits for both known and novel samples simultaneously.
>
> **c. Difference with existing OSR approaches.** The existing OSR methods propose to create **more empty regions in the representation space** with additional placeholders or manifold mix-up, such as in Zhou et al. (2021). Forcing the creation of empty spaces in the representation space with additional regularization, such as in the adversarial reciprocal point learning (ARPL) by Chen et al. [2021], does not handle the tougher cases for a better open set recognition (please see lines 40-53 in page 1). In such a case, the model learns such a feature space so that there still **remains semantic proximity between the harder novel and known samples**, making the harder unknown samples unable to be detected. Our work improves the performance with temperature modulation, which traverses spectrum (from instance-specific to class-specific) with our proposed scheduling to explore the trade-off and extract benefits of both the ends.
>
> **d. Significant improvement on the tougher cases.** Finally, we gain significant improvements in the case of the tougher semantic shift benchmarks, which was **mostly ignored in the previous literature** despite having a good number of new open set recognition methods since their introduction.
>
> However, we appreciate the concern of the reviewer regarding the novelty of our work and hope that this discussion clarifies the distinction of our work from the previous literature. In the revised manuscript, we have added brief details for further clarification.
>
> Anna Kukleva, Moritz Böhle, Bernt Schiele, Hilde Kuehne, and Christian Rupprecht. Temperature schedules for self-supervised contrastive methods on long-tail data. arXiv preprint, 2023.
>
> Feng Wang and Huaping Liu. Understanding the behaviour of contrastive loss. In CVPR, 2021.
>
> Chen, G., Peng, P., Wang, X., & Tian, Y. (2021). Adversarial reciprocal points learning for open set recognition. IEEE Tran. on PAMI.
>
> Da-Wei Zhou, Han-Jia Ye, and De-Chuan Zhan. Learning placeholders for open-set recognition. In CVPR, 2021

---

> ### Author Response · Authors · 2025-11-21
>
> **Comment:** Weakness 2: The P-NegCosSch and M-NegCosSch do not seem to perform better than the simpler exponential increase and linear increase.
>
> **Response:**
> We thank the reviewer for their careful insight.
>
> We want to reiterate that the **linear and exponential increases are not existing baselines, rather we also propose these schedules** in this paper **alongside our negative cosine-based** schedules (P-NegCosSch, M-NegCosSch). While our NegCosSch outperforms in most of the cases, our **claim is that the collective gain by the proposed temperature modulation scheme (by our schedules) is fundamentally superior** to the most common constant baselines while implementing the loss functions, backed with reasonable explanations. We aim to clarify the issue better in our paper's presentation.
>
> Our results show that all of our proposed schedules (P-NegCosSch, M-NegCosSch. linear, and exponential) consistently outperform the constant baseline across all datasets and metrics. The reviewer is correct that the performances among our different proposed schedules are highly competitive and a single one may not win in all cases.
>
> Please note that we underline the best results in Table 1 while comparing the different temperature schedules. In terms of the single best result across a column (total 18 cases – 5 cases each for CUB, Aircraft and Scars and 3 cases for TinyImageNet), our **P-NegCosSch achieves the best results at a maximum number (8 out of 18) of cases and M-NegCosSch wins at 4 cases**.
>
> While simple monotonic schedules like linear and exponential achieve better results in some cases, our NegCosSchs achieve better results in most of the cases, especially across all three SSBs (CUB, Aircraft, and Scars). This demonstrates the generality and robustness of our approach.
>
> We understand that this issue needs to be addressed in the main results. We address the issue in the revised manuscript **(Lines 362-365 in Section 5.1)**.
>
> **Comment:**
>
> Weakness 3: The motivation for the periodic temperature schedule could be further discussed. Also, it does not seem to outperform the monotonic increasing version.
>
> **Response:**
>
> We acknowledge the reviewer’s concern that the reason for periodicity requires further clarification.
>
> While it is true that M-NegCosSch demonstrates competitive overall results, we maintain that P-NegCosSch offers a distinct, critical value. The reason is that the optimal **solution is inherently data- and task-dependent**. The success of M-NegCosSch shows that the primary benefit is derived from the smooth negative cosine increase, which is a significantly good schedule (argued in Section 4.3). **Periodicity refines the solution, potentially discovering a better solution, which helps to improve the performance in some critical cases**.
>
> **To recap:** we argued that starting with a low temperature, the model provides priority to fewer neighbors, learning the coarse structure of representation space, resulting in sharper boundary and all samples remain distributed. As temperature increases, the model prioritizes more neighbors and gradually pulls the positive samples to their own cluster. The unknown samples remain distant because the feature extractor refines over the coarse representation space learned earlier.
>
> Here the optimizer has no mechanism to refine, even if a much better solution exists nearby. In order to find a better feature extractor, we need to explore the trade-off between instance-specific and group-wise features. We need to continue the exploration multiple times so that the solution does not get stuck in the first suboptimal solution. The periodic restarts act as a mechanism to **refine** the feature extractor. Each time a cycle begins, the model is encouraged to refine its previously learned feature extractor and explore the feature space again, **potentially discovering a better solution** over the course of training.
>
> **P-NegCosSch outperforms M-NegCosSch in several cases.**—such as for the **CUB and Aircraft datasets on CE and BackMix** (as demonstrated in our new implementation results in Table 2), and for the open set metrics on **TinyImageNet with ARPL and BackMix** —which confirms that refining the initial solution with **periodicity can help significantly** when tackling nuanced feature spaces. The added periodicity provides this control for specific, challenging problems.
>
> Moreover, the work in Kukleva et al. [2023] introduces modulating the temperature in a periodic manner to improve the self-supervised long tailed recognition.
>
> Ultimately, our main claim is that **our proposed schedules are superior to the constant baseline**. We acknowledge the strong competitiveness of the M-NegCosSch and have briefly added details in the updated manuscript (lines 279-282 section 4.3) to clarify the motivation for both schedulers: M-NegCosSch is the robust, one-parameter-less choice, while P-NegCosSch is the necessary for refining the feature space on the hardest cases.

---

> ### Author Response · Authors · 2025-11-21
>
> **Comment:** Weakness 4. Part of the motivation is from semantic features vs instance/content features, further analysis with NegCosSch would be significant.
>
> **Response:**
>
> We thank the reviewer for identifying this crucial gap. We agree that the transition from the constant baseline to the follow up discussions require further clarification and analysis. First we try to clarify further on the semantic features vs instance features and then, we aim to provide **further analysis based on the geometric properties of the learned representation space**. **Our proposed schedule simultaneously achieves a lower intra-class scatter, a higher inter-class margin and a higher average distance from unknowns to prototypes.** We believe this quantitative analysis significantly boosts our representation learning narrative and provides a crucial justification for the performance improvements observed. We have **revised section 4.2** and the results and discussions are included in the revised manuscript **(Appendix F.1, Table 3)**.
>
> **Additional Clarification.** In section 4.2, we discussed the extreme ends of having either too class-specific or instance-specific representation spaces. To reiterate, a lower temperature forces the model toward learning instance-specific features. The model cannot generalize. A higher temperature forces the model toward class-specific (semantic) features. It encourages smoother decision boundaries and better generalization. However, the unknown representations also get close to the known clusters, resulting in high open set risk.
>
> Every constant temperature is locked into a compromise between these two extremes and the middle value of temperature can only achieve a trade-off between these two. However, for effective OSR, a model needs to capture a delicate combination of both the desirable properties– good class-specific representations while having room for instance-level discriminating power within the class. The way to minimize the total OSR risk is to dynamically leverage the benefits of both regimes.
>
> Therefore, we introduce the cosine schedule in Section 4.3 as a logical solution: it is a smooth function designed to traverse the spectrum from instance-specific representation learning to class-specific generalization. This systematic, smooth adjustment maximizes the model's ability to maintain tight clusters for known classes while still generalizing effectively for the unknown samples.
>
> **Further Analysis.**  To demonstrate the observed effects, we now conduct a **quantitative clustering diagnosis** to strengthen our representation learning narrative (presented in Table 3 Appendix F.1 in the revised manuscript)
>
> We now compute the geometric properties in the representation space for our proposed schedules and the constant baselines: intra/inter-class scatter, and the average distance of unknown samples to the nearest prototype.
>
> Our analysis of the constant temperature baselines reveals the critical OSR trade-off discussed in Figure 1 and in Section 3. A **lower temperature** achieves a **higher intra-class scatter and a lower inter-class margin** (because of the instance specific features), indicating poor separation. However, this also achieves a **higher average distance from unknown samples to the nearest prototype** of known classes, indicating a **desirable property for better OSR**.
> A higher temperature achieves a lower intra-class scatter and a higher inter-class margin, resulting in better closed set separation.
>
> Table: Representation space geometry analysis.
> |  | | CUB | | | Scars  | |
> | :--- | :--- | :--- | :--- | :--- | :---  | :---  |
> | Schedule  | intra-class scatter ($\downarrow$) | inter-class margin ($\uparrow$) | unknowns' distance to nearest prototype ($\uparrow$)  | intra-class scatter ($\downarrow$) | inter-class margin ($\uparrow$) | unknowns' distance to nearest prototype ($\uparrow$) |
> | Const. ($\tau = 0.5$)  | 0.1463  | 0.5503  | $\textbf{0.4172}$   | 0.0919  | 0.6587  | $\textbf{0.3869}$  |
> | Const. ($\tau = 2.0$)  | $\textbf{0.1252}$  | $\textbf{0.7805}$  | 0.3974   | $\textbf{0.0609}$  | $\textbf{0.9129}$  | 0.3797  |
> | Const. ($\tau = 1.0$)  | 0.1349  | 0.6586  | 0.41   | 0.0754  | 0.7923  | 0.3814  |
> | P-CosSch  | 0.1287  | 0.6832  | 0.4037   | 0.072  | 0.8127  | 0.3826  |
> | P-NegCosSch  | $\textbf{0.1318}$  | $\textbf{0.7741}$  | $\textbf{0.423}$    | $\textbf{0.068}$  | $\textbf{0.8951}$  | $\textbf{0.4002}$  |
>
>  The ideal OSR solution requires simultaneously achieving a lower intra-class scatter and a high inter-class margin and a higher average distance from unknowns to prototypes. The table demonstrates that our proposed schedule (NegCosSch) successfully achieves **towards both the desirable properties**. This structural superiority confirms that our schedule better utilizes the entire representation space.

---

> ### Author Response · Authors · 2025-11-21
>
> **Comment** Question 1: Since P-NegCosSch does not seem to generally outperform M-NegCosSch, what are the main reasons? Could the 2nd half of a cycle "undo" the 1st half of a cycle?
>
> **Response:**
>
> Thank you for your question. As discussed before and noted in the manuscript, M-NegCosSch is overall a good temperature scheduling scheme that seems to perform well across the different benchmarks. **In some cases, it outperforms P-NegCosSch. This indicates that the second half of the cycle may push toward the instance specific representation learning** (Please see lines 279-281 in page 6). However, as discussed in the previous response, the optimal solution is inherently data- and task-dependent. **P-NegCosSch achieves outperforms in several cases**—such as for the CUB and Aircraft datasets on CE and BackMix (as demonstrated in our new results in Table 2), and for open set metrics on TinyImageNet dataset with ARPL and BackMix baselines. This confirms that **refining the initial solution with periodicity can help significantly** in achieving an improved representation space depending on the data characteristics.
>
> We hope this discussion clarifies the issue.
>
> **Comment** Question 2: p6: "Otherwise, P in Eq. (7) can be chosen by dividing E by the number of cycle?" How does one choose the number of cycles?
>
> **Response:**
>
> We thank the reviewer for this question, which has helped us clarify an important aspect of our proposed method.
>
> We agree that the selection of P (the period) or the number of cycles introduces a new hyperparameter that may require careful tuning. However, we find that the performance of our P-NegCosSch is not highly sensitive to the precise value of P. We performed an ablation study for $P \in$ {100, 200} on TinyImageNet (reported in Table 6 in Appendix F.4) and found that the **performance was not significantly impacted by this choice**.
>
> Based on this observation, our recommendation is as the following:
>
> For P-NegCosSch, we found P=200 can be a robust default choice that performs well across our benchmarks as we demonstrated in most of the results. The main takeaway is that the **users do not need to invest in a comprehensive tuning for P; a reasonable value is sufficient.**
>
> Users who wish to avoid this hyperparameter entirely can adopt our **M-NegCosSch method as a competitive alternative** as shown in most of our results in Table 1 and 2 that does not require setting a period P, thus incurring no tuning expense for this parameter. However, the periodic version offers feature space refinement as explained in our previous response.
>
> Therefore, our schedules provide a strong alternative to standard baselines without requiring extensive tuning for P. We have clarified in Section 4.3, lines 305- 307 of the revised paper.

---

> > ### Author Response · Authors · 2025-12-04
> > **Summary of Comprehensive Responses to Reviewer T1Y8**
> >
> > We are thankful for the reviewer's feedback and have significantly revised the manuscript to fully address all concerns.
> >
> > The reviewer expressed concerns regarding the novelty of our approach. We clarify that our work introduces a novel extension of temperature scaling to supervised open set scenarios, focusing on managing unknown sample representations, which was not covered by prior self-supervised or closed-set studies. The core novelty lies in our Negative Cosine Scheduling (NegCosSch), a clever modulation designed to sequentially benefit both OSR and closed-set classification for multiple loss families. We substantiated this distinction by showing that NegCosSch significantly outperforms the regular Cosine Schedule (CosSch) (Table 1).
> >
> > We addressed concerns regarding the performance of our various schedules (P-NegCosSch, M-NegCosSch, linear, exponential). We clarified that all proposed schedules consistently outperform the constant baseline. Furthermore, P-NegCosSch achieves the single best result in the majority of cases in Table 1, demonstrating its general robustness. Regarding the motivation for the periodic scheduler, we explained that while M-NegCosSch is a strong, one-parameter-less choice, P-NegCosSch offers critical value by enabling feature space refinement. This periodicity helps significantly in achieving better representations, as confirmed by its improved performance in specific challenging cases (Table 2).
> >
> > Finally, we strengthened our representation learning narrative by performing a quantitative geometric analysis (Table 3, Appendix F.1). This analysis demonstrates that our NegCosSch successfully achieves towards the ideal OSR goal—simultaneously realizing low intra-class scatter, high inter-class margin, and large distance for unknown samples from the known class clusters—a structural improvement that any fixed-temperature baselines cannot match. We also clarified that $P=200$ serves as a robust default choice for P-NegCosSch, making extensive tuning unnecessary.

---

### Official Review · Reviewer_CfPf · 2025-11-02

**Soundness:** 3
**Presentation:** 3
**Contribution:** 3
**Rating:** 4
**Confidence:** 4

**Summary:**

This paper investigates the open-set recognition (OSR) problem, aiming to enhance the model's ability to identify samples of novel semantic classes unseen during training when tested. The authors propose a modulated representation learning method that dynamically adjusts the temperature parameter in the loss function by introducing temperature scheduling, especially a novel negative cosine scheduling. This enables the model to focus on instance-level features to form rough decision boundaries in the early stages of training and gradually transition to semantic-level features to smooth the boundaries. Experiments demonstrate that the proposed method improves both OSR and closed-set performance across various loss functions and benchmark datasets without additional computational overhead.

**Strengths:**

The idea of temperature scheduling is novel and insightful. In particular, the negative cosine scheduling achieves a smooth transition of task switching by reverse-adjusting the temperature, which is more effective than fixed temperature or traditional cosine scheduling. The method does not require introducing additional regularization or generating synthetic samples, reducing computational costs and providing a concise yet efficient solution to the OSR problem.

**Weaknesses:**

Although comparisons with baseline methods (e.g., ARPL, SupCon) have been conducted, some recent OSR methods based on augmentation or multi-expert models (e.g., Wang et al. 2024) are not included. It is recommended to incorporate a broader range of comparisons in the rebuttal or future work to fully demonstrate the superiority.
The performance improvement is limited on benchmarks with a small number of training classes (e.g., CIFAR), indicating that the method may be more suitable for scenarios with a rich number of classes. The authors should explicitly discuss this limitation and explore ways to extend it to small-scale data.
The selection of the temperature range (τ⁺, τ⁻) and period P relies on heuristic rules (e.g., τ⁺=τ+Δ). Despite parameter tuning, there is a lack of theoretical guidance. It is suggested that the authors analyze the boundaries of parameter sensitivity or provide adaptive adjustment strategies to enhance robustness.

**Questions:**

1. On Page 1 (near line 242), does "group-wise features" specifically refer to "class-wise features" (i.e., class-level features)? If so, it is recommended to uniformly use the standard terminology "class-wise" to avoid terminology confusion. If not, please explain the specific differences between "group-wise features" and "class-wise features" to help readers better understand the technical perspectives in the paper.
2. The summation index a in Formula (2) is not defined. Does a iterate over the set A(i), representing all samples in the batch except the anchor i?

---

> ### Author Response · Authors · 2025-11-21
> **Response to the comments by Reviewer CfPf**
>
> **Overall Response:** We thank the reviewer for mentioning the strengths and novelty of our work and expressing their constructive concerns. We hope that the following discussions will provide additional clarifications for a better understanding of the work and in re-evaluating the quality of the work. We believe that addressing the concerns will significantly improve the quality of the paper.
>
> **Comment:** Weakness: Although comparisons with baseline methods (e.g., ARPL, SupCon) have been conducted, some recent OSR methods based on augmentation or multi-expert models (e.g., Wang et al. 2024) are not included.
>
> **Response:**
>
> We thank the reviewer for this excellent suggestion to evaluate our method's complementarity with stronger, contemporary OSR baselines. This helps us clarify that **our proposed scheduling is an independent contribution that can enhance the recent OSR methods**.
>
> As requested, we have **applied our proposed schedules** (P-NegCosSch and M-NegCosSch) on top of the recent OSR method– **BackMix by Wang et al. (2025)** (from their official implementation). The BackMix is an augmentation-based scheme designed to improve OSR by reducing learned spurious correlations between an image's foreground and its background.
>
> First, to validate our setup, we have reproduced BackMix on TinyImageNet and achieved 81.23% of AUROC, which is consistent with the 80.4% of AUROC reported in the original paper.
>
> From the table below, we observe that our **proposed schedules boost the performance of the BackMix baseline across all three metrics** in the majority of cases. This is a significant finding, as it shows our method's benefits generalize beyond simple baselines to SOTA augmentation-based frameworks.
>
> Table: Performance on **BackMix baseline** both including and without the proposed schedules. Results shown on Easy/Hard splits.
> | | | CUB | | | Aircraft | |
> | :--- | :--- | :--- | :--- | :--- | :--- | :--- |
> | Method | Acc | AUROC | OSCR | Acc | AUROC | OSCR |
> | Constant | 82.12 | 82.39 / 72.99 | 67.94 / 60.32 | 90.53 | 92.41 / 82.47 | 83.75 / 75.09 |
> | M-NegCoSch (ours) | $\textbf{82.84}$ | $\textbf{83.97 / 74.66}$ | $\textbf{69.71 / 62.1}$ |$\textbf{ 91.37}$ | 92.2 / 84.43 | $\textbf{84.19 / 77.36}$ |
> | P-NegCoSch (ours) | $\textbf{83.98}$ | $\textbf{84.54 / 74.13}$ |$\textbf{ 71.23 / 62.66}$ | $\textbf{91.49}$ | 92.3 / 83.68 | $\textbf{84.35 / 76.72}$ |
> | | | Scars | | | TinyImageNet | |
> | Constant | 96.81 | 93.23 / 84.39 | 90.33 / 81.82 | 82.32 | 81.23 | 67.1 |
> | M-NegCoSch (ours) | $\textbf{97.52}$ | $\textbf{94.86 / 86.48}$ |$\textbf{ 92.56 / 84.48}$ |$\textbf{ 82.6}$ | $\textbf{81.62}$ | $\textbf{67.37}$ |
> | P-NegCoSch (ours) | $\textbf{97.37}$ | $\textbf{94.76 / 86.04}$ | $\textbf{92.31 / 83.89}$ | $\textbf{82.5}$ | $\textbf{81.72}$ | $\textbf{67.42}$ |
>
> We have added these results in the revised manuscript **(Section 5.2, Table 2). These new results significantly strengthen the paper by demonstrating the robustness and general applicability of our schedules.**
>
> **Multi-expert approach by Wang et al. (2024).** we note that its **architecture multiplies the model size by the number of experts** after some shared layers. This can introduce computational overhead when training on larger benchmarks with models like ResNet50. Moreover, all the results in our manuscript are reported keeping the model architecture fixed. The paper reported OSR performance only on the regular OSR benchmarks. If we **compare the reported open set AUROC on TinyImageNet (0.80) against our simple P-NegCoshSch on cross-entropy loss with label smoothing (83.02)**, we can infer that a basic inclusion of our proposed schedule can outperform the method without incurring such cost.
>
> Wang, Y., Mu, J., Huang, H., Wang, Q., Zhu, P., & Hu, Q. (2025). BackMix: Regularizing Open Set Recognition by Removing Underlying Fore-Background Priors. IEEE Transactions on Pattern Analysis and Machine Intelligence.
>
> Wang, Y., Mu, J., Zhu, P., & Hu, Q. (2024, March). Exploring diverse representations for open set recognition. In Proceedings of the AAAI Conference on Artificial Intelligence (Vol. 38, No. 6, pp. 5731-5739).

---

> ### Author Response · Authors · 2025-11-21
>
> **Comment:** Weakness: It is recommended to incorporate a broader range of comparisons in the rebuttal or future work to fully demonstrate the superiority.
>
> **Response:**
>
> We appreciate the reviewer's concern to include a broader range of comparisons. **Besides incorporating the BackMix** as discussed above, we have **significantly expanded our experimental scope** in the following ways.
>
> **1. Architectural and Comparative Breadth.**
>
> **ViT Architecture:** we implement a vision transformer model on all three SSBs and combine the results in Table 8. The observed improvements demonstrate the generalizability of the proposed schedules on the more recent ViT backbone (Appendix F.4).
>
> **Prototypical methods:** Proposed schedules have shown performance improvements on the prototypical contrastive learning (Appendix F.7).
>
> **2. Analytical and Statistical Strength.** to demonstrate the superiority of the proposed schedule, we have incorporated:
>
> **Scoring robustness:** a comprehensive study showing that the **benefits of our schedules are maintained irrespective of the OSR scoring rule** (e.g., Energy, ODIN, confidence, OpenAUC, Cosine- head) (Table 7).
>
> **Feature space analysis:** We have introduced a new discussion to analyze the geometry of the learned feature space, demonstrating that our schedules result in **improved intra/inter-class scatter and separability of unknowns** to the known clusters (Table 4, Appendix F.1).
>
> **Statistical Significance:** We performed Wilcoxon signed-rank tests across all relevant metrics to rigorously **validate the significance** of the observed improvements (Table 6).
>
> These comprehensive additions significantly expand our evaluation and clarify the general applicability of our proposed temperature schedules.
>
> **Comment:**
>
> The performance improvement is limited on benchmarks with a small number of training classes (e.g., CIFAR), indicating that the method may be more suitable for scenarios with a rich number of classes. The authors should explicitly discuss this limitation and explore ways to extend it to small-scale data.
>
> **Response:**
>
> Thank you for your comment. The reviewer is correct in stating this concern regarding the small improvements on the CIFAR benchmarks and the limitation should be discussed in the main text.
>
> The proposed temperature modulation results in only a small amount of improvement in the open set AUROC and similar closed set accuracy for the cases where the number of training classes is small. We suspect this because with a low number of training classes, it is easier for the baseline model (with a fixed temperature) to find the optimal relationship between each class in the representation space, where it becomes more difficult with more training classes and our modulation can gain higher benefits in these tougher cases. Moreover, the open set recognition AUROC on the CIFAR+10 and CIFAR+50 benchmarks achieves substantial performance with the baseline models, leaving very little scope for improvement, whereas **the tougher SSBs achieve lower baseline OSR performance, where the benefits of our proposed schedules are realized the most**.
>
> Although we **discussed this limitation previously in Appendix F.6, we have explicitly added a brief discussion on this limitation** in the main draft of the updated manuscript (See lines 486 - 490, Section 5.3 in page 10). We leave exploring ways to improve the temperature modulation to small-scale data for future work considering its separate focus and limited rebuttal period.

---

> ### Author Response · Authors · 2025-11-21
>
> **Comment:** The selection of the temperature range (τ⁺, τ⁻) and period P relies on heuristic rules (e.g., τ⁺=τ+Δ). Despite parameter tuning, there is a lack of theoretical guidance. It is suggested that the authors analyze the boundaries of parameter sensitivity or provide adaptive adjustment strategies to enhance robustness.
>
> **Response:**
>
> We thank the reviewer for this comment, which has helped us clarify an important aspect of our proposed method. We confirm that **we derived our parameter heuristics from a systematic analysis**.
>
> Our claim is that users do not need to explicitly tune for both $\tau^+$ and $\tau^-$. To support this, we show that the proposed heuristics, such as $\tau^- = \tau - \Delta$, $\tau^+ = \tau + \Delta$, **are not mere heuristics but they are guided by parameter sensitivity analysis**.
>
> We reiterate and **highlight** the experiments presented in **Figure 4** Appendix F.5. For example, in Figure 4(a), we **varied the range $(\tau^+, \tau^-)$** for our proposed schedules and $\tau$ for the constant baseline by choosing them from {0.025, 0.05, 0.1, 0.2, 0.3, 0.4, 0.5} while implementing SupCon on TinyImageNet and performed the sensitivity analysis. We present the results in the form of quadruplet-wise comparisons in Figure 4 among the obtained performances. In these quadruplet comparisons, we compare our NegCosSch($\tau^+$, $\tau^-$) with CosSch($\tau^+$, $\tau^-$)  and Const($\tau$) when the constant temperature is either set to $\frac{1}{2}(\tau^+ + \tau^-)$ or to $\tau^-$. We observe that our proposed TS yields a better performance than CosSch and constant temperatures for most of the quadruplet comparisons. As per the suggestion, we present exactly the same results of Figure 4(a) in the Tables below.
>
> Table: AUROC for the constant baseline by varying $\tau$.
> | $\tau$ | 0.05 | 0.1 | 0.2 | 0.3 | 0.4 | 0.5 |
> | :--- | :--- | :--- | :--- | :--- | :--- | :--- |
> | AUROC | 82.02 | 82.34 | $\textbf{82.85}$ | 82.80 | 82.60 | 82.13 |
>
> Table: AUROC for NegCosSch by varying $\tau^-$ and $\tau^+$.
> | $\tau^-$ | 0.05 | 0.1 | 0.2 | 0.3 | 0.4 |
> | :--- | :--- | :--- | :--- | :--- | :--- |
> | $\tau^+$ |
> | 0.1 | 82.53 |
> | 0.2 | 82.66 | 82.63 |
> | 0.3 | 82.65 | 82.76 | $\textbf{83.03}$ |
> | 0.4 | 82.61 | $\textbf{83.09}$ | $\textbf{83.03}$ | $\textbf{82.93}$ |
> | 0.5 | 82.53 | 82.72 | 82.65 | 82.63 | 82.33 |
>
> **These results derive our proposed heuristics** that for a given constant temperature, we can construct the range in this manner $\tau^- = \tau - \Delta, \tau^+ = \tau + \Delta$ and this works reasonably well without the need to extensive tuning both $\tau^+$ and $\tau^-$. The proposed heuristic is a principled simplification, derived directly from the observed performance trends in Figure 4.
>
> **Regarding the choice of $P$:** Based on our ablations (Table 9), we found P=200 can be a robust default choice for P-NegCosSch that performs well from most of our results across the benchmarks. The main takeaway is that the **users do not need to invest in a comprehensive tuning for P**; a reasonable functional value is sufficient that allows for sufficient optimization exploration. Users who wish to **avoid this hyperparameter** entirely can adopt our **M-NegCosSch method as a competitive alternative** as shown in most of our results in Table 1 and Table 2. M-NegCosSch does not require setting a period P, thus incurring no tuning expense for this parameter.
>
> This allows users to rely on the derived heuristics instead of incurring the expense of a full parameter sweep for every new dataset.
>
> We reiterate that the heuristic relationship between parameters, which were actually derived from extensive parameter sensitivity. However, we understand that it was not well mentioned in the previous submission. We have **added these details for further clarification (lines 305 - 313 in Section 4.3)** in the revised manuscript.

---

> ### Author Response · Authors · 2025-11-21
>
> **Comment:** Weakness: Despite parameter tuning, there is a lack of theoretical guidance.
>
> **Response:**
>
> The authors appreciate the concern of the reviewer regarding the lack of theoretical guidance. Below, we aim to provide justification for the proposed heuristics.
>
> We clarify that an optimal constant temperature $\tau$ is inherently dataset-dependent and must be determined by empirical tuning.
>
> We argue that the heuristic ($\tau^+ = \tau + \Delta$, $\tau^- = \tau - \Delta$) is guided by our analysis in Section 3. Since our gradient analysis established that the optimal $\tau$ provides the best starting trade-off between the instance-specific and class-specific features for constant baseline, **our schedule is engineered to perturb this optimum ($\tau$ serves as the center)**.
>
> Crucially, if $\Delta$ is chosen large, the **range becomes too wide, and the structural stability in the feature space may break down**, as high $\Delta$ disrupts the cluster boundaries beyond recovery. Starting with excessively low $\tau^-$ disrupts the initial formation of the semantic structure, which cannot be recovered later. Furthermore, an excessively high $\tau^+$ encourages over-adjustment from the learned features in the low-$\tau$ phase, pushing the model toward extreme class-specific features and removing instance-level discriminating power.
>
> Therefore, the heuristic is designed to maintain control: maximizing performance by ensuring **the range is centered precisely around the optimal operating point while avoiding structural collapse caused by large widths in the temperature range.**
> We added this discussion briefly in the revised manuscript (lines 306-307, 311-316 Section 4.3).
>
> **More on theory:**
>
> Please note that we derive the effects of temperature scaling with gradient analysis (Section 3). The results align with the theoretical analysis from existing literature for the self-supervised case. For example, the work in Wang and Liu [2021] calculated gradients of losses with respect to pairwise similarity to see the effect of a fixed temperature scaling. Moreover, Kukleva et al. [2023] analyzed the gradient w.r.t. the representation of a negative sample.
>
> Essentially, we utilize the limitations of constant scaling in both extremes to **propose a novel and clever temperature modulation.** To summarize the benefit: starting with a **lower temperature**, the model learns a **distributed representation space** where unknown samples have higher distances to the known samples. Gradually **increasing the temperature then pulls the known samples towards their own cluster, keeping the unknowns distant**, thereby achieving benefits for both known and novel samples simultaneously.
>
> **Quantitative clustering diagnosis.** We compute the geometric properties in the representation space, such as intra/inter-class scatter, and the average distance from unknown samples to the nearest known class-prototype, to demonstrate and strengthen our representation learning narrative (**Table 3 in the revised manuscript**).
>
> Our analysis of the constant temperature baselines reveals the critical OSR trade-off discussed in Figure 1 and in Section 3. A **lower temperature achieves a higher intra-class scatter and a lower inter-class margin** (because of the instance-specific features), indicating poor separation. However, this also achieves a **higher average distance from unknown samples to the nearest prototype** of known classes, indicating a **desirable property for better OSR**.
> A higher temperature achieves a lower intra-class scatter and a higher inter-class margin, resulting in better closed set separation.
>
> Table: Representation space geometry analysis.
> |  | | CUB | | | Scars  | |
> | :--- | :--- | :--- | :--- | :--- | :---  | :---  |
> | Schedule  | intra-class scatter ($\downarrow$) | inter-class margin ($\uparrow$) | unknowns' distance to nearest prototype ($\uparrow$)  | intra-class scatter ($\downarrow$) | inter-class margin ($\uparrow$) | unknowns' distance to nearest prototype ($\uparrow$) |
> | Const. ($\tau = 0.5$)  | 0.1463  | 0.5503  | $\textbf{0.4172}$   | 0.0919  | 0.6587  | $\textbf{0.3869}$  |
> | Const. ($\tau = 2.0$)  | $\textbf{0.1252}$  | $\textbf{0.7805}$  | 0.3974   | $\textbf{0.0609}$  | $\textbf{0.9129}$  | 0.3797  |
> | Const. ($\tau = 1.0$)  | 0.1349  | 0.6586  | 0.41   | 0.0754  | 0.7923  | 0.3814  |
> | P-CosSch  | 0.1287  | 0.6832  | 0.4037   | 0.072  | 0.8127  | 0.3826  |
> | P-NegCosSch  | $\textbf{0.1318}$  | $\textbf{0.7741}$  | $\textbf{0.423}$    | $\textbf{0.068}$  | $\textbf{0.8951}$  | $\textbf{0.4002}$  |
>
> The ideal OSR solution requires simultaneously achieving **a lower intra-class scatter, a higher inter-class margin and a higher average distance from unknowns to prototypes**. The table demonstrates that **our proposed schedule successfully achieves towards both the desirable properties**. This structural superiority confirms that our schedule better utilizes the entire representation space.

---

> ### Author Response · Authors · 2025-12-03
>
> **Comment:** Question 1. On Page 1 (near line 242), does "group-wise features" ... ... better understand the technical perspectives in the paper.
>
> **Response:**
>
> Yes, these terms refer to the same concept. Thank you for pointing out this issue. All these terms are unified as “class-specific features” in the updated manuscript.
>
> **Comment:** Question 2.The summation index a in Formula (2) is not defined. Does a iterate over the set A(i), representing all samples in the batch except the anchor i?
>
> **Response:**
>
> Yes, the index $a$ in the supervised contrastive loss is iterated over all index in the multi-viewed batch except the anchor index i. The formulation is exactly the same as in the original paper by Khosla et al (2020). Please note that we defined it in Eq. (2) as $a \in I \setminus$ { i }, which is the same as $A(i)$. We define $A(i) = I \setminus $ { i } in the updated manuscript.
>
> Anna Kukleva, Moritz Böhle, Bernt Schiele, Hilde Kuehne, and Christian Rupprecht. Temperature schedules for self-supervised contrastive methods on long-tail data. arXiv preprint arXiv:2303.13664, 2023.
>
> Feng Wang and Huaping Liu. Understanding the behaviour of contrastive loss. In CVPR, 2021.

---

> > ### Author Response · Authors · 2025-12-03
> > **Summary of Comprehensive Responses to Reviewer CfPf**
> >
> > We again thank the reviewer for their constructive feedback. We have rigorously addressed all concerns, resulting in significant improvements incorporated into the revised manuscript.
> >
> > To address the need for comparisons against stronger baselines, we integrated our proposed schedules with the state-of-the-art augmentation-based method, BackMix (Wang et al. 2025). Our new experimental results confirm that our schedules consistently boost the performance of the BackMix baseline, demonstrating that our scheduling is an independent and generalizable contribution (Table 2 in the revision). We also demonstrated that our method provides performance superior to the multi-expert approach (Wang et al. 2024) on TinyImageNet without incurring its architectural overhead. We further expanded the experimental scope by including results on a ViT architecture and Prototypical Contrastive Learning, incorporating significance tests for statistical validation, and confirming the robustness of our schedules across various OSR scoring rules (Table 7).
> >
> > We clarified that our parameter heuristics were derived from systematic parameter sensitivity analysis (Figures 4 and supporting tables). We provided theoretical guidance by explaining that our schedule is designed to strategically modulate the optimal constant temperature, maximizing performance while avoiding structural collapse in the feature space. This is also supported by a new quantitative feature space analysis (Table 3), which demonstrates that our NegCosSch successfully achieves the ideal OSR goal of simultaneously realizing low intra-class scatter, high inter-class margin, and large distance for unknown samples—a combination that fixed-temperature baselines cannot match. Finally, we clarified the observed limitation that the proposed temperature modulation yields only small improvements on CIFAR benchmarks where baseline AUROC rates are already very high, leaving little room for significant gains compared to the tougher SSBs. These additions substantially strengthen the theoretical basis and general applicability of our proposed method.

---

### Author Response · Authors · 2025-11-22
**Overall Response to Everyone**

We appreciate the reviewers' efforts and thank them for their insightful and comprehensive feedback. We are encouraged that the reviewers commend the paper for proposing a **novel, insightful and general temperature modulation** scheme, NegCosSch, for open set recognition (OSR) (Reviewers CfPf, TEAQ).

We appreciate the reviewers' acknowledgement that the idea of repurposing temperature is a **simple yet creative removal of fixed-$\tau$ limitation** that constrain training dynamics (Reviewers CfPf, TEAQ). This approach is pragmatic as it can be **seamlessly integrated into existing OSR pipelines** (Reviewer TEAQ) for **multiple loss families** (Reviewers CfPf, TEAQ) without adding computational overhead (Reviewers CfPf, TEAQ), or generating synthetic samples (Reviewer CfPf).

We are glad that the reviewers find the paper **well-written and easy to follow** (Reviewers T1Y8, u1CY) and that the **proposed schedules are well-motivated** (Reviewer TEAQ). The reviewers also commend the **extensive evaluation** (Reviewer u1CY), which covers multiple benchmarks of varying difficulty, multiple losses, and consistently demonstrates significant improvements over previous baselines (Reviewers T1Y8, TEAQ). The paper's overall **quality is enhanced by the additional experiments** provided in Appendix (Reviewer u1CY), including **gradient analyses and UMAP visualizations** to illustrate how training evolves (Reviewer TEAQ), thereby strengthening  the experiments and paper's logical flow (Reviewer u1CY).

We reiterate the novelty of the proposed work and briefly address the major comments below.

**Novelty.**
Our approach presents a novel contribution to the OSR problem by **extending the temperature scaling**—traditionally applied in closed-set or self-supervised learning contexts—**to the design of more effective representation learning in OSR**. Primarily, we focus on how temperature scaling influences the resulting representations, with particular attention to **improving the discrimination and robustness of representations for novel, unseen classes during inference.** This focus on temperature’s impact in the open settings marks a significant departure from prior work and contributes a fresh perspective to learning more generalizable representation space for OSR.

We propose a **novel temperature modulation** that fundamentally enhances both open and closed task performance by focusing on training a more robust feature extractor. Our strategy learns discriminative features that can separate known classes from potential novel samples by **traversing the instance vs. semantic feature trade-off**, a significant contrast to existing OSR methods that merely rely on pushing boundaries to create vacant regions.

We demonstrate significant improvements by integrating our temperature scheduling into **multiple loss families**, such as contrastive, cross-entropy and ARPL losses, especially for the **tougher semantic shift benchmarks, mostly ignored in literature.**

**Parameter sensitivity, derived heuristics.** Our derived heuristic relation from the sensitivity tests to set the range $(\tau^+, \tau^-)$ in our schedules from the baseline $\tau$ **works sufficiently well**. This **removes the need to explicitly tune both** values. Our heuristic is also structurally informed–it prevents semantic-structure collapse in the representation space. Users who wish to avoid the period (having minimal impact) can adopt our **M-NegCosSch as a competitive alternative**.

**Representation space analysis.** we strengthen our representation learning narrative with a quantitative geometric analysis (Table 3, Appendix F.1). The analysis demonstrates that our NegCosSch successfully achieves towards the ideal OSR goal—simultaneously realizing low intra-class scatter, high inter-class margin, and large distance for unknown samples from the known-class clusters—a structural improvement that any fixed-temperature baselines cannot match.

**Competitiveness amongst our schedules and periodicity.** We clarify that the collective gain by the proposed schedules (P-NegCosSch, M-NegCosSch, linear, exponential) over baselines indicates the superiority of our temperature modulation. While M-NegCosSch is a strong, competitive, and one-parameter-less choice, P-NegCosSch helps significantly in achieving better representations, as confirmed by its improved performance in some specific cases (Table 2).

Further, we clarify more on semantic features vs instance features, motivation for label smoothing and vision transformer baselines, the limitations and tuning procedure.

Finally, we **significantly expand our experimental scope**. Details in the response below. These comprehensive additions significantly expand our evaluation and clarify the applicability of our proposed schedules. We believe that addressing all the comments has improved our work.

---

> ### Author Response · Authors · 2025-12-02
> **Final Remarks + Manuscript Revision**
>
> Dear Reviewers, ACs, SACs and PCs,
>
> We are glad to let you know that we have carefully incorporated the suggestions into the revised manuscript. In the uploaded revised PDF (edits marked in blue), the **main changes addressing the comments** are:
>
> **Architectural and Comparative Breadth.** To test the general applicability of our schedules, we have included a broader range of comparisons:
>
> **1. SOTA Baseline:** We have implemented and evaluated our schedules on the recent BackMix (TPAMI 2025) baseline. This demonstrates that our schedules offer an additive benefit that enhances contemporary augmentation-based methods, such as the BackMix **(Table 2 Section 5.2)**.
>
> **2. ViT Architecture:** We have implemented a vision transformer (ViT) model to test the generalizability of the proposed schedules on the more recent ViT backbone. The performance improvements on the ViT model along with the main results on VGG and ResNet-based architectures confirm the applicability of our proposed schedules across a range of model architectures. **(Appendix F.4)**
>
>  **3. Prototypical Methods:** We have evaluated our proposed schedules on Prototypical Learning **(Appendix F.7)** to show compatibility of our proposed schedules with prototypical contrastive loss.
>
> **Analytical and Statistical Strength.** to demonstrate the superiority of the proposed schedule, we have incorporated:
>
> **4. Feature space analysis:** We have introduced a new section to analyze the geometry of the learned representation space, demonstrating that our schedules result in superior intra/inter-class scatter and separability of unknown samples to the known clusters **(Table 4, Appendix F.1)**.
>
> **5. Scoring robustness:** a comprehensive study showing that the benefits of our schedules are maintained irrespective of the OSR scoring rule (e.g., Energy, ODIN, confidence, OpenAUC, Cosine- head) **(Table 7, Appendix F.3)**.
>
> **6. Statistical Significance:** We perform statistical significance tests across all relevant metrics to rigorously demonstrate that the observed improvements are statistically significant **(Table 6, Appendix F.2)**.
>
> **Further clarifications** on
>
> **7. The novelty:** please see the response above and **lines 108-114, Section 1** for details.
>
> **8. Parameter sensitivity, derived heuristics and the cost for tuning:** Our empirically observed heuristic from the sensitivity tests to set the temperature range $(\tau^+, \tau^-)$ in our schedules from a constant baseline works well, removing the need to tune both values. Our heuristic is also structurally informed– the heuristic prevents the semantic-structure collapse in the representation space by setting a good operating temperature in the center of the range with a reasonable width. We explain our heuristics from this theoretical aspect **(lines 311-316 Section 4.3)**. Moreover, users who wish to avoid the period (having minimal impact) can adopt our M-NegCosSch as a competitive alternative (Section 4.3).
>
> **9. Rationale for temperature modulation:** We have elaborated on the semantic features vs instance features and added details to further clarify on the rationale for temperature modulation (**Section 4.2**).
>
> 10. **More:--**
>
> We clarify further on the **competitiveness amongst our proposed schedules** (Section 5.2), the **motivation for periodicity** (section 4.3), **motivation for the label smoothing baselines**, why label smoothing may fail in some cases but still our schedules improve the corresponding baseline (Section 5.2), discuss **limitations**, explain the hyperparameter tuning procedure, explain why some of the existing OSR methods incur significant overhead, whereas our schedules do not incur any overhead.
>
> We believe that these new, comprehensive additions significantly expand our evaluation and clarify the general applicability of our proposed temperature schedules.

---

### Meta-Review · Area_Chair_vmCt · 2026-01-06

**Summary:**

This paper proposes temperature-modulated representation learning for open set recognition (OSR), introducing novel schedules (e.g., Negative Cosine Schedule, NegCosSch) that dynamically adjust the temperature parameter in loss functions. The schedules enable models to transition from instance-level to class-specific feature learning, improving both OSR and closed-set performance without computational overhead. Experiments on TinyImageNet and semantic shift benchmarks (SSBs) show consistent gains across cross-entropy, contrastive, and ARPL losses, with larger improvements for datasets with more training classes.

Reviewers valued the method’s simplicity, broad applicability, and empirical rigor. Key concerns centered on novelty, theoretical grounding, parameter sensitivity, evaluation breadth, and statistical validation. The major concerns were addressed by the rebuttal. Overall, the strengths of the paper overwhelms the weaknesses and the paper is recommended for acceptance.

**Reviewer Concerns:**

Key Concerns Raised:

- Novelty and theoretical grounding (TEAQ, T1Y8): Adaptation of existing temperature scheduling to OSR lacks strong theoretical ties to open-set boundary properties; novelty is perceived as curatorial.
- Parameter sensitivity (CfPf, u1CY): Heuristic choices of temperature range (τ^+,τ^−) and period P lack systematic justification or adaptive strategies.
- Evaluation gaps (TEAQ, u1CY): Limited comparisons with SOTA OSR baselines (e.g., BackMix, iCausalOSR); insufficient architectural diversity (ViT deferred to appendix); no domain shift testing.
- Statistical rigor (TEAQ): Missing dispersion (std) and significance testing for reported improvements.
- Clarity of experimental design (u1CY): Unclear motivation for label smoothing (LS) and ViT experiments; ambiguous validation set split and metric distinctions (OSCR vs. OpenAUC).

Addressed in Rebuttal:

- Novelty/theory: Clarified distinction from prior closed-set/self-supervised scheduling; added geometric analysis of representation space (intra/inter-class scatter, unknown sample distance) to link schedules to OSR properties.
- Parameter sensitivity: Derived heuristics from sensitivity analysis (e.g., τ^+=τ+Δ, τ^−=τ−Δ); showed P=200 is a robust default; offered M-NegCosSch as a parameter-reduced alternative.
- Evaluation breadth: Added comparisons with BackMix (SOTA augmentation baseline) and prototypical contrastive learning; included ViT results on SSBs; validated across OSR scoring rules (Energy, ODIN, cosine head).
- Statistical rigor: Reported std across 5 seeds, performed Wilcoxon rank tests, and provided significance p-values.
- Experimental clarity: Explained LS-induced performance drops (logit suppression); justified ViT for architectural robustness; detailed validation set split (20% of training data) and metric differences (OSCR vs. OpenAUC).

Outstanding/Partially Addressed:

- Theoretical depth: While geometric analysis strengthens the narrative, formal links between scheduling and equilibrium/regret minimization in OSR remain limited.
- Domain shift: No cross-dataset OSR or domain shift testing (acknowledged as beyond scope).
- iCausalOSR comparison: Unavailable public implementation prevented fair comparison (noted as future work).

**Reviewer Scores:**

CfPf: 4 → Likely to 6

T1Y8: 4 → Likely to 6

u1CY: 4 → Likely to 6

TEAQ: 6 → Remains 6

---

### Decision · Program_Chairs · 2026-01-26

Accept (Poster)